# The emergence of clusters in self-attention dynamics

**Borjan Geshkovski**
MIT
borjan@mit.edu

**Cyril Letrouit**
MIT
letrouit@mit.edu

**Yury Polyanskiy**
MIT
yp@mit.edu

**Philippe Rigollet**
MIT
rigollet@mit.edu

## Abstract

Viewing Transformers as interacting particle systems, we describe the geometry of learned representations when the weights are not time-dependent. We show that particles, representing tokens, tend to cluster toward particular limiting objects as time tends to infinity. Cluster locations are determined by the initial tokens, confirming context-awareness of representations learned by Transformers. Using techniques from dynamical systems and partial differential equations, we show that the type of limiting object that emerges depends on the spectrum of the value matrix. Additionally, in the one-dimensional case we prove that the self-attention matrix converges to a low-rank Boolean matrix. The combination of these results mathematically confirms the empirical observation made by Vaswani et al. [VSP$^+$17] that *leaders* appear in a sequence of tokens when processed by Transformers.

## 1 Introduction

The introduction of Transformers in 2017 [VSP$^+$17] marked a turning point in the AI revolution, powering breakthroughs in natural language modeling and computer vision. With remarkable empirical success, Transformers enable large language models to compute very powerful representations using the self-attention mechanism. Yet, little is known about the geometric structure of these representations. As the size of these models grows at an astonishing rate, the need to understand their inner workings is becoming a pressing scientific challenge. In this work, we make a first step in this direction by describing the geometry of learned representations.

To provide a transparent presentation of our findings, we take a leaf out of the literature on continuous-time dynamics such as neural ordinary differential equations (ODEs) [CRBD18, Wei17, HR17]. By viewing layers as a time variable, this formalism has emerged as a flexible mathematical framework to implement and study ResNets [HZRS16a] as particular discrete-time versions of a parametrized dynamics of the form

$$\dot{x}(t) = f_\theta(x(t)), \qquad t \in [0, T].$$

Here $\theta$ is the trained parameter of a neural network and $f_\theta$ is characterized by the precise architecture of the ResNet[1]. In turn, an input (e.g., an image) $x(0) \in \mathbb{R}^d$ is mapped to its representation $x(T)$.

Unlike neural ODEs and ResNets, the representation map of Transformers is not solely a function of an individual input $x(0) \in \mathbb{R}^d$ but rather of a set/sequence $(x_1(0), \ldots, x_n(0))$ of $n \geqslant 1$ $d$-dimensional tokens. These tokens then evolve in time by interacting with each other per the self-attention mechanism. Namely, following [SABP22], we view tokens as particles, and the transformer dynamics as an interacting particle system of the form

$$\dot{x}_i(t) = \sum_{j=1}^{n} P_{ij}(t) V x_j(t), \qquad t \in [0, +\infty), \tag{1}$$

---

[1]A classical choice is $\theta = (W, A, b) \in \mathbb{R}^{d \times d} \times \mathbb{R}^{d \times d} \times \mathbb{R}^d$ and $f_\theta(x) = W\sigma(Ax + b)$ where $\sigma$ is an elementwise nonlinearity such as the ReLU ([HZRS16b]).

37th Conference on Neural Information Processing Systems (NeurIPS 2023).

for any $i \in [n]$, where $P_{ij}(t)$ are the entries of a $n \times n$ stochastic matrix $P(t)$, given by

$$P_{ij}(t) := \frac{e^{\langle Qx_i(t), Kx_j(t) \rangle}}{\sum_{\ell=1}^{n} e^{\langle Qx_i(t), Kx_\ell(t) \rangle}}, \qquad (i,j) \in [n]^2. \qquad (2)$$

Here the matrices $Q$ (Query), $K$ (Key), and $V$ (Value) are learned from data. Note that $Q, K$ need not be square. The $n \times n$ matrix $P(t)$ is called *self-attention matrix*. The wording *attention* stems precisely from the fact that $P_{ij}(t)$ captures the attention given by token $i$ to token $j$ relatively to all tokens $\ell \in [n]$. The matrices $Q$ and $K$ in (2) warp the geometry of the input tokens, so that a trained attention matrix contains weights which indicate semantic relations between words. Such conclusions have been drawn in the context of language processing tasks in [VSP+17, Figures 3-5].

Our goal is to showcase the fact that self-attention, which itself is the core novelty of Transformers, entails a clustering effect. To that end, we focus on the pure self-attention dynamics described in (1). In particular, we do not model variations such as multiple heads, feed-forward layers, and layer normalization that are typically adjoined to self-attention dynamics of (1). However, on this last point, we note that our theoretical findings indicate that without any normalization, the dynamics (1) can diverge in some (or even all) directions over time. We leave these additional questions for future research; see Section 6.

## 1.1 Organization of the paper and summary of contributions

The goal of this paper is to characterize clustered representations of a *trained* Transformer by studying the asymptotic behavior of a sequence of tokens $(x_1(t), \ldots, x_n(t))$ as they evolve through the layers of a transformer architecture using the dynamics (1). In this setup, a Transformer is completely described by the weight matrices $(Q, K, V)$ obtained during training. Note that we assume that these three matrices are *time-independent*. While this assumption is motivated by mathematical convenience, it is worth noting that such weight-sharing scenarios are in fact used in practice—see, e.g., ALBERT [LCG+20]—as they drastically reduce the number of parameters of a network.

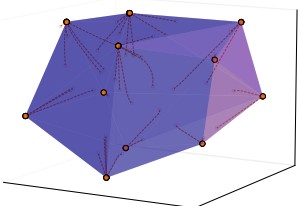

Figure 1: For $V = I_3$ tokens cluster toward the vertices of a convex polytope (Theorem 3.1).

With parameters $(Q, K, V)$ fixed, tokens are subject to collective dynamics that we call *transformer dynamics*. While these dynamics are reminiscent of existing models for opinion dynamics and flocking, they present they own mathematical challenges requiring ad-hoc tools to study their asymptotic behavior.

The main conclusion of our analysis is that the set of tokens $\{x_1(t), \ldots, x_n(t)\}$, appropriately rescaled, tends to a *clustered configuration* as $t \to \infty$. Our theoretical findings justify the empirical observation made in [VSP+17] that *leaders* appear in a sequence of tokens when processed by Transformers. We now list our main contributions.

*(i)* As a warm-up to the geometric characterization of the limits of sequences of tokens, we show in Section 2 that when $d = 1$ and $V > 0$, the self-attention matrix $P(t)$ converges to a low-rank matrix with entries 0 and 1 as $t \to +\infty$ thus revealing the emergence of a small number of leaders that drive the transformer dynamics. The restriction $d = 1$ follows from technical considerations, and some pathological phenomena may occur in higher dimensions (see Remark 5). But numerical experiments (as well as past empirical work) indicate that the result may extend to higher dimensions for almost all initial sequences of tokens.

*(ii)* In Section 3 we first focus on the case $V = I_d$ as a natural canonical choice that enables us to establish some of the main tools of the paper. We introduce a time re-scaling reminiscent of the layer normalization heuristics to alleviate the possible divergence of tokens. We show that along this scale the tokens converge to the boundary of a convex polytope. For almost all initial sequences they even converge to the vertices of the polytope, the number of which is significantly smaller than $n$. This elucidates the clustering phenomenon. (See Fig. 1.) When $V = -I_d$, all tokens following the dynamics (1) collapse to 0.

*(iii)* We build on these results and in Section 4 consider the case wherein $V$ is only assumed to have a simple and positive leading eigenvalue. This setting is much closer to reality and corresponds to

actual learned matrices $V$ (see Figure 10). We show that along the particular timescale, tokens cluster toward one of at most three hyperplanes which are determined by the corresponding eigenvector.

*(iv)* In Section 5 we complete the results of Sections 3 and 4 by addressing the case where the leading eigenvalue has multiplicity. This results in clustering toward the vertices of a convex polytope in some directions, and a linear subspace in the others.

*(v)* We also prove the global existence and uniqueness of solutions of all dynamics considered in this work (including the mean field limit). We refer the reader to Appendix A for more details.

We also observed numerically that our conclusions extend to more compound architectures (see Conjecture 4.2, Section 6 and Appendix F).

| Value | Key and Query | Limit geometry | Reference |
|---|---|---|---|
| $V = I_d$ | $Q^\top K > 0$ | vertices of convex polytope | Theorem 3.1 |
| $\lambda_1(V) > 0$, simple | $\langle Q\varphi_1, K\varphi_1 \rangle > 0$ | union of 3 parallel hyperplanes | Theorem 4.1 |
| $V$ paranormal | $Q^\top K > 0$ | polytope $\times$ subspaces | Theorem 5.1 |
| $V = -I_d$ | $Q^\top K = I_d$ | single cluster at origin* | Theorem C.5 |

Table 1: Summary of the clustering results of this work. *All results except for the case $V = -I_d$ hold for the time-scaled dynamics (4).

**Remark 1** (Discrete time). *While we focus on the idealized setting of self-attention dynamics in continuous-time, this is solely done for convenience and all of our methods are straightforwardly applicable to the discrete-time setting. (See also Remark 4.) The discrete-time analog of* (1) *with time-step $\Delta t > 0$ (equal to 1 in practice) is simply the forward Euler iteration*

$$x_i((k+1)\Delta t) = x_i(k\Delta t) + \Delta t \sum_{j=1}^{n} \left( \frac{e^{\langle Qx_i(k\Delta t), Kx_j(k\Delta t)\rangle}}{\sum_{\ell=1}^{n} e^{\langle Qx_i(k\Delta t), Kx_\ell(k\Delta t)\rangle}} \right) V x_j(k\Delta t), \qquad k \in \mathbb{N}. \quad (3)$$

**Notation.** We denote by $\langle \cdot, \cdot \rangle$ and $\|\cdot\|$ the Euclidean dot product and norm respectively, and we use the shorthand $[n] := \{1, \ldots, n\}$. For any matrix $M \in \mathbb{R}^{d \times d}$, we order its eigenvalues (repeated according to multiplicity) by decreasing order of modulus: $|\lambda_1(M)| \geq \ldots \geq |\lambda_d(M)|$. We denote by $\|M\|_{\mathrm{op}}$ the $\ell^2$—operator norm of the matrix $M$, equal to the largest singular value of $M$. Given a set $S \subset \mathbb{R}^d$, we define the distance of a point $x \in \mathbb{R}^d$ to $S$ as $\mathrm{dist}(x, S) := \inf_{s \in S} \|x - s\|$, and by $\mathrm{conv}(S)$ the convex hull of $S$.

### Related work

Our study and results build on several different lines of work, and we draw some parallels in what follows.

**Analysis of attention-based models.** Given the widespread use of Transformers in natural language processing, there has been a surge of interest in understanding the function and significance of attention layers within these models. In [YBR+20], the authors show that when treated as discrete-time systems with additional dense layers and multiple heads appended to the core attention mechanism, Transformers exhibit the universal approximation property. In [LLH+20], the authors present, to the best of our knowledge, the first interacting particle systems perspective on Transformers. They then leverage the similarities between Transformers (with an additional feed-forward layer compared to (1)) and convection-diffusion equations to slightly improve the performance of Transformers by employing a Strang-Marchuk splitting scheme for time discretization. In [SABP22], the authors interpret system (1) as the characteristics of a continuity equation. Drawing on the similarities between (1) and Sinkhorn iterations, they propose a novel architecture dubbed *Sinkformer*, which possesses the desirable property of being a Wasserstein gradient flow.

**Quadratic complexity of Transformers.** The major computational challenge of Transformers is their high computational complexity, particularly when processing long sequences. Transformers require quadratic time and space complexity to process sequences, because each self-attention layer contains $n^2$ products of the form $\langle Qx_i, Kx_j \rangle$ (for $i, j \in [n]$). The empirical observation that the self-attention matrix $P$ is close to a low rank matrix—see [LWLQ22, Section 4.4] for references—is cited

as the inspiration behind *Linformers* [WLK+20] and the fine-tuning algorithm LoRA [HysW+22]. For both approaches, the low-rank structure is imposed rather than extracted from $P$ itself. Other methods called *sparse attention* and *block attention* have been proposed to reduce the quadratic complexity—see [WLK+20, Section 2.2] for references. In the spirit of these works, a foreshadowing of the clustering mechanism was invoked in [VKF20], where queries are clustered into groups, again in view of reducing the quadratic complexity of self-attention. We point out that [DCL21] previously demonstrated that without skip connections, the dynamics trivializes and all tokens quickly lump together into a single tight cluster. Our work, in contrast, shows that in the presence of skip connections a rich cluster structure emerges.

Compared to the usual BERT, ALBERT [LCG+20] uses parameter-sharing across layers, meaning that the weight matrices $Q, K, V$ in (1)-(2) do not depend on time, as in the present paper. This does not reduce the theoretical $O(n^2)$ complexity of the original Transformer, but, quoting [LCG+20], it "significantly reduce[s] the number of parameters for BERT without seriously hurting performance, thus improving parameter-efficiency. An ALBERT configuration similar to BERT-large has 18x fewer parameters and can be trained about 1.7x faster. The parameter reduction techniques also act as a form of regularization that stabilizes the training and helps with generalization".

**Neural collapse.** Our results and conclusions bear a resemblance to some geometric aspects of neural collapse for classification tasks [PHD20]. A key geometric aspect of neural collapse is the observation that, during the training of deep neural networks, the representation of different classes in the later layers of the network tends to form a tight cluster around the vertices of a simplex. The emergence of a simplex structure in the representation space provides insights into how the neural network organizes and separates the different classes.

**Clustering in interacting particle systems.** The transformer dynamics (1) have a strong connection to the vast literature on nonlinear systems arising in the modeling of opinion dynamics and flocking phenomena. In addition to the classical Kuramoto model describing synchronization/clustering of oscillators [Kur75, ABV+05], the model which is most similar to (1) is the Krause model [Kra00]

$$\dot{x}_i(t) = \sum_{j=1}^n a_{ij}(x_j(t) - x_i(t)), \qquad a_{ij} = \frac{\phi(\|x_i - x_j\|^2)}{\sum_{k=1}^n \phi(\|x_i - x_k\|^2)}.$$

which is non-symmetric in general ($a_{ij} \neq a_{ji}$), much like (1). When $\phi$ is compactly supported, it has been shown in [JM14] that the particles $x_i(t)$ assemble in several clusters as $t \to +\infty$. Other models of opinion dynamics and flocking have been proposed and studied, among which the Vicsek model [VCBJ+95], the Hegselmann-Krause model [HK02] and the Cucker-Smale model [CS07]. These models may also exhibit a clustering behavior under various assumptions (see [MT14, CHH+16, HKPZ19] and the references therein). The transformer dynamics are also closely related to the dynamics employed in mean-shift clustering [Che95], and this work indirectly sheds some light on its theoretical properties.

The analysis of transformer dynamics presents unique mathematical challenges that cannot be addressed using the tools developed for these more primitive models. In particular, our work demonstrates how different choices for the parameters lead to remarkably diverse clustering patterns. Much more remains to be discovered and this work is a first attempt a rigorous mathematical analysis of these synthetic dynamics.

## 2 Asymptotic low-rankness of the self-attention matrix

As mentioned in Section 1.1, numerical experiments in [WLK+20] show that the self-attention matrix $P$, defined in (2), has an almost low-rank structure. This observation has then been leveraged to reduce the quadratic complexity in the sequence length $n$ which is inherent to Transformers, resulting in a non-negligible decrease in the cost of training.

As a warm-up to deriving complete geometric representations of the dynamics, our first result shows, in the simple $1d$ case that $P(t)$ indeed converges exponentially fast toward a matrix which is typically both Boolean and low-rank (see Fig. 3). Although there are clear obstructions to a rigorous extension of this result to higher dimensions (Remark 5), numerical experiments appear to show that this result holds in greater generality, for almost all initial sequences (Appendix F).

To set this up, we introduce the set $\mathscr{P}$ of $n \times n$ matrices having the form illustrated in Fig. 2, where the asterisks denote arbitrary non-negative real numbers which add up to 1. The row of asterisks may actually be any row between the first and the last one.

**Theorem 2.1** (Self-attention matrix converges to a low-rank Boolean matrix)**.** *Let* $d = 1$. *Suppose that the scalars* $(Q, K, V)$ *satisfy* $V > 0$ *and* $QK > 0$. *For any initial sequence of pairwise distinct tokens* $(x_1(0), \ldots, x_n(0)) \in \mathbb{R}^n$, *there exists some* $P^* \in \mathscr{P}$ *such that the self-attention matrix* $P(t)$ *defined in* (2) *converges to* $P^*$ *as* $t \to +\infty$.

$$
P_{\sigma_1}
\begin{bmatrix}
1 & 0 & \ldots & 0 \\
\vdots & \vdots & \vdots & \vdots \\
1 & 0 & \ldots & 0 \\
* & * & \ldots & * \\
0 & \ldots & 0 & 1 \\
\vdots & \vdots & \vdots & \vdots \\
0 & \ldots & 0 & 1
\end{bmatrix}
P_{\sigma_2}
$$

Figure 2: Elements in $\mathscr{P}$, where $P_{\sigma_i} \in \mathbb{R}^{n \times n}$ are some permutation matrices, and asterisks denote arbitrary non-negative reals which add up to 1.

The proof may be found in Appendix B. The rate of convergence toward $P^*$ is in fact doubly exponential in $t$ for coefficients outside the row of asterisks in Fig. 2. The proof the theorem also reveals that for almost all initial sequences of pairwise distinct tokens, $P^*$ is actually of rank 1 or 2, i.e., the row of asterisks is equal to either $e_1 = (1, 0, \ldots, 0)$ or $e_n = (0, \ldots, 0, 1)$.

The interpretation of Theorem 2.1 is that in the $1d$ case, at most three tokens *capture the attention* of all tokens except at most one. Typically, these *leading* tokens are those carrying the largest amount of information. This is also illustrated in Fig. 4. Since the tokens $x_i$ here evolve on $\mathbb{R}$, the right-most and left-most ones (which typically tend toward $\pm\infty$) capture the attention of all the others.

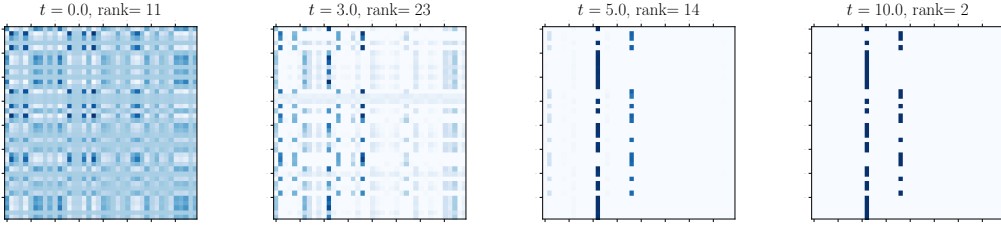

Figure 3: An illustration of the asymptotics of $P(t)$ entailed by Theorem 2.1 for $n = 40$ tokens, with $Q = K = 1$ and $V = 1$. (See Appendix F for details on computing.) Increasing $n$ has no effect on this behavior of $P(t)$—see Fig. 11.

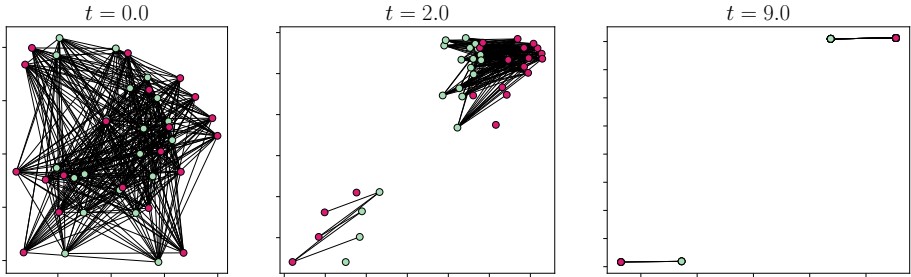

Figure 4: The clouds $\{Kx_i(t)\}_{i \in [20]}$ (green) and $\{Qx_j(t)\}_{j \in [20]}$ (purple) for $d = 2$ where pairwise points of clouds are connected by a line of width equal to $P_{ij}(t)$. Here $V > 0$ and $Q > 0$ are random matrices and $K = I_2$. The creation of clusters is reflected by the rank $\leqslant 2$ structure of the self-attention matrix $P(t)$. This interaction echoes findings illustrated in the original paper [VSP+17]—for instance, Figures 3-5 therein.

## 3  Clustering toward vertices of convex polytopes

In the rest of the paper, we seek to taxonomize various *clustering* results for the solutions to (4) when $t \to +\infty$, depending the sign and the multiplicity of the eigenvalues of $V$. We begin by focusing

on what may appear to be the most natural[2] case $V = I_d$, as is also done in [SABP22]. In fact, we demonstrate (theoretically and numerically) later on, clustering is a generic phenomenon which holds under much less restrictive assumptions.

The transformer dynamics considered in (1) does not contain a layer normalization mechanism typically encountered in practice [VSP+17]. In absence of such a device, tokens may diverge to infinity as in Theorem 2.1. In fact, the norm of the tokens $x_i(t)$ typically diverges exponentially toward $+\infty$ for any $d$: this is expected, by analogy with the non-trivial solutions to $\dot{y}(t) = y(t)$.

To remedy this situation, we take inspiration from the solution $y(t) = e^{tV} y(0)$ to $\dot{y}(t) = V y(t)$. Namely, for any $i \in [n]$ we consider the *rescaled* tokens $z_i(t) := e^{-tV} x_i(t)$, which solve

$$\dot{z}_i(t) = \sum_{j=1}^{n} \left( \frac{e^{\langle Qe^{tV} z_i(t), Ke^{tV} z_j(t) \rangle}}{\sum_{k=1}^{n} e^{\langle Qe^{tV} z_i(t), Ke^{tV} z_k(t) \rangle}} \right) V(z_j(t) - z_i(t)) \qquad \text{for } t \in [0, +\infty). \quad (4)$$

The initial condition remains the same: $x_i(0) = z_i(0)$ for any $i \in [n]$. More importantly, the coefficients of the self-attention matrix for the rescaled tokens $z_i(t)$ are the same as those for the original tokens $x_i(t)$. Whence, the conclusion of Theorem 2.1 also applies to the dynamics (4). We see this rescaling of tokens as a mathematically justified surrogate for the layer normalization.

The appearance of the exponential factor within the self-attention kernel facilitates the analysis of (4) compared to (1), and it is in fact instrumental in the proofs of all results that follow. Each result on the rescaled tokens $z_i(t)$ then gives information on the dynamics of the original tokens $x_i(t)$ by virtue of the relation $x_i(t) = e^{tV} z_i(t)$.

We are now able to state the main result of this section on the case $V = I_d$. The following theorem shows that the tokens $z_i(t)$ evolving per dynamics (4) converge to the boundary of a convex polytope as $t \to +\infty$. We present here a simplified but weaker version of our result for convenience, and refer the reader to Theorem C.1 in the appendix for a complete statement.

**Theorem 3.1** (Convergence to points on the boundary of a convex polytope). *Suppose $V = I_d$ and $Q^\top K > 0$. Then, for any initial sequence of tokens $\{z_i(0)\}_{i \in [n]} \subset \mathbb{R}^d$, there exists a convex polytope $\mathcal{K} \subset \mathbb{R}^d$ such that for any $i \in [n]$, $z_i(t)$ converges either to $0$ or to some point on $\partial \mathcal{K}$ as $t \to +\infty$.*

The convex polytope $\mathcal{K}$ is completely determined by the initial sequence of tokens, and $Q^\top K$ (refer to Claim 1). Numerical experiments (e.g. Fig. 5) also lead us to claim that for almost all initial sequences of tokens, one should expect convergence of $z_i(t)$ ($i \in [n]$) toward some vertex of $\mathcal{K}$. (Furthermore, the number of vertices of $\mathcal{K}$ is often found to be significantly smaller than $n$.) It may however happen that for initial sequences taken in some null set (not seen when tokens are drawn at random) some tokens converge to other points of the boundary $\partial \mathcal{K}$, namely in the interior of facets. On the other hand, for generic choices of initial sequences, we do not see a way to predict $\mathcal{K}$ explicitly besides running the full dynamics.

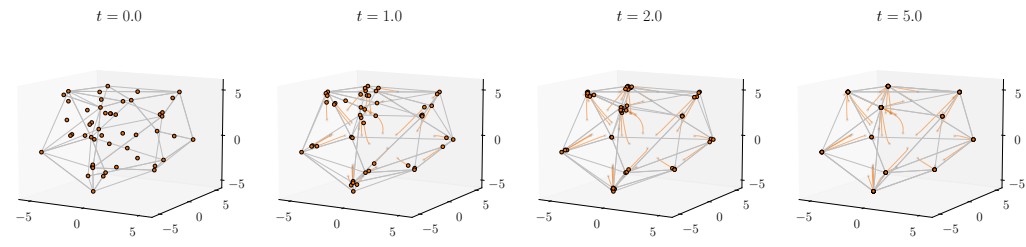

Figure 5: A toy example illustrating Theorem 3.1 with $n = 40$ tokens in $\mathbb{R}^3$. Here $Q = K = I_3$. The tokens converge to one of the vertices (*leaders*) of the limiting convex polytope.

Recall that the points $x_i(t) = e^t z_i(t)$ when $V = I_d$ follow the original dynamics (1). Akin to Theorem 2.1, this result also shows the emergence of a set of *leaders* (given by the vertices of $\mathcal{K}$)

---

[2]Note that the case $V = -I_d$ may appear equally natural. For such a choice of $V$, we show in Appendix C.2 that the dynamics converge to a single cluster located at the origin. Multiplicative constants preserving the sign, i.e., $V = \pm c I_d, c > 0$ trivially yield the same conclusions.

attracting all tokens as $t$ grows. It has been experimentally observed (first in [VSP$^+$17]) that in trained Transformers, tokens focus their attention on local leaders in a way that seems to reproduce the syntactic and semantic structure of sentences.

The proof of Theorem 3.1 is postponed to Appendix C, and amounts to a couple of effects entailed by the dynamics. First of all, the convex hull of the particles is shrinking over time (Proposition C.2). This is due to the fact that the distance of the particle nearest to any half-space (not containing the particles) increases with time. On the other hand, the convex hull ought not collapse since particles which have not concentrated near the boundary of the limiting polytope will continue to increase in magnitude until they themselves reach this boundary (Step 2 in the proof). This occurs due to the time-rescaling.

**Remark 2.** *Assuming $Q^\top K > 0$ does not seem to be essential for our conclusions; instead, it guides the direction of the proof. To emphasize the broader validity of our conclusion beyond this specific assumption, we conducted additional experiments (refer to Section F.6) which suggest that Theorem 3.1 (as well as Theorems 4.1 and 5.1 stated below) holds in more generality.*

**Remark 3** (Rate of convergence). *Although Theorem 3.1 (as well as Theorems 4.1 and 5.1 stated below) does not specify a rate of convergence toward $\partial\mathcal{K}$, we expect (and observe through numerics) that convergence happens very quickly—after few layers, most tokens are already clustered. What "few layers" means here necessarily depends on the typical modulus of the initial tokens, since the dynamics (1) is not invariant under multiplication of all initial conditions by a fixed real number.*

**Remark 4** (Discrete time). *As alluded to in Remark 1, all our results extend to the discrete-time Transformers (3). Indeed, just as in the continuous-time case, there is a natural rescaled dynamics, which is the discrete analogue of (4): if we set $R = I_d + V\Delta t$, and assume that $R$ is invertible (which is the case for sufficiently small $\Delta t$), then $z_i(k\Delta t) = R^{-k}x_i(k\Delta t) := z_i^{[k]}$ satisfies*

$$z_i^{[k+1]} = z_i^{[k]} + \Delta t \sum_{j=1}^{n} \left( \frac{e^{\langle QR^k z_i^{[k]}, KR^k z_j^{[k]} \rangle}}{\sum_{\ell=1}^{n} e^{\langle QR^k z_i^{[k]}, KR^k z_\ell^{[k]} \rangle}} \right) R^{-1}V\left( z_j^{[k]} - z_i^{[k]} \right), \qquad k \in \mathbb{N}.$$

*The proofs of Theorems 2.1, C.5, 3.1, 4.1, and 5.1 carry through with straightforward modifications.*

*Let us provide some comments on the proof of Theorem 3.1 in the discrete-time setting, for the sake of completeness. First of all, Proposition C.2 holds intuitively because for all integers $i \in [n]$ and $k \geqslant 1$,*

$$z_i^{[k+1]} = \frac{1}{1+\Delta t} \left( z_i^{[k]} + \Delta t \sum_{j=1}^{n} P_{ij}^{[k]} z_j^{[k]} \right) \in \mathrm{conv}\left( \left\{ z_j^{[k]} \right\}_{j\in[n]} \right).$$

*We then define the candidate set of limit points as in (37), and Claim 1 holds without any change in the statement or in the proof. Then, just as in Steps 2 and 3 in the proof of C.1, we can first show that if $z_i^{[k]}$ is not already near some point in the candidate limit set, it will keep moving toward the boundary of the convex polytope. Finally, we can prove that tokens cannot circulate indefinitely between different points on the boundary. The combination of these arguments would establish the convergence of each token toward some point in the set given by (37).*

## 4 Clustering toward hyperplanes

While being a natural example to consider, value matrices found empirically are much more general than $V = I_d$, which we considered in the previous section. We now turn our attention to a significantly more general setting of value matrices, which we formalize as follows.

**Definition 1.** *We call $(Q, K, V)$ a good triple if the two following conditions are satisfied:*

- *the eigenvalue of $V$ with largest modulus is real, positive, and simple; namely,*

$$\lambda_1(V) > |\lambda_2(V)| \geqslant \ldots \geqslant |\lambda_d(V)|.$$

- *$\langle Q\varphi_1, K\varphi_1 \rangle > 0$ for any $\varphi_1 \in \mathbb{R}^d$ lying on the line $\ker(V - \lambda_1(V)I_d)$.*

The second condition simply states that the quadratic form $\langle Q\cdot, K\cdot \rangle$ is positive definite along the eigenspace associated to the leading eigenvalue of $V$. Note also that if all entries of $V$ are positive,

the first condition is automatically satisfied by virtue of the Perron-Frobenius theorem. In fact, this assumption is generic. On the one hand, it is satisfied by some pre-trained value matrices for ALBERT (Figure 10). On the other hand, numerical experiments indicate that a constant fraction (about $14\%$) of matrices from the real Ginibre ensemble in dimension $d = 128$—this proportion is known to vanish as $d \to \infty$, albeit very slowly [RS14].

Our clustering result in the setting of good triples can be summarized as follows: the coordinate $\langle z_i(t), \frac{\varphi_1}{\|\varphi_1\|} \rangle$ of any token $z_i(t)$ along the eigenspace spanned by $\varphi_1$ converges, as $t \to +\infty$, toward one among possibly 3 real scalars. Consequently, all the tokens $z_i(t)$ converge toward one among at most three parallel hyperplanes; see Fig. 6 for an illustration.

**Theorem 4.1** (Convergence toward $\leqslant 3$ hyperplanes). *Assume that $(Q, K, V)$ is a good triple in the sense of Definition 1. Then, for any initial sequence of tokens $\{z_i(0)\}_{i \in [n]} \subset \mathbb{R}^d$, there exist at most three parallel hyperplanes in $\mathbb{R}^d$ such that for any $i \in [n]$, the distance of the solution $z_i(t)$ to (4) to one of these hyperplanes converges to 0 as $t \to +\infty$.*

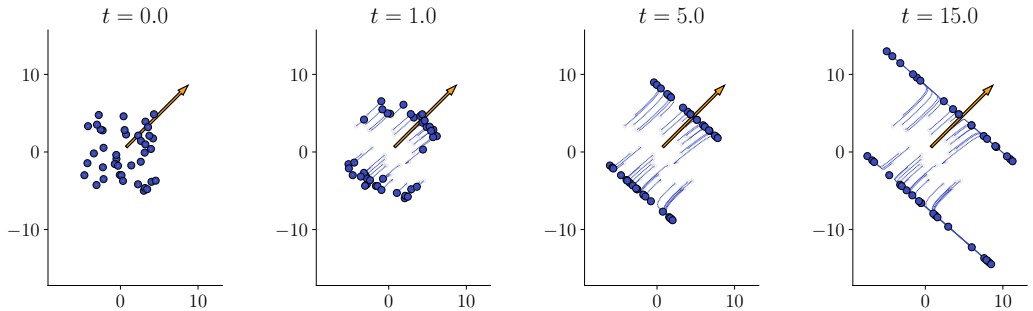

Figure 6: Illustrating Theorem 4.1 with $n = 40$ tokens in $\mathbb{R}^2$. Here $Q = K = I_2$, $V$ is a random symmetric matrix with eigenvalues $\{1.35, -0.07\}$, and $\varphi_1 = (0.76, 0.65)$. The components of the tokens in the direction of $\varphi_1$ (orange arrow) cluster over time. (See Figures 13–14 for examples in $\mathbb{R}^3$.) We also observe that tokens typically cluster toward only two hyperplanes—a third one (passing through the origin) may appear for non-generic initial sequences. The hyperplanes are perpendicular to $\varphi_1$ since $V$ is diagonalizable.

The proof may be found in Appendix D. The important role played by $\lambda_1(V)$ in the dynamics may be seen in (4): the component of $z_i(t)$ along $\varphi_1$ determines the size of $e^{tV} z_i(t)$ in the exponent appearing in (4). The tokens $z_j(t)$ attracting other tokens $z_i(t)$ are those for which this component along $\varphi_1$ is largest in modulus. This attraction process forms the clusters. These *leaders*, as in all our results, have been empirically observed to be the ones carrying the largest amount of information in the sentence (see Supplementary material in [VSP+17]).

Furthermore, Theorem 4.1 can also be interpreted in more classical machine learning terms. On the one hand, it can be seen as an instance of *K-flats clustering* [BM00, Vid11]—points in the input sequence are clustered, based on their intrinsic similarity, to at most 3 "flats" of dimension $d - 1$. On the other hand, it ensures that for a good triple $(Q, K, V)$, (4) generates a *linearly separable* representation of tokens.

### Beyond a single direction?

Numerical experiments (e.g., Fig. 7) indicate that a similar phenomenon emerges for more complex $V$. We formulate following conjecture which is a natural generalization of Theorem 4.1.

**Conjecture 4.2** (Codimension conjecture). *Let $k \geqslant 1$ be the number of eigenvalues of $V$ with positive real part. Then there exist at most three parallel Euclidean subspaces of $\mathbb{R}^d$ of codimension $k$ such that for any $i \in [n]$, the distance of $z_i(t)$ to one of these subspaces converges to 0 as $t \to +\infty$.*

## 5   A mix of hyperplanes and polytopes

We now turn our attention to an even more general version of Theorem 4.1, which does not require the leading eigenvalue of $V$ to be simple. The resulting theorem can be viewed as a combination of

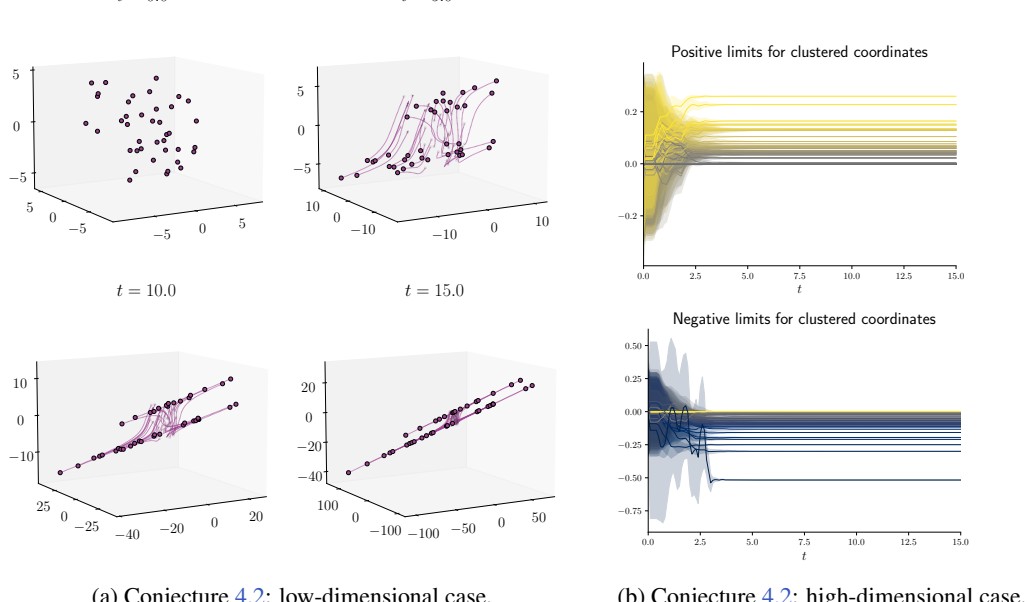

(a) Conjecture 4.2: low-dimensional case.   (b) Conjecture 4.2: high-dimensional case.

Figure 7: (a) $n = 40$, $d = 3$ and $Q = K = I_3$ with $V$ a random matrix with eigenvalues $\{1.96, -0.22, 0.25\}$. The $k = 2$ positive eigenvalues of $V$ generate attraction between the tokens and even convergence in the corresponding eigenspaces–this explains the codimension $k$ statement. The negative eigenvalue generates a repulsive effect between the tokens, and we see a divergence along two lines (note the different scales between the four figures). (b) $n = 256$, $d = 128$, with $(Q, K, V)$ fixed random matrices and $V$ symmetric. For each coordinate $j$ corresponding to a positive eigenvalue, the variance of the set $\{\varphi_j^*(z_i(t)) : i \in [n]\}$ (shaded area) tends to 0 with $t$, while the mean (solid lines) converges to one among two real scalars: one positive (top figure), one negative (bottom) figure. Coordinates corresponding to negative eigenvalues diverge (Fig. 15).

Theorem 4.1 and Theorem 3.1. Specifically, we assume that $V$ behaves as the identity when acting on the eigenspace of the leading eigenvalue. This property is automatically satisfied if $V$ is normal—so that its eigenvectors form an orthonormal basis—so we call such a $V$ *paranormal*.

**Definition 2.** *We call $(Q, K, V)$ a* good triple with multiplicity *if the following conditions hold:*

*(i) $Q^\top K$ is positive definite: $Q^\top K > 0$;*

*(ii) $V$ is paranormal: there exist two linear subspaces $\mathscr{F}, \mathscr{G} \subset \mathbb{R}^d$ which are invariant under $V$, and such that $\mathscr{F} \oplus \mathscr{G} = \mathbb{R}^d$, $V_{|\mathscr{F}} = \lambda\mathrm{Id}$ for $\lambda > 0$, and $\rho(V_{|\mathscr{G}}) < \lambda$, where $\rho(\cdot)$ denotes the spectral radius (the maximal modulus of eigenvalues).*

An example of such a $V$ is used for Fig. 8. We may now state our main result in the setting of good triples with multiplicity. The proof may be found in Appendix E.

**Theorem 5.1** (Clustering for $\lambda_1$ with multiplicity). *Suppose that $(Q, K, V)$ is a good triple with multiplicity in the sense of Definition 2. Then, for any initial sequence $\{z_i(0)\}_{i\in[n]} \subset \mathbb{R}^d$, there exists a bounded convex polytope $\mathcal{K} \subset \mathscr{F}$ such that setting $\mathscr{H} := (\partial\mathcal{K} \cup \{0\}) \times \mathscr{G}$, for any $i \in [n]$, we have $\mathrm{dist}(z_i(t), \mathscr{H}) \to 0$ as $t \to +\infty$.*

# 6   Outlook

Several important directions regarding the mathematical theory of Transformers remain unexplored. An important extension of our work would amount to studying *multi-headed* Transformers—borrowing the notation from Remark 4, they amount to:

$$x_i^{[k+1]} = x_i^{[k]} + \Delta t \sum_{h=1}^{H} \sum_{j=1}^{n} \left( \frac{e^{\langle Q_h x_i^{[k]}, K_h x_j^{[k]} \rangle}}{\sum_{\ell=1}^{n} e^{\langle Q_h x_i^{[k]}, K_h x_\ell(k) \rangle}} \right) V_h x_j^{[k]}, \qquad k \in \mathbb{N}.$$

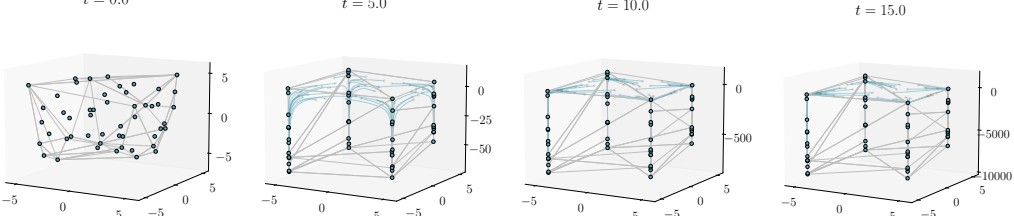

Figure 8: Illustrating Theorem 5.1 with $n = 40$ tokens in $\mathbb{R}^3$. As before, $Q = K = I_d$, and we take $V = \mathrm{diag}(1, 1, -\frac{1}{2})$. A convex polytope $\mathcal{K}$ emerges before time $5$, toward which two coordinates of the tokens cluster, and persists throughout the evolution, while the tokens diverge along the coordinate corresponding to the eigenvalue $-\frac{1}{2}$ (note the different scales between the four figures).

For each $h \in [H]$ (corresponding to a different *head*), the weight matrices $Q_h, K_h, V_h$ are constant. Proofs regarding clustering or convergence of the self-attention matrix for such dynamics is an open problem. Preliminary numerical investigations seem to indicate that interesting clustering phenomena also occur in this context. A characterization or properties of optimal weights by invoking the optimal control correspondence in the spirit of [Wei17] is also an interesting avenue for future research.

## Acknowledgments and Disclosure of Funding

We thank Pierre Ablin, Léonard Boussioux, Enric Boix-Adsera, Gabriel Peyré, Yair Shenfeld and Emmanuel Trélat for helpful discussions. C.L. was supported by the Simons Foundation Grant 601948, DJ. P.R. is supported by NSF grants IIS-1838071, DMS-2022448, and CCF-2106377. Y.P. is supported in part by the MIT-IBM Watson AI Lab.

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
