## Supplementary material for the paper "The emergence of clusters in self-attention dynamics"

This appendix is organized as follows:

- **Appendix A**: Well-posedness results.
- **Appendix B**: Convergence of the self-attention matrix to a low-rank matrix (proof of Theorem 2.1).
- **Appendix C**: Clustering towards vertices of convex polytopes (proofs of Theorems C.5 and 3.1).
- **Appendix D**: Clustering towards hyperplanes (proof of Theorem 4.1).
- **Appendix E**: A mix of hyperplanes and polytopes (proof of Theorem 5.1).
- **Appendix F**: Numerical experiments.

## A    Well-posedness

We collect several facts regarding the global-in-time existence and uniqueness of solutions to all systems under consideration. Throughout the remainder of the paper, we use the terminology "tokens" and "particles" interchangeably.

To prove these results, we leverage the underlying continuity equation (see (5)). For the sake of future use, we prove a more general well-posedness result for the continuity equation than what is needed in this paper.

### A.1    Notation.

We denote by $\mathcal{P}_c(\mathbb{R}^d)$ the set of compactly supported probability measures on $\mathbb{R}^d$, and by $\mathcal{P}_2(\mathbb{R}^d)$ the set of probability measures $\mu$ on $\mathbb{R}^d$ having finite second moment: $\int_{\mathbb{R}^d} \|x\|^2 \, \mathrm{d}\mu(x) < +\infty$. Let $C^0(\mathbb{R}; \mathcal{P}_c(\mathbb{R}^d))$ denote the Banach space of continuous curves $\mathbb{R} \ni t \mapsto \mu(t) \in \mathcal{P}_c(\mathbb{R}^d)$. Here $\mathcal{P}_c(\mathbb{R}^d)$ is endowed with the weak topology, which coincides with the topology induced by the Wasserstein distance $W_p$ for any $p \in [1, +\infty)$.

As seen below, for compactness purposes regarding solutions to the continuity equation, we consider an additional property on the support of such curves, summarized by the following definition.

**Definition 3** (Equi-compactly supported curves)**.** *The set $C_{\mathrm{co}}^0(\mathbb{R}; \mathcal{P}_c(\mathbb{R}^d))$ consists of all elements $\mu \in C^0(\mathbb{R}; \mathcal{P}_c(\mathbb{R}^d))$ such that for any $t_0, t_1 \in \mathbb{R}$, there exists a compact subset $\mathcal{K} \subset \mathbb{R}^d$ such that $\mathrm{supp}(\mu(t)) \subset \mathcal{K}$ for any $t \in [t_0, t_1]$.*

We emphasise that there exist elements in $C^0(\mathbb{R}; \mathcal{P}_c(\mathbb{R}^d))$ which do not satisfy this property with regard to their support—e.g., $\mu(t) = (1 - e^{-\frac{1}{t^2}})\delta_0 + e^{-\frac{1}{t^2}}\delta_{\frac{1}{t}}$.

### A.2    Well-posedness of the ODEs

For any initial datum, i.e. a sequence of $n$ points in $\mathbb{R}^d$, the dynamics (1) is well-posed, in the sense that it admits a unique solution defined for all times.

**Proposition A.1.** *For any initial datum $\mathbf{X}_0 = (x_1^0, \ldots, x_n^0) \in (\mathbb{R}^d)^n$, there exists a unique Lipschitz continuous function $\mathbb{R} \ni t \mapsto \mathbf{X}(t) = (x_1(t), \ldots, x_n(t))$ such that $x_i(\cdot)$ solves (1) and satisfies $x_i(0) = x_i^0$ for any $i \in [n]$.*

We postpone the proof which is seen as a corollary of the well-posedness for the corresponding continuity equation. It follows that the equation (4) is also well-posed:

**Proposition A.2.** *For any initial datum $\mathbf{Z}_0 = (z_1^0, \ldots, z_n^0) \in (\mathbb{R}^d)^n$, there exists a unique Lipschitz continuous function $\mathbb{R} \ni t \mapsto \mathbf{Z}(t) = (z_1(t), \ldots, z_n(t))$ such that $z_i(\cdot)$ solves (4) and satisfies $z_i(0) = z_i^0$ for any $i \in [n]$.*

*Proof of Proposition A.2.* Since the equations (1) and (4) are related by the change of variables $x_i(t) = e^{tV} z_i(t)$, Proposition A.2 is an immediate consequence of Proposition A.1. $\square$

## A.3 The continuity equation

To prove Proposition A.1, we show a more general result concerning global existence and uniqueness of solutions to the corresponding continuity equation[3]

$$\begin{cases} \partial_t \mu + \operatorname{div}(\mathcal{X}[\mu]\mu) = 0 & \text{in } (0, +\infty) \times \mathbb{R}^d \\ \mu_{|t=0} = \mu_0 & \text{in } \mathbb{R}^d, \end{cases} \tag{5}$$

when $\mathcal{X}[\mu]$ is the *attention kernel*

$$\mathcal{X}[\mu](x) := \frac{\displaystyle\int_{\mathbb{R}^d} e^{\langle Qx, Ky\rangle} Vy \, \mathrm{d}\mu(y)}{\displaystyle\int_{\mathbb{R}^d} e^{\langle Qx, Ky\rangle} \, \mathrm{d}\mu(y)}. \tag{6}$$

We will make use of the following notion of solution.

**Definition 4.** *Fix $\mu_0 \in \mathcal{P}_c(\mathbb{R}^d)$. We say that $t \mapsto \mu(t) =: \mu_t$ is a solution to the Cauchy problem (5) if $\mu \in C^0_{\mathrm{co}}(\mathbb{R}, \mathcal{P}_c(\mathbb{R}^d))$, the function*

$$\mathbb{R} \ni t \mapsto \int_{\mathbb{R}^d} g(x) \, \mathrm{d}\mu_t(x)$$

*is absolutely continuous for every $g \in C^\infty_c(\mathbb{R}^d)$, and*

$$\int_{\mathbb{R}^d} g(x) \, \mathrm{d}\mu_t(x) = \int_{\mathbb{R}^d} g(x) \, \mathrm{d}\mu_0(x) + \int_0^t \int_{\mathbb{R}^d} \langle \nabla g(x), \mathcal{X}[\mu_t](x)\rangle \, \mathrm{d}\mu_s(x) \, \mathrm{d}s$$

*holds for almost every $t \in \mathbb{R}$.*

We will make use of the following lemma regarding (6).

**Lemma A.3.** *For any $R > 0$ there exists a constant $C_1(R) > 0$ such that for any $\mu, \nu \in \mathcal{P}_c(\mathbb{R}^d)$ with support in $B(0, R)$,*

$$\|\mathcal{X}[\mu]\|_{L^\infty(\mathbb{R}^d; \mathbb{R}^d)} \leqslant \|V\|_{\mathrm{op}} R, \tag{7}$$

$$\|\nabla_x \mathcal{X}[\mu]\|_{L^\infty(\mathbb{R}^d; \mathbb{R}^{d\times d})} \leqslant 2\|Q^\top K\|_{\mathrm{op}}\|V\|_{\mathrm{op}} R^2 \tag{8}$$

$$\|\mathcal{X}[\mu](\cdot) - \mathcal{X}[\nu](\cdot)\|_{L^\infty(B(0,R); \mathbb{R}^d)} \leqslant C_1(R) W_2(\mu, \nu). \tag{9}$$

*Proof.* We henceforth set $\mathsf{G}(x, y) := e^{\langle Qx, Ky\rangle}$. To show (7), since $\mathsf{G} > 0$ we see that for any $x \in \mathbb{R}^d$,

$$\|\mathcal{X}[\mu](x)\| \leqslant \|V\|_{\mathrm{op}} \frac{\displaystyle\int_{B(0,R)} \mathsf{G}(x,y)\|y\| \, \mathrm{d}\mu(y)}{\displaystyle\int_{B(0,R)} \mathsf{G}(x,y) \, \mathrm{d}\mu(y)} \leqslant \|V\|_{\mathrm{op}} R.$$

We now show (8). Note that $\nabla_x \mathsf{G}(x, y) = Q^\top K y \mathsf{G}(x, y)$, thus, arguing as above, we find

$$\|\nabla_x \mathcal{X}[\mu](x)\| \leqslant \frac{\displaystyle\int_{B(0,R)} \|\nabla_x \mathsf{G}(x,y)\|\|Vy\| \, \mathrm{d}\mu(y)}{\displaystyle\int_{B(0,R)} \mathsf{G}(x,y) \, \mathrm{d}\mu(y)}$$

$$+ \|V\|_{\mathrm{op}} \frac{\displaystyle\int_{B(0,R)} \mathsf{G}(x,y)\|y\| \, \mathrm{d}\mu(y)}{\displaystyle\int_{B(0,R)} \mathsf{G}(x,y) \, \mathrm{d}\mu(y)} \frac{\displaystyle\int_{B(0,R)} \|\nabla_x \mathsf{G}(x,y)\| \, \mathrm{d}\mu(y)}{\displaystyle\int_{B(0,R)} \mathsf{G}(x,y) \, \mathrm{d}\mu(y)}$$

$$\leqslant 2\|Q^\top K\|_{\mathrm{op}}\|V\|_{\mathrm{op}} R^2.$$

---

[3]which can be seen as a mean-field limit, and is sometimes also referred to as a *Vlasov equation*.

We finally prove (9). Using the fact that

$$\int_{\mathbb{R}^d} \mathsf{G}(x,y)\,\mathrm{d}\mu(y) \geqslant \left(\inf_{(x,y)\in B(0,R)^2} \mathsf{G}(x,y)\right)\mu(B(0,R)),$$

–with an analogous bound for $\nu$–, we see that it suffices to bound

$$\left|\int_{\mathbb{R}^d} \mathsf{G}(x,y)Vy\,\mathrm{d}\mu(y)\int_{\mathbb{R}^d} \mathsf{G}(x,y)\,\mathrm{d}\nu(y) - \int_{\mathbb{R}^d} \mathsf{G}(x,y)Vy\,\mathrm{d}\nu(y)\int_{\mathbb{R}^d} \mathsf{G}(x,y)\,\mathrm{d}\mu(y)\right|$$

from above. We rewrite this difference by making $\mu - \nu$ appear artificially, and we then use the triangle inequality along with the fact that both $\int_{\mathbb{R}^d} \mathsf{G}(x,y)Vy\,\mathrm{d}\mu(y)$ and $\int_{\mathbb{R}^d} \mathsf{G}(x,y)\,\mathrm{d}\mu(y)$ are bounded from above (by $e^{\|Q^\top K\|_{\mathrm{op}}R^2}\max(1,\|V\|_{\mathrm{op}}R)$). We thus end up with the task of bounding from above the absolute values of

$$\int_{\mathbb{R}^d} \mathsf{G}(x,y)(\,\mathrm{d}\nu - \mathrm{d}\mu)(y) \qquad \text{and} \qquad \int_{\mathbb{R}^d} \mathsf{G}(x,y)Vy(\,\mathrm{d}\nu - \mathrm{d}\mu)(y). \tag{10}$$

For the first integral, from the Kantorovich-Rubinstein duality we deduce

$$\left|\int_{\mathbb{R}^d} \mathsf{G}(x,y)(\,\mathrm{d}\nu - \mathrm{d}\mu)(y)\right| \leqslant \|\mathsf{G}(x,\cdot)\|_{C^{0,1}(B(0,R))}W_1(\mu,\nu). \tag{11}$$

We now recall the following inequality relating Wasserstein distances of different orders: for any $p \geqslant 1$ and any bounded set $B$, for all Radon measures $\mu, \nu$ supported in $B$,

$$W_1(\mu,\nu) \leqslant W_p(\mu,\nu) \leqslant \operatorname{diam}(B)^{1-\frac{1}{p}}W_1(\mu,\nu)^{1/p}. \tag{12}$$

Using (12) and the fact that the Lipschitz constant $\|\mathsf{G}(x,\cdot)\|_{C^{0,1}(B(0,R))}$ is uniformly bounded for $\|x\| \leqslant R$ by some $C_R > 0$ in (11), we end up with

$$\left|\int_{\mathbb{R}^d} \mathsf{G}(x,y)(\,\mathrm{d}\nu - \mathrm{d}\mu)(y)\right| \leqslant C_R W_2(\mu,\nu).$$

The same chain of inequalities applies to the second integral in (10) (with the additional multiplier $\|V\|_{\mathrm{op}}R$), which finally leads us to (9). $\qquad\square$

The following existence and uniqueness result is adapted from [PRT15, Theorem 2.3]. In fact, the result holds true for any vector field $\mathcal{X}[\mu]$ on $\mathbb{R}^d$ satisfying conditions analog to those entailed by Lemma A.3.

**Proposition A.4.** *For any initial condition $\mu_0 \in \mathcal{P}_c(\mathbb{R}^d)$, the Cauchy problem* (5) *admits a unique solution $\mu \in C^0_{\mathrm{co}}(\mathbb{R}; \mathcal{P}_c(\mathbb{R}^d))$ in the sense of Definition 4.*

*Furthermore, we have the following stability estimate for solutions: for any $R > 0$ and $T > 0$, there exists a constant $C(T,R) > 0$ such that for any $\mu_0, \nu_0 \in \mathcal{P}_c(\mathbb{R}^d)$ with support in $B(0,R)$,*

$$W_2(\mu(t),\nu(t)) \leqslant e^{C(T,R)t}W_2(\mu_0,\nu_0) \tag{13}$$

*for any $t \in [0,T]$, where $\mu(t)$ and $\nu(t)$ solve* (5) *with initial conditions $\mu_0$ and $\nu_0$ respectively.*

Results of this nature can be found in the literature—see for instance [PRT15]. They are however not sufficient for our purposes. We wrote Proposition A.4 in the $W_2$ setting instead of the usual $W_1$ setting (used for instance for the classical *Dobrushin estimate* [Dob79, Gol13]) because it allows to extend the results of [WHL19] without difficulty from classical ResNets to self-attention dynamics. We recall that the goal of [WHL19] is to import classical (mean-field) optimal control tools such as the Pontryagin maximum principle and the analysis of Hamilton-Jacobi-Bellman equations to deep learning, and relies heavily on $W_2$ estimates (e.g., in [WHL19, Section 4]).

*Proof of Proposition A.4.* To ease reading, we split the proof in three parts.

**Part 1: Existence.** Fix an arbitrary $T > 0$. For $k \geqslant 1$, set

$$\tau_k := \frac{T}{2^k}.$$

We define a sequence of curves $\mu^k : [0,T] \to \mathcal{P}_c(\mathbb{R}^d)$ by the following scheme[4]:

---

[4]In other words we "freeze" the vector field $\mathcal{X}$ on each interval of the form $[\ell\tau_k, (\ell+1)\tau_k)$, and during this time interval, we follow the flow generated by this vector field starting from $\mu^k(\ell\tau_k)$.

(i) $\mu^k(0) := \mu_0$;

(ii) $\mu^k(\ell\tau_k + t) := \left(\Phi^t_{\mathcal{X}[\mu^k(\ell\tau_k)]}\right)_{\#} \mu^k(\ell\tau_k)$ for $\ell \in \{0, \ldots, 2^k - 1\}$ and $t \in (0, \tau_k]$,

where for any $x \in \mathbb{R}^d$, $\Phi^t_{\mathcal{X}[\mu^k(\ell\tau_k)]}(x)$ is the unique solution to the Cauchy problem

$$\begin{cases} \dot{y}(t) = \mathcal{X}[\mu^k(\ell\tau_k)](y(t)) & \text{on } [0, \tau_k] \\ y(0) = x. \end{cases}$$

(The above problem indeed has a unique solution for any $x \in \mathbb{R}^d$ by virtue of the Cauchy-Lipschitz theorem, using (8).) By construction, $\mu^k \in C^0([0, T]; \mathcal{P}_c(\mathbb{R}^d))$ for any $k \geqslant 1$.

We begin by showing that there exists a radius $R = R(T) > 0$ independent of $k$ such that $\operatorname{supp}(\mu^k(t)) \subset B(0, R)$ for any $k \geqslant 1$ and $t \in [0, T]$. To this end, for any $t \in [0, T]$ and $k \geqslant 1$, let $R_k(t) > 0$ denote the smallest positive radius[5] such that $\operatorname{supp}(\mu^k(t)) \subset B(0, R_k(t))$. We will first look to show that

$$\operatorname{supp}(\mu^k(\ell\tau_k + t)) \subset B(0, R_k(\ell\tau_k) + t\|V\|_{\mathrm{op}}R_k(\ell\tau_k)). \tag{14}$$

Let $x \in \operatorname{supp}(\mu^k(\ell\tau_k + t))$, thus $\mu^k(\ell\tau_k + t)(B(x, \varepsilon)) > 0$ for any $\varepsilon > 0$. By the change of variables formula, we find that

$$\int_{(\Phi^t_{\mathcal{X}[\mu^k(\ell\tau_k)]})^{-1}(B(x,\varepsilon))} \mathrm{d}\mu^k(\ell\tau_k)(z) > 0.$$

Consequently $(\Phi^t_{\mathcal{X}[\mu^k(\ell\tau_k)]})^{-1}(B(x, \varepsilon)) \cap \operatorname{supp}(\mu^k(\ell\tau_k)) \neq \varnothing$, and let $z$ be an element lying in this set. From the Duhamel formula, we gather that

$$\Phi^t_{\mathcal{X}[\mu^k(\ell\tau_k)]}(z) =: y(t) = z + \int_0^t \mathcal{X}[\mu^k(\ell\tau_k)](y(s))\,\mathrm{d}s.$$

Since $z \in (\Phi^t_{\mathcal{X}[\mu^k(\ell\tau_k)]})^{-1}(B(x, \varepsilon))$, we find that

$$\left\| z + \int_0^t \mathcal{X}[\mu^k(\ell\tau_k)](y(s))\,\mathrm{d}s - x \right\| \leqslant \varepsilon.$$

Using the triangle inequality, (7), and since $z \in \operatorname{supp}(\mu^k(\ell\tau_k))$ implies $z \in B(0, R_k(\ell\tau_k))$, we deduce that

$$\|x\| \leqslant \varepsilon + t\|V\|_{\mathrm{op}}R_k(\ell\tau_k) + R_k(\ell\tau_k).$$

Since $\varepsilon > 0$ is arbitrary, this inequality yields (14). We now use (14) to prove the original claim. Using the definition of the radius $R_k(t)$, we evaluate (14) at $t = \tau_k$ and find

$$R_k((\ell + 1)\tau_k) \leqslant (1 + \|V\|_{\mathrm{op}}\tau_k)R_k(\ell\tau_k).$$

By induction, we deduce that

$$R_k(\ell\tau_k) \leqslant (1 + \|V\|_{\mathrm{op}}\tau_k)^\ell R_k(0),$$

whence

$$R_k(\ell\tau_k) \leqslant \left(1 + \|V\|_{\mathrm{op}}\frac{T}{2^k}\right)^{2^k} R_k(0) < e^{\|V\|_{\mathrm{op}}T}R_0,$$

where $R_0 > 0$ denotes the smallest positive radius such that $\operatorname{supp}(\mu_0) \subset B(0, R_0)$. Since the above bound is independent of $k$, the claim follows, yielding the desired radius $R = R(T) > 0$ bounding the support of every element in the sequence. In turn, we also deduce that $\mu^k \in C^0_{\mathrm{co}}(\mathbb{R}; \mathcal{P}_c(\mathbb{R}^d))$ for any $k \geqslant 1$.

Using the above fact, along with (7) and the definition of $\mu^k(\ell\tau_k + t)$, we find that

$$W_2\left(\mu^k(\ell\tau_k + t), \mu^k(\ell\tau_k)\right) \leqslant \|V\|_{\mathrm{op}}Rt$$

---

[5]This radius always exists, since $\mu^k(t)$ is compactly supported.

for any $\ell \in \{0, \dots, 2^k - 1\}$, $t \in (0, \tau_k]$ and $k \geqslant 1$. Gluing these inequalities (for different $\ell$ and $t$) with the triangle inequality yields

$$W_2\left(\mu^k(t), \mu^k(s)\right) \leqslant \|V\|_{\mathrm{op}} R |t - s|$$

for any $t \in [0, T]$. Since $\mu^k(0) = \mu_0$ for any $k \geqslant 1$, and since $\mathcal{P}_2(\mathbb{R}^d)$ is the completion of $\mathcal{P}_c$ for the Wasserstein distance $W_2$, the Arzelà-Ascoli theorem implies the existence of a subsequence uniformly converging to some $\mu^* \in C^0([0, T]; \mathcal{P}_2(\mathbb{R}^d))$. Since for any $t \in [0, T]$ the curves $\mu^k(t)$ have their support enclosed in $B(0, R)$ for any $k \geqslant 1$, we even deduce that $\mu^* \in C^0_{\mathrm{co}}(\mathbb{R}, \mathcal{P}_c(\mathbb{R}^d))$. Note moreover that $\mu^*(0) = \mu_0$ and that

$$W_2(\mu^*(t), \mu^*(s)) \leqslant \|V\|_{\mathrm{op}} R |t - s|$$

for any $t, s \in [0, T]$.

The fact that $\mu^*$ is a solution of (5) follows exactly from the same computations as in [PRT15, p. 4711-4712], starting from (A.2) therein. We do not reproduce here this argument since the computations are the same word for word. The fact that for any $T > 0$ we have $\sup_{t \in [0,T]} W_1(\mu^*(t), \mu^k(t)) \to 0$ as $k \to +\infty$, which is instrumental in [PRT15, p. 4711-4712], follows in our case from the left-hand-side of (12).

**Part 2: Uniqueness.** Regarding uniqueness, we proceed as follows. We first recall the following estimate from [PR16, Proposition 4]. Let $p \geqslant 1$, let $t \geqslant 0$, let $v, w \in C^{0,1} \cap L^\infty([0, t] \times \mathbb{R}^d; \mathbb{R}^d)$ (both with Lipschitz constant $L > 0$, say), and let $\mu, \nu \in \mathcal{P}_c(\mathbb{R}^d)$. Then

$$W_p\left((\Phi^t_v)_\# \mu, (\Phi^t_w)_\# \nu\right) \leqslant e^{\frac{p+1}{p} Lt} W_p(\mu, \nu) + \frac{e^{\frac{Lt}{p}}(e^{Lt} - 1)}{L} \|v - w\|_{L^\infty([0,t] \times \mathbb{R}^d; \mathbb{R}^d)}. \qquad (15)$$

Now assume that there are two solutions $\mu$ and $\nu$ of (5), with a spatial support that is locally bounded in time, and having the same initial condition. Define $v(t, x) := \mathcal{X}[\mu(t)](x)$ and $w(t, x) := \mathcal{X}[\nu(t)](x)$. Also set

$$t_0 := \inf\{t \geqslant 0 \colon W_2(\mu(t), \nu(t)) \neq 0\},$$

and assume that $t_0 \neq +\infty$. Fix $T > t_0$ and take $R > 0$ such that $\mu_t$ and $\nu_t$ are supported in $B(0, R)$ for any $t \in [0, T]$. Using (15) with $p = 2$, and setting $C_2(R) := 2\|Q^\top K\|_{\mathrm{op}} \|V\|_{\mathrm{op}} R^2$ in (8), we find

$$W_2(\mu(t_0 + s), \nu(t_0 + s)) \leqslant e^{2C_2(R)s} W_2(\mu(t_0), \nu(t_0))$$
$$+ e^{C_2(R)s} \frac{e^{C_2(R)s} - 1}{C_2(R)} \sup_{\tau \in [t_0, t_0 + s]} \|v(\tau, \cdot) - w(\tau, \cdot)\|_{L^\infty(\mathbb{R}^d)}.$$

Choose $s > 0$ sufficiently small so that $e^{C_2(R)s} - 1 \leqslant 2C_2(R)s$. Then, by virtue of (9) and the fact that $W_2(\mu(t_0), \nu(t_0)) = 0$, we deduce

$$W_2(\mu(t_0 + s), \nu(t_0 + s)) \leqslant 2s e^{C_2(R)s} \sup_{\tau \in [t_0, t_0 + s]} W_2(\mu(\tau), \nu(\tau)). \qquad (16)$$

We choose $s' > 0$ satisfying both $e^{C_2(R)s'} - 1 \leqslant 2C_2(R)s'$ and $2s' e^{C_2(R)s'} < 1$. Applying (16) to every $s \in [0, s']$ we obtain

$$\sup_{s \in [0,s']} W_2(\mu(t_0 + s), \nu(t_0 + s)) \leqslant 2s' e^{C_2(R)s'} \sup_{\tau \in [t_0, t_0 + s']} W_2(\mu(\tau), \nu(\tau))$$
$$< \sup_{s \in [0,s']} W_2(\mu(t_0 + s), \nu(t_0 + s)),$$

which is a contradiction. Therefore $\mu(t) \equiv \nu(t)$ for any $t \geqslant 0$, which proves uniqueness, as desired.

**Part 3: Stability.** We do not detail the proof of estimate (13), which is very similar to the proof of (2.3) in Theorem 2.3 of [PRT15]: it follows from (15) with $p = 2$, and the argument after (A.7) in [PRT15], with $W_2$ instead of $W_1$. See also [PR13, Theorem 3]. $\qquad \square$

We conclude this section with the proof of Proposition A.1, which follows as a corollary of the above derivations.

*Proof of Proposition A.1.* We first show existence. We apply Proposition A.4 with $\mu_0 := \frac{1}{n} \sum_{j=1}^n \delta_{x_i^0}$, which in turn yields a solution $\mu(t)$ to (5). Following the proof of Proposition A.4, we also know that this solution satisfies $\mu(t) = (\Phi_{\mathcal{X}[\mu(t)]}^t)_{\#}\mu_0$ for any $t \in \mathbb{R}$, and the vector field $\mathcal{X}[\mu(t)]$ satisfies the assumptions of the Cauchy-Lipschitz theorem. In particular, $\mu(t)$ is of the form $\mu(t) = \frac{1}{n} \sum_{j=1}^n \delta_{x_i(t)}$ for some Lipschitz curves $\mathbb{R} \ni t \mapsto x_i(t)$, for $i \in [n]$. Then $t \mapsto \mu(t) = \frac{1}{n} \sum_{j=1}^n \delta_{x_i(t)}$ is a solution to the Cauchy problem (5)-(6) in the sense of Definition 4.

Secondly, we show uniqueness. Suppose that $\mathbf{X}(t) = (x_1(t), \ldots, x_n(t))$ and $\mathbf{X}^*(t)$ are two Lipschitz solutions to (1), with the same initial conditions. Then for a.e. $t \geqslant 0$, using the equation (1) and the fact that the attention matrix coefficients $P_{ij}(t)$ defined in (2) belong to $[0,1]$, we obtain

$$\frac{1}{2} \frac{\mathrm{d}}{\mathrm{d}t} \max_{i \in [n]} \|x_i(t)\|^2 \leqslant \|V\|_{\mathrm{op}} \max_{i \in [n]} \|x_i(t)\|^2$$

(and analogously for $x_i^*(t)$). Using Grönwall's inequality, we deduce the existence of two constants $c_1, c_2 > 0$ such that for any $t > 0$ and for any $i \in [n]$, $\|x_i(t)\|$ and $\|x_i^*(t)\|$ are bounded from above by $c_1 e^{c_2 t}$. It then follows that the empirical measures $\mu(\cdot) = \frac{1}{n} \sum_{j=1}^n \delta_{x_i(\cdot)}$ and $\mu^*(\cdot) = \frac{1}{n} \sum_{j=1}^n \delta_{x_i^*(\cdot)}$ belong to $C_{\mathrm{co}}^0(\mathbb{R}, \mathcal{P}_c(\mathbb{R}^d))$. Moreover, they satisfy $\mu(t) = (\Phi_{\mathcal{X}[\mu(t)]}^t)_{\#}\mu_0$ and $\mu^*(t) = (\Phi_{\mathcal{X}[\mu^*(t)]}^t)_{\#}\mu_0$ and are thus solutions to (5). Using the uniqueness result of Proposition A.4, we obtain that $\mu = \mu^*$ which concludes the proof. $\qquad \square$

# B   Convergence of the self-attention matrix: proof of Theorem 2.1

Throughout this section we focus on the following dynamics:

$$\dot{x}_i(t) = \sum_{j=1}^n \left( \frac{e^{\langle x_i(t), x_j(t) \rangle}}{\sum_{k=1}^n e^{\langle x_i(t), x_k(t) \rangle}} \right) x_j(t). \tag{17}$$

Note that for $d = 1$, the dot products in (17) are just multiplications of scalars.

We begin with the following observation, which holds for any $d \geqslant 1$.

**Lemma B.1.** *For any $x_1, \ldots, x_n \in \mathbb{R}^d$, the function $f : \mathbb{R}^d \to \mathbb{R}$ defined by*

$$f : x \mapsto \log \left( \sum_{j=1}^n e^{\langle x, x_j \rangle} \right) \tag{18}$$

*is convex.*

*Proof.* Using the elementary inequality $(a + b) \geqslant 2(ab)^{\frac{1}{2}}$ for any $a, b \geqslant 0$, we have

$$\exp(f(x) + f(y)) = \left( \sum_{j=1}^n \exp(\langle x, x_j \rangle) \right) \left( \sum_{j=1}^n \exp(\langle y, x_j \rangle) \right)$$

$$= \frac{1}{2} \sum_{j=1}^n \sum_{k=1}^n \left[ \exp\left( \langle x, x_j \rangle + \langle y, x_k \rangle \right) + \exp(\langle x, x_k \rangle + \langle y, x_j \rangle) \right] \tag{19}$$

$$\geqslant \sum_{j=1}^n \sum_{k=1}^n \exp \left( \left\langle \frac{x + y}{2}, x_j + x_k \right\rangle \right) \tag{20}$$

$$= \exp \left( 2f \left( \frac{x + y}{2} \right) \right).$$

Taking the $\log$ on both sides yields the statement. $\qquad \square$

The following lemma also holds for any $d \geqslant 1$.

**Lemma B.2.** *Let $\mathbb{R} \ni t \mapsto \{x_i(t)\}_{i \in [n]}$ be a solution to (17). Then for any $i, j \in [n]$, the map $\mathbb{R} \ni t \mapsto \|x_i(t) - x_j(t)\|$ is non-decreasing.*

*Proof.* The dynamics (17) can be equivalently written as

$$\dot{x}_i(t) = \nabla f(x_i(t))$$

where $f$ is as in (18). By convexity of $f$ (Lemma B.1),

$$\frac{1}{2}\frac{\mathrm{d}}{\mathrm{d}t}\|x_i(t) - x_j(t)\|^2 = \langle \dot{x}_i(t) - \dot{x}_j(t), x_i(t) - x_j(t) \rangle$$
$$= \langle \nabla f(x_i(t)) - \nabla f(x_j(t)), x_i(t) - x_j(t) \rangle \geqslant 0,$$

as desired. $\qquad\square$

We now present the proof of Theorem 2.1, which assumes $d = 1$. We recall that in the statement, $V$ is a positive scalar, but by reparametrizing time we may assume that $V = 1$, so the $1d$ dynamics under consideration is really given by (17). Also, to ease notations we focus on $QK = 1$, but the proof adapts straightforwardly to the setting $QK > 0$ assumed in the statement of Theorem 2.1.

As seen in Section B.1, it is not difficult to prove the convergence of the coefficients $P_{ij}(t)$ of the attention matrix for indices $i \in [n]$ for which $x_i(t)$ becomes unbounded as $t \to +\infty$. This is the case for at least $n - 1$ of the particles $x_i(t)$ (Lemma B.6). But should one particle $x_i(t)$ remain bounded, proving the convergence of $P_{ij}(t)$ for $j \in [n]$ is slightly tedious (Section B.2).

Since $d = 1$, up to relabeling, we can order the initial collection of particles (which, we recall, are assumed distinct):

$$x_1(0) < \ldots < x_n(0).$$

We set

$$c := \min_{i \in [n-1]} |x_{i+1}(0) - x_i(0)|. \tag{21}$$

According to Lemma B.2, we have $|x_i(t) - x_j(t)| \geqslant c$ for any $i \neq j$ and any $t \geqslant 0$. In particular, particles never "collide".

## B.1 Results about unbounded particles

In this section we gather several results concerning the indices $i$ corresponding to particles $x_i(t)$ which are not uniformly bounded in time. In particular, in Lemma B.4 we show that for such indices $i$, $P_{ij}(t)$ converges toward 0 or 1 for any $j \in [n]$.

**Lemma B.3.** *Let $A > 0$ denote the unique positive real number satisfying $A^2 = n^2 \exp(-A^2)$. If $x_n(t_0) > A$ for some time $t_0 \geqslant 0$, then there exists $c_1 > 0$ such that $x_n(t) \geqslant c_1 e^t$ for any sufficiently large $t > 0$. Similarly, if $x_1(t_0) < -A$ for some $t_0 \geqslant 0$, then $x_1(t) \leqslant -c_1 e^t$ for any sufficiently large $t > 0$.*

*Proof.* The two cases are symmetric since the evolution (17) commutes with the involution of $(\mathbb{R}^d)^n$ given by $(x_1, \ldots, x_n) \mapsto (-x_1, \ldots, -x_n)$. We thus focus on the case $x_n(t_0) > A$.

If $x_n(t) \geqslant 0$ for some $t \geqslant 0$, then

$$\dot{x}_n(t) = \sum_{j=1}^{n} \left( \frac{e^{x_n(t)(x_j(t) - x_n(t))}}{\sum_{k=1}^{n} e^{x_n(t)(x_k(t) - x_n(t))}} \right) x_j(t) \tag{22}$$

$$\geqslant \frac{x_n(t)}{1 + (n-1)e^{-cx_n(t)}} + \sum_{\{j \in [n]\,:\, x_j(t) < 0\}} e^{x_n(t)(x_j(t) - x_n(t))} x_j(t) \tag{23}$$

$$\geqslant \frac{x_n(t)}{1 + (n-1)e^{-cx_n(t)}} - n\frac{e^{-x_n(t)^2}}{x_n(t)} \tag{24}$$

$$\geqslant \frac{x_n(t)}{n} - n\frac{e^{-x_n(t)^2}}{x_n(t)}. \tag{25}$$

We provide some detail on the above sequence of inequalities. First of all, to pass from (22) to (23), we use

$$e^{x_n(t)(x_k(t) - x_n(t))} \leqslant e^{-cx_n(t)}$$

for $j = n$ and for any $k \in [n]$ (which holds by virtue of (21)), combined with the fact that

$$\sum_{k=1}^{n} e^{x_n(t)(x_k(t)-x_n(t))} \geqslant 1$$

for all indices $j$ such that $x_j(t) < 0$. To pass from (23) to (24), we use $e^{x_n(t)z}z \geqslant -\frac{1}{x_n(t)}$, which holds for any $z \leqslant 0$.

For any $B > A$, we clearly have

$$\frac{B}{n} - n\frac{e^{-B^2}}{B} > 0.$$

We then deduce from (24) and the fact that $x_n(t_0) > A$ that $x_n(t) \to +\infty$ as $t \to +\infty$. Moreover due to the fact that the expression in (25) is bounded from below by $\frac{x_n(t)}{2n}$ whenever $x_n(t)$ is sufficiently large, we deduce that

$$x_n(t) \geqslant c_0 e^{\frac{t}{2n}}$$

for any sufficiently large $t > 0$.

Coming back to (24), we find that for sufficiently large $t > 0$,

$$\dot{x}_n(t) \geqslant x_n(t)\left(\frac{1}{1+(n-1)e^{-cc_0 e^{\frac{t}{2n}}}} - e^{-c_0^2 e^{\frac{t}{n}}}\right).$$

This implies that

$$\frac{\mathrm{d}}{\mathrm{d}t}\log(x_n(t)) \geqslant 1 - O\left(e^{-\frac{t}{3n}}\right),$$

whence

$$\log(x_n(t)) \geqslant t + O(1)$$

for sufficiently large $t > 0$, as desired. $\qquad\square$

Here and in what follows, $\delta_{jk}$ denotes the Kronecker symbol.

**Lemma B.4.** *If $i \in [n]$ is such that $x_i(t)$ is not uniformly bounded with respect to $t > 0$, then $x_i(t)$ converges to either $-\infty$ or $+\infty$ as $t \to +\infty$. Moreover,*

1. *if $x_i(t) \to +\infty$, then for any $j \in [n]$, $P_{ij}(t)$ converges to $\delta_{nj}$ as $t \to +\infty$, with doubly exponential rate.*

2. *if $x_i(t) \to -\infty$, then for any $j \in [n]$, $P_{ij}(t)$ converges to $\delta_{1j}$ as $t \to +\infty$, with doubly exponential rate.*

*Proof.* We assume that $x_i(t)$ is not uniformly bounded with respect to $t > 0$. Without loss of generality, we assume that there exists a sequence of positive times $\{t_k\}_{k=1}^{+\infty}$ with $t_k \to +\infty$ such that $x_i(t_k) \to +\infty$. Necessarily, $x_n(t_k) \to +\infty$. We notice that if $x_i(t) > 0$ for some $t \geqslant 0$, then, arguing as in (22)–(23)–(24), we have

$$\dot{x}_i(t) = \sum_{j=1}^{n}\left(\frac{e^{x_i(t)(x_j(t)-x_n(t))}}{\sum_{k=1}^{n}e^{x_i(t)(x_k(t)-x_n(t))}}\right)x_j(t) \geqslant \frac{x_n(t)}{n} - \frac{n}{x_i(t)}e^{-x_i(t)x_n(t)}. \qquad (26)$$

For sufficiently large integers $k \geqslant 1$, from (26) we get $\dot{x}_i(t_k) > 0$ and $\dot{x}_n(t_k) > 0$. But as $x_i$ and $x_n$ increase, the lower bound in (26) becomes larger. It follows that

$$\dot{x}_i(t) \geqslant \frac{x_n(t)}{2n} \geqslant \frac{x_i(t)}{2n}$$

for sufficiently large $t$, implying that $x_i(t) \to +\infty$ with exponential rate as $t \to +\infty$.

We now prove point 1. regarding $P(t)$. We assume that $x_i(t) \to +\infty$ as $t \to +\infty$. In this case, for $j \neq n$ (namely $j \in [n-1]$),

$$P_{ij}(t) = \frac{e^{x_i(t)x_j(t)}}{\displaystyle\sum_{k=1}^{n}e^{x_i(t)x_k(t)}} \leqslant e^{x_i(t)(x_j(t)-x_n(t))} \leqslant e^{-cx_i(t)},$$

thus $P_{ij}(t)$ converges to $0$ as $t \to +\infty$ (with doubly exponential rate). Consequently, we also deduce that

$$P_{in}(t) = 1 - \sum_{j=1}^{n-1} P_{ij}(t)$$

converges to $1$, also with doubly exponential rate, as $t \to +\infty$.

The case where $x_i(t) \to -\infty$ is symmetric. This concludes the proof. $\qquad \square$

Our last result is useful in the next section.

**Lemma B.5.** *For any $i \in [n]$ such that $x_i(t)$ is not uniformly bounded with respect to $t > 0$, there exists some $\gamma_i \in \mathbb{R}$, $\gamma_i \neq 0$ such that $x_i(t) = \gamma_i e^t + o(e^t)$ as $t \to +\infty$.*

*Proof.* Without loss of generality we assume that $x_i(t) \to +\infty$ as $t \to +\infty$. For $j \neq n$, we find

$$P_{ij}(t) = \frac{e^{x_i(t)x_j(t)}}{\displaystyle\sum_{k=1}^{n} e^{x_i(t)x_k(t)}} = \frac{e^{x_i(t)(x_j(t)-x_n(t))}}{\displaystyle\sum_{k=1}^{n} e^{x_i(t)(x_k(t)-x_n(t))}} \leqslant e^{-cx_i(t)}.$$

Consequently,

$$P_{in}(t) \geqslant 1 - n e^{-cx_i(t)}.$$

Therefore, using Lemma B.3 and the fact that $x_i(t) \geqslant b_i e^{\frac{t}{2n}}$ for some $b_i > 0$ (thanks to (26)), we gather that

$$\dot{x}_i(t) \geqslant \left( 1 - n e^{-cx_i(t)} \right) x_n(t) - n e^{-cx_i(t)} c_1 e^t$$

$$\geqslant \left( 1 - n e^{-cb_i e^{\frac{t}{2n}}} \right) x_n(t) - n e^{-cb_i e^{\frac{t}{2n}}} c_1 e^t \qquad (27)$$

for some $c_1 > 0$ independent of $t$. We also notice that due to (17), $\dot{x}_i(t) \leqslant x_n(t)$. Using (27), firstly for $i = n$, together with the trivial upper bound $x_n(t) \leqslant C e^t$ for some $C > 0$ independent of $t$ (immediately seen from (17)), we obtain

$$\dot{x}_n(t) = x_n(t) \left( 1 + o\left( e^{-cb_i e^{\frac{t}{3n}}} \right) \right)$$

as $t \to +\infty$, which yields

$$x_n(t) = \gamma_n e^t + o(e^t)$$

for some $\gamma_n > 0$. Now using (27) for the index $i$, we gather that

$$\dot{x}_i(t) = x_n(t) + o\left( e^{-cb_i e^{\frac{t}{3n}}} \right),$$

and so we deduce that

$$x_i(t) = \gamma_n e^t + o(e^t).$$

Similarly, if $x_i(t) \to -\infty$, then $x_i(t) = \gamma_1 e^t + o(e^t)$. This proves Lemma B.5 (and shows that $\gamma_i \in \{\gamma_1, \gamma_n\}$). $\qquad \square$

## B.2 Results about bounded particles

In this section we collect results concerning particles which remain uniformly bounded in time. The following lemma entails that there can be at most one particle with this property.

**Lemma B.6.** *Consider*

$$\mathscr{B} := \left\{ i \in [n] : x_i(\cdot) \in L^\infty([0, +\infty)) \right\}.$$

*Then $\#\mathscr{B} \in \{0, 1\}$.*

*Proof.* We first prove that either $x_1(t) \to -\infty$ or $x_n(t) \to +\infty$ as $t \to +\infty$. By contradiction, if this is not the case, then by Lemma B.3, $(x_1(t), \ldots, x_n(t)) \in [-A, A]^n$ for any $t \geqslant 0$. We denote by $\mathscr{I}$ the set of configurations $(x_1^*, \ldots, x_n^*) \in [-A, A]^n$ such that $|x_i^* - x_j^*| \geqslant |x_i(0) - x_j(0)| > 0$ for any distinct $i, j \in [n]$. For any $\mathbf{X}^* = (x_1^*, \ldots, x_n^*) \in \mathscr{I}$, the function $f$ defined in (18) (with anchor points given by $\mathbf{X}^*$) is strictly convex—the equality in the inequality between (19) and (20) is never achieved. Therefore, the proof of Lemma B.2 shows that if $\mathbf{X}^*$ is seen as an initial datum for the dynamics (17), then

$$v(\mathbf{X}^*) := \frac{\mathrm{d}}{\mathrm{d}t}_{|t=0} |x_1^*(t) - x_n^*(t)| > 0.$$

Since $\mathscr{I}$ is compact, $v_0 := \inf_{\mathbf{X}^* \in \mathscr{I}} v(\mathbf{X}^*) > 0$. Hence, $t \mapsto |x_1(t) - x_n(t)|$ grows at least linearly, which is a contradiction.

We may therefore assume without loss of generality that $x_1(t) \to -\infty$ as $t \to +\infty$. We prove that $x_n(t)$ converges to either $-\infty$, or 0, or $+\infty$, as $t \to +\infty$. We assume in the sequel that $x_n(t)$ does not converge to $-\infty$ or 0. For any $i \in [n]$, if there exists $\varepsilon > 0$ and a sequence of positive times $\{s_k\}_{k=1}^{+\infty}$ tending to $+\infty$ such that $x_i(s_k) \leqslant -\varepsilon$, then it follows from (26) that $x_i(t) \to -\infty$. Therefore, by our assumptions, we have $\liminf_{t \to +\infty} x_n(t) \geqslant 0$. Also, since $x_n(t) \not\to 0$, there exists $\varepsilon > 0$ and a sequence of positive times $\{t_k\}_{k=1}^{+\infty}$ tending to $+\infty$ such that $x_n(t_k) \geqslant \varepsilon$ for any integer $k \geqslant 1$. For any $t \geqslant 0$ such that $x_n(t) \geqslant \varepsilon$, we introduce the set of indices

$$N(t) = \{i \in [n] : x_i(t) < 0\},$$

and we write

$$\dot{x}_n(t) \geqslant \frac{e^{x_n(t)^2} x_n(t)}{\sum\limits_{k=1}^n e^{x_n(t) x_k(t)}} + \frac{\sum\limits_{j \in N(t)} e^{x_j(t) x_n(t)} x_j(t)}{\sum\limits_{k=1}^n e^{x_n(t) x_k(t)}} \geqslant \frac{\varepsilon}{n} + \frac{1}{e^{\varepsilon^2}} \sum\limits_{j \in N(t)} e^{\varepsilon x_j(t)} x_j(t). \tag{28}$$

According to Lemma B.4, any point $x_i(t)$ which takes negative values for arbitrarily large times and does not converge to $-\infty$ has to converge to 0. Therefore, the second term in the lowermost bound in (28) is lower bounded by $-\frac{\varepsilon}{2n}$ for sufficiently large $t$. All in all, we gather that $\dot{x}_n(t) \geqslant \frac{\varepsilon}{2n}$ and $x_n(t)$ converges to $+\infty$ as $t \to +\infty$. If it converges to 0, then necessarily $x_{n-1}(t) \to -\infty$ by combining Lemma B.2 with Lemma B.4. This proves Lemma B.6 in this case.

From now on we assume that $x_n(t) \to +\infty$. Using (26) we see that if there exists $\varepsilon > 0$ such that $x_i(t) > \varepsilon$ for an unbounded sequence of times $t$, then $x_i(t) \to +\infty$. The same is true symmetrically when $x_i(t) < -\varepsilon$ for an unbounded sequence of times $t$. Thus if $i \in \mathscr{B}$, necessarily $x_i(t) \to 0$. By Lemma B.2 this can be true for at most one index $i$, which concludes the proof of Lemma B.6. $\quad\square$

If $\mathscr{B} = \varnothing$, Theorem 2.1 follows from Lemma B.4. From now on, we assume that $\#\mathscr{B} = 1$, and we denote by $i_0 \in [n]$ its unique element. We distinguish two cases: either $i_0 \in \{1, n\}$ (Lemma B.7), or $i_0 \notin \{1, n\}$ (Lemma B.8).

**Lemma B.7.** *If $x_n(t)$ is bounded as $t \to +\infty$, then $P_{nn}(t) \to 1$, and $P_{nj}(t) \to 0$ for any $j \in [n-1]$, as $t \to +\infty$. Similarly, if $x_1(t)$ is bounded as $t \to +\infty$, then $P_{11}(t) \to 1$, and $P_{1j}(t) \to 0$ for any $j \in [n-1]$, as $t \to +\infty$.*

*Proof.* The two cases ($t \mapsto x_n(t)$ bounded or $t \mapsto x_1(t)$ bounded) are symmetric since the evolution (17) commutes with the involution of $(\mathbb{R}^d)^n$ given by $(x_1, \ldots, x_n) \mapsto (-x_1, \ldots, -x_n)$. Whence, we only address the first one: we assume that $x_n(t)$ is bounded as $t \to +\infty$. We first notice that all particles $x_j(t)$ for $j \in [n-1]$ tend to $-\infty$ as $t \to +\infty$ due to Lemma B.6. We now prove the following properties:

1. $x_n(t) > 0$ for any sufficiently large $t$;

2. $x_n(t) \to 0$ as $t \to +\infty$;

3. for any $j \in [n-1]$, $P_{nj}(t) \to 0$ as $t \to +\infty$.

To prove point 1., we notice that for sufficiently large $t$, $x_i(t) \leqslant 0$ for any $i \in [n-1]$. If in addition $x_n(t) \leqslant 0$, then due to (17), all $x_i(t)$ ($i \in [n]$) remain negative and due to (17), $x_n(t) \to -\infty$ as $t \to +\infty$, which is a contradiction.

For point 2., we fix $\varepsilon > 0$, and set

$$\mathsf{T}_\varepsilon^+ := \{t \geqslant 0 \colon x_n(t) \geqslant \varepsilon\}.$$

We prove that if $\mathsf{T}_\varepsilon^+$ is unbounded, then $x_n(t) \to +\infty$ as $t \to +\infty$, which is a contradiction. As a consequence, $\mathsf{T}_\varepsilon^+$ is bounded for any $\varepsilon > 0$, which implies (in conjunction with point 1.) that $x_n(t) \to 0$ as $t \to +\infty$. So let us assume that $\mathsf{T}_\varepsilon^+$ is unbounded. We notice that for any $\delta > 0$, if $t \in \mathsf{T}_\varepsilon^+$ is sufficiently large then

$$\left| e^{x_n(t)x_j(t)} x_j(t) \right| \leqslant \delta$$

for any $j \in [n-1]$ since $x_j(t) \to +\infty$ as $t \to +\infty$. Therefore,

$$\sum_{j=1}^n e^{x_n(t)x_j(t)} x_j(t) \geqslant e^{\varepsilon^2} \varepsilon - (n-1)\delta \geqslant 0,$$

where we took $\delta > 0$ sufficiently small for the last inequality to hold. Consequently,

$$\dot{x}_n(t) = \frac{\displaystyle\sum_{j=1}^n e^{x_n(t)x_j(t)} x_j(t)}{\displaystyle\sum_{j=1}^n e^{x_n(t)x_j(t)}} \geqslant \frac{e^{x_n(t)^2} x_n(t) - (n-1)\delta}{e^{x_n(t)^2} + n - 1}.$$

It is not difficult to see that this implies that $x_n(t) \to +\infty$ as $t \to +\infty$, which is a contradiction.

For point 3., we first notice that for any $j \neq n$, since $x_j(t) \to -\infty$,

$$\dot{x}_j(t) = \sum_{k=1}^n \left( \frac{e^{x_j(t)(x_k(t)-x_n(t))}}{\sum_{\ell=1}^n e^{x_j(t)(x_\ell(t)-x_n(t))}} \right) x_k(t) \leqslant \frac{x_1(t)}{n} + \frac{n}{\varepsilon} e^{-x_j(t)x_n(t)}.$$

Using Lemma B.3, we deduce the existence of some $c_2 > 0$ such that

$$x_j(t) \leqslant -c_2 e^t$$

for any sufficiently large $t > 0$. We now prove that for any $j \neq n$,

$$x_j(t)x_n(t) - x_n(t)^2 \xrightarrow[t \to +\infty]{} -\infty. \tag{29}$$

Due to the ordering of the particles, it is enough to prove (29) for $j = n - 1$. Fix $j = n - 1$ and $\kappa > 0$, and assume that

$$x_n(t)x_j(t) \geqslant x_n(t)^2 - \kappa$$

for some $t \geqslant 0$. Then, using the fact that

$$x_n(t)x_j(t) \geqslant x_n(t)x_k(t)$$

for any $k \in [n-2]$, we get

$$P_{nj}(t) \geqslant \frac{e^{x_j(t)x_n(t)}}{e^{x_n(t)^2} + (n-1)e^{x_n(t)x_j(t)}} \geqslant \varepsilon,$$

where $\varepsilon = \frac{1}{n+e^\kappa}$. We obtain

$$\dot{x}_n(t) \leqslant P_{nn}(t)x_n(t) + P_{nj}(t)x_j(t) \leqslant x_n(t) + \varepsilon x_j(t),$$

hence

$$\begin{aligned}
\frac{\mathrm{d}}{\mathrm{d}t}\big(x_n(t)(x_n(t) - x_j(t))\big) &= \dot{x}_n(t)(2x_n(t) - x_j(t)) - x_n(t)\dot{x}_j(t) \\
&\leqslant (x_n(t) + \varepsilon x_j(t))(2x_n(t) - x_j(t)) - x_n(t)\dot{x}_j(t) \\
&= -\varepsilon x_j(t)^2 + x_n(t)(2\varepsilon x_j(t) + 2x_n(t) - x_j(t) - \dot{x}_j(t)) \\
&\leqslant -\varepsilon x_j(t)^2 + x_n(t)(2x_n(t) - 2x_1(t)), \tag{30}
\end{aligned}$$

where in the last line we used the fact that $\dot{x}_j(t) \geqslant x_1(t)$, which is due to (17), and that $x_1(t) < x_j(t)$, which is due to the ordering of the particles. Since $x_j(t) \leqslant -c_2 e^t$ and $x_1(t) \geqslant -c_1 e^t$, the upper bound in (30) is negative if $t$ is large enough. We therefore conclude that for any fixed $\kappa$, if there exist unbounded times $t$ such that $x_n(t)x_j(t) \geqslant x_n(t)^2 - \kappa$, then $x_n(t)x_j(t) \geqslant x_n(t)^2 - \kappa$ for any $t$ large enough. But this is excluded since $x_n(t) > 0$ and $x_j(t) \to -\infty$ as $t \to +\infty$. This concludes the proof of (29), and the lemma follows by plugging this information into the definition of $P_{nj}(t)$. $\qquad\square$

**Lemma B.8.** *If $i_0 \notin \{1, n\}$ and $x_{i_0}(t)$ remains uniformly bounded in t, then for any $j \in [n-1]$, there exists some $\alpha_j \in [0, 1]$ such that $P_{i_0 j}(t) \to \alpha_j$ as $t \to +\infty$.*

*Proof.* Assume that $i_0 \notin \{1, n\}$. Then $x_1(t) \to -\infty$ and $x_n(t) \to +\infty$ as $t \to +\infty$. Also, $x_{i_0}(t) \to 0$ due to (26).

We write $x_{i_0}(t) = y_{i_0}(t)e^{-t}$. Since $\gamma_n > 0$ and $\gamma_1 < 0$, we notice that the function

$$g : \theta \mapsto \frac{\displaystyle\sum_{i \in [n] \setminus \{i_0\}} e^{\gamma_i \theta} \gamma_i}{1 + \displaystyle\sum_{i \in [n] \setminus \{i_0\}} e^{\gamma_i \theta}}$$

takes value $-\infty$ at $-\infty$, and $+\infty$ at $+\infty$, and has a positive derivative. Thus, it takes the value $0$ exactly once, and we denote this point by $\theta_0$. We prove that $y_{i_0}(t) \to \theta_0$ as $t \to +\infty$. We observe that

$$e^{x_{i_0}(t)^2} = 1 + o(1).$$

Using Lemma B.5 we have

$$\dot{y}_{i_0}(t) = e^t \dot{x}_{i_0}(t) - y_{i_0}(t)$$

$$= (P_{i_0 i_0}(t) - 1)y_{i_0}(t) + e^{2t} \sum_{j \in [n] \setminus \{i_0\}} \left( \frac{e^{y_{i_0}(t)(\gamma_j + o(1))}}{1 + o(1) + \displaystyle\sum_{k \in [n] \setminus \{i_0\}} e^{y_{i_0}(t)(\gamma_k + o(1))}} \right)(\gamma_j + o(1)).$$

We recognize that the sum in the above expression is roughly equal to $g(y_{i_0})$. If the latter is not close to 0 for large times, then $\dot{y}_{i_0}(t)$ necessarily have a huge magnitude due to the $e^{2t}$ factor, leading to a contradiction. Fix $\varepsilon > 0$. If $y_{i_0}(t) > \theta_0 + \varepsilon$ for some large time $t > 0$, then, noticing that

$$|y_{i_0}(t)| = e^t |x_{i_0}(t)| = o(e^t), \tag{31}$$

we get

$$\dot{y}_{i_0}(t) = o(e^t) + e^{2t}\Big(g\big(y_{i_0}(t) + o(y_{i_0}(t))\big)\Big).$$

But $g(y_{i_0}(t)) \geqslant \delta = \delta(\varepsilon)$, and hence

$$\dot{y}_{i_0}(s) \geqslant \frac{\delta}{2}e^{2s}$$

for any larger time $s \geqslant t$, which contradicts (31). We get a similar contradiction if $y_{i_0}(t) < \theta_0 - \varepsilon$ for large enough $t$. This concludes the proof that $y_{i_0}(t) \to \theta_0$.

As a consequence, $x_{i_0}(t)x_i(t) \to \theta_0 \gamma_i$ for any $i \neq i_0$, and we deduce Lemma B.8. $\qquad\square$

### B.3 Concluding the proof of Theorem 2.1

*Proof of Theorem 2.1.* By Lemma B.6, there is at most one index $i_0 \in [n]$ for which the particle $x_{i_0}(t)$ remains bounded for any $t > 0$. In turn, for any $i \in [n] \setminus \{i_0\}$, we may invoke Lemma B.4 which entails that $P_{ij}(t)$ converges to either $\delta_{1j}$ or $\delta_{nj}$ as $t \to +\infty$ (with doubly exponential rate). And by ordering of the particles, for indices $i_1 \leqslant i_2$ different from $i_0$, and $P_{i_1 j}(t) \to \delta_{nj}$ then necessarily $P_{i_2 j}(t) \to \delta_{nj}$ as well. Consequently, all but at most one row of $P(t)$ converge to either $e_1 = (1, 0, \ldots, 0)$ or $e_n = (0, \ldots, 1)$ as $t \to +\infty$. For the $i_0$-th row, we may invoke either Lemma B.7 or Lemma B.8. The former applies if $i_0 \in \{1, n\}$, and entails that the $i_0$-th row of $P(t)$ converges either to $e_1$ or $e_n$, while the latter applies if $i_0 \notin \{1, n\}$, and entails that the $i_0$-th row of $P(t)$ converges to some vector $\alpha \in \mathbb{R}^d$ with non-negative entries. Finally, since the $i_0$-th row of $P(t)$ has entries which sum up to 1, then so does $\alpha$. These conclusions lead us to a final limit matrix $P^*$ which has precisely the form indicated in Fig. 2 (namely, $P^* \in \mathscr{P}$), as desired. $\qquad\square$

**Remark 5** (Higher dimensions). *The extension of Theorem 2.1 to $d \geqslant 2$ is not straightforward due to rare pathological situations. For example, suppose $d = 2$, $n = 2$, and the initial configuration $x_1(0) = (1, \varepsilon)$ and $x_2(0) = (1, -\varepsilon)$. One can check that $x_i(t) \to (1, 0)$ as $t \to +\infty$, for $i = 1, 2$, which means that a single cluster appears. However, the self-attention matrix converges toward the identity (which has rank 2). Therefore, it is not true in full generality that the rank of the limiting self-attention matrix is equal to the number of clusters as $t \to +\infty$, although we believe that the result is true for almost all initial conditions.*

# C   Clustering toward vertices of convex polytopes: proofs of Theorems C.5 and 3.1

In this section, we focus on proving the result in the case
$$V = I_d.$$
We also provide a full picture of the behavior of the dynamics in the case $V = -I_d$ in Appendix C.2.

## C.1   Clustering towards vertices of convex polytopes: Theorem 3.1

In this section, we prove Theorem C.1—namely, we show that particles $\{z_i(t)\}_{i \in [n]}$ following the rescaled dynamics
$$\dot{z}_i(t) = \sum_{j=1}^{n} \left( \frac{e^{e^{2t}\langle Az_i(t), Az_j(t)\rangle}}{\sum_{k=1}^{n} e^{e^{2t}\langle Az_i(t), Az_k(t)\rangle}} \right) (z_j(t) - z_i(t)) \tag{32}$$
converge, as $t \to \infty$, toward points lying on the boundary of a particular convex polytope. In (32) we made use of the shorthand notation
$$A := \left( Q^\top K \right)^{\frac{1}{2}}. \tag{33}$$
The precise statement is the following:

**Theorem C.1.** *Suppose $V = I_d$ and $Q^\top K > 0$. Then, for any initial datum $\{z_i(0)\}_{i \in [n]} \subset \mathbb{R}^d$, the solution to (32) is such that its convex hull $\mathrm{conv}\left(\{z_i(t)\}_{i \in [n]}\right)$ converges to some convex polytope $\mathcal{K} \subset \mathbb{R}^d$ as $t \to +\infty$. Furthermore, let $\mathcal{V} = \{v_1, \ldots, v_m\}$ ($m \leqslant n$) denote the set of vertices of $\mathcal{K}$, and consider*
$$\mathcal{S} := \left\{ x \in \mathcal{K} \colon \|Ax\|^2 = \max_{j \in [m]} \langle Ax, Av_j \rangle \right\},$$
*with $A$ defined in (33). Then $\mathcal{S}$ has finite cardinality, and $\mathcal{V} \subset \mathcal{S} \subset \partial\mathcal{K} \cup \{0\}$. Finally, for any $i \in [n]$ there exists a point $\bar{z} \in \mathcal{S}$ such that $z_i(t) \to \bar{z}$ as $t \to +\infty$. In particular, $z_i(t)$ converges either to some point on the boundary of $\mathcal{K}$, or to $0$.*

### C.1.1   The convex hull is shrinking

To prove Theorem C.1, we begin with the following illustrative result.

**Proposition C.2.** *Suppose $V = I_d$ and $Q^\top K > 0$. Then the solution $\{z_i(\cdot)\}_{i \in [n]}$ to (32) is such that $t \mapsto \mathrm{conv}(\{z_i(t)\}_{i \in [n]})$ is non-increasing in the sense of set-inclusion.*

*Proof of Proposition C.2.* Fix $t > 0$ and let $H \subset \mathbb{R}^d$ be a closed half-space which does not contain any of the points $z_i(t)$. We define the map
$$\alpha : s \mapsto \min_{i \in [n]} \mathrm{dist}(z_i(s), H)$$
for $s \geqslant 0$. We claim that
$$\alpha \text{ is non-decreasing on } [t, +\infty). \tag{34}$$
Before proving (34), let us show how to conclude the proof of Proposition C.2 using this claim. It follows from (34) that if $\mathrm{conv}(\{z_i(t)\}_{i \in [n]}) \cap H = \varnothing$, then $\mathrm{conv}(\{z_i(t')\}_{i \in [n]}) \cap H = \varnothing$ for any $t' \geqslant t$. Writing the convex set $\mathrm{conv}(\{z_i(t)\}_{i \in [n]})$ as
$$\mathrm{conv}(\{z_i(t)\}_{i \in [n]}) = \bigcap_{\substack{H' \text{ open half-space} \\ \mathrm{conv}(\{z_i(t)\}_{i \in [n]}) \subset H'}} H' = \bigcap_{\substack{H \text{ closed half-space} \\ \mathrm{conv}(\{z_i(t)\}_{i \in [n]}) \cap H = \varnothing}} \mathbb{R}^d \backslash H,$$

we get that $\mathrm{conv}(\{z_i(t')\}_{i\in[n]}) \subset \mathrm{conv}(\{z_i(t)\}_{i\in[n]})$ for any $t' \geqslant t$.

We now turn to the proof of the claim (34). Denoting by $\mathbf{n}$ the unit outer normal to $H$ and by $\mathrm{proj}_H$ the orthogonal projection onto the closed set $H$, we have

$$\mathrm{dist}(x, H) = \langle x - \mathrm{proj}_H(x), \mathbf{n}\rangle.$$

If $t \mapsto x(t)$ is a differentiable curve, writing $\dot{x}(t) = \langle \dot{x}(t), \mathbf{n}\rangle\mathbf{n} + v(t)$ where $v(t) \in H$ we have $\frac{\mathrm{d}}{\mathrm{d}t}(\mathrm{proj}_H(x(t))) = v(t)$, whence

$$\frac{\mathrm{d}}{\mathrm{d}t}\mathrm{dist}(x(t), H) = \langle \dot{x}(t), \mathbf{n}\rangle. \tag{35}$$

Let $T > t$ denote the infimum of the times for which one of the points $z_i(t)$ lies in $H$. Now fix $s \in [t, T)$, and denote by $M(s)$ the set of indices $i \in [n]$ such that $\mathrm{dist}(z_i(s), H)$ is minimal. For $h \to 0$, we have

$$\alpha(s + h) = \min_{i\in M(s)} \mathrm{dist}(z_i(s+h), H) = \min_{i\in M(s)}\left(\mathrm{dist}(z_i(s), H) + h\frac{\mathrm{d}}{\mathrm{d}t}\mathrm{dist}(z_i(s), H) + o(h)\right)$$

$$= \alpha(s) + h\left(\min_{i\in M(s)}\frac{\mathrm{d}}{\mathrm{d}t}\mathrm{dist}(z_i(s), H)\right) + o(h).$$

Consequently,

$$\frac{\mathrm{d}\alpha}{\mathrm{d}t}(s) = \min_{i\in M(s)}\frac{\mathrm{d}}{\mathrm{d}t}\mathrm{dist}(z_i(s), H).$$

Moreover, for any $i \in M(s)$, one has

$$\frac{\mathrm{d}}{\mathrm{d}t}\mathrm{dist}(z_i(s), H) \overset{(35)}{=} \langle \dot{z}_i(s), \mathbf{n}\rangle = \sum_{j=1}^{n} P_{ij}(s)\langle z_j(s) - z_i(s), \mathbf{n}\rangle \geqslant 0,$$

where the last inequality comes from the fact that each term in the sum is non-negative, since $i \in M(s)$. This proves (34) (and, as a byproduct, that $T = +\infty$). $\qquad\square$

The following fact immediately ensues.

**Corollary C.3.** *For any $i \in [n]$ and $t \geqslant 0$, $z_i(t) \in \mathrm{conv}(\{z_i(0)\}_{i\in[n]})$. In particular, $z_i(\cdot)$ is uniformly bounded in time.*

### C.1.2 Proof of Theorem C.1

*Proof of Theorem C.1.* As a consequence of Proposition C.2, the set $\mathrm{conv}(\{z_i(t)\}_{i\in[n]})$ converges as $t \to +\infty$ toward some convex polytope $\mathcal{K}$. In the remainder of the proof, we look to show that the particles $z_i(t)$ can in fact converge only to some well-distinguished points lying on the boundary of this polytope.

**Step 1. The candidate set of limit points.** We denote by $\mathcal{V} = \{v_1, \ldots, v_m\}$ the set of vertices of $\mathcal{K}$. Writing any $x \in \mathcal{K}$ as a convex combination of these vertices: $x = \sum_{j=1}^{m}\alpha_j v_j$ for some weights $\alpha_j \geqslant 0$ with $\sum_{j=1}^{m}\alpha_j = 1$, we gather that

$$\|Ax\|^2 = \left\langle Ax, \sum_{j=1}^{m}\alpha_j Av_j\right\rangle = \sum_{j=1}^{m}\alpha_j\langle Ax, Av_j\rangle \leqslant \max_{j\in[m]}\langle Ax, Av_j\rangle. \tag{36}$$

Let $\mathcal{S} \subset \mathcal{K}$ denote the set of points $w \in \mathcal{K}$ such that

$$\|Aw\|^2 = \max_{j\in[m]}\langle Aw, Av_j\rangle. \tag{37}$$

The following holds—we postpone the proof to after that of the theorem.

**Claim 1.** $\mathcal{V} \subset \mathcal{S}$. *Moreover, if $0 \in \mathcal{K}$, then $0 \in \mathcal{S}$. Finally, $\mathcal{S} \subset \partial\mathcal{K}\cup\{0\}$, and $\mathcal{S}$ has finite cardinality.*

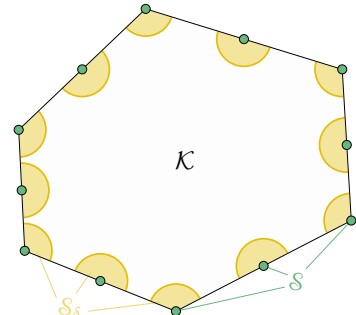

Figure 9: An example configuration of the sets $\mathcal{S}$ and $\mathcal{S}_\delta$ in $\mathbb{R}^2$. The set $\mathcal{S}$ consists of all green nodes along the boundary of $\partial\mathcal{K}$, while $\mathcal{S}_\delta$ is the union of all yellow "hemispheres". The latter are pairwise disjoint and are the connected components of $\mathcal{S}_\delta$, which we denote by $\mathscr{C}_k$, for $k \in [M]$.

Now, for $\delta > 0$, we define the set $\mathcal{S}_\delta$ of points in $\mathcal{K}$ at distance at most $\delta$ from $\mathcal{S}$:

$$\mathcal{S}_\delta := \{x \in \mathcal{K} \colon \operatorname{dist}(x, \mathcal{S}) \leqslant \delta\}.$$

Since $\mathcal{S}$ is finite, there exists a sufficiently small $\delta_0 > 0$ such that for any $\delta \leqslant \delta_0$, the set $\mathcal{S}_\delta$ has $M := \#\mathcal{S}$ connected components, with any two of these connected components being separated by a distance of at least $\delta_0$. Our goal is to prove that for any $i \in [n]$, and for sufficiently large $t$, the particle $z_i(t)$ remains in one of these connected components. In the sequel, we fix $i \in [n]$.

**Step 2.** $z_i(t)$ **must grow if it is not already in** $\mathcal{S}_\delta$**.** We now prove that there exists some $\gamma = \gamma(\mathcal{K}) > 0$ (depending only on the geometry of $\mathcal{K}$) such that for any $\delta \in (0, \delta_0]$, there exists $T(\delta) > 0$ such that if $t \geqslant T(\delta)$ and $z_i(t) \notin \mathcal{S}_\delta$, then

$$\frac{\mathrm{d}}{\mathrm{d}t}\|Az_i(t)\|^2 \geqslant \gamma\delta. \tag{38}$$

To this end, we observe that

$$\frac{1}{2}\frac{\mathrm{d}}{\mathrm{d}t}\|Az_i(t)\|^2 = \langle A\dot{z}_i(t), Az_i(t)\rangle = \sum_{j=1}^n \left( \frac{e^{\langle Az_i(t), Az_j(t)\rangle e^{2t}}}{\sum_{k=1}^n e^{\langle Az_i(t), Az_k(t)\rangle e^{2t}}} \right) \langle A(z_j(t) - z_i(t)), Az_i(t)\rangle$$

$$= \sum_{j=1}^n \underbrace{\left( \frac{e^{a_j(t)e^{2t}}}{\sum_{k=1}^n e^{a_k(t)e^{2t}}} \right) a_j(t)}_{:= b_j(t)} \tag{39}$$

where we have set

$$a_j(t) := \langle A(z_j(t) - z_i(t)), Az_i(t)\rangle.$$

(To obtain the last equality in (39), divide both the numerator and the denominator by $e^{\|Az_i(t)\|^2 e^{2t}}$.) The following holds.

**Claim 2.** *There exists some constant $\gamma' = \gamma'(\mathcal{K}) > 0$ depending only on the geometry of $\mathcal{K}$ such that the following holds. Fix $\delta \in (0, \delta_0]$. There exists $T'(\delta) > 0$ such that if $t \geqslant T'(\delta)$ and $z_i(t) \notin \mathcal{S}_\delta$, then there exists $j \in [n]$ such that $a_j(t) \geqslant \gamma'\delta$.*

We postpone the proof of this claim to after that of the theorem. We seek to use this claim in obtaining a lower bound of $b_j(t)$ for any $j$, whenever $\delta$ is small enough and $t$ is large enough. Since by Corollary C.3, for any $j \in [n]$, $t \mapsto z_j(t)$ is uniformly bounded on $[0, +\infty)$, we gather that $a_j(\cdot) \in L^\infty(0, +\infty)$. So, we may set

$$\kappa := \max_{j \in [n]} \sup_{t \geqslant 0} |a_j(t)|.$$

Let $t \geqslant 0$ be fixed. We define

$$B(t) := \{j \in [n] \colon a_j(t) \geqslant 0\}.$$

We pick an index $j_0(t)$ maximizing $a_j(t)$, namely

$$j_0(t) \in \arg\max_{j \in [n]} a_j(t).$$

Observe that $j_0(t) \in B(t)$ since $a_{j_0(t)}(t) \geqslant a_i(t) = 0$. Clearly

$$b_j(t) \geqslant 0 \qquad \text{for all } j \in B(t). \tag{40}$$

In fact, we also have

$$b_{j_0(t)}(t) \geqslant \frac{a_{j_0(t)}(t)}{n}. \tag{41}$$

Now suppose that $j \notin B(t)$; since $a_j(t) \geqslant -\kappa$, and

$$\frac{e^{a_j(t)e^{2t}}}{\displaystyle\sum_{k=1}^{n} e^{a_k(t)e^{2t}}} \leqslant \frac{1}{\displaystyle\sum_{k=1}^{n} e^{a_k(t)e^{2t}}} \leqslant e^{-a_{j_0(t)}e^{2t}},$$

we gather that

$$b_j(t) \geqslant -\kappa e^{-a_{j_0(t)}(t)e^{2t}} \qquad \text{for all } j \in [n] \backslash B(t). \tag{42}$$

Using (40), (41) and (42) in (39), we find

$$\frac{1}{2} \frac{\mathrm{d}}{\mathrm{d}t} \|Az_i(t)\|^2 \geqslant \frac{a_{j_0(t)}(t)}{n} - \kappa n e^{-a_{j_0(t)}(t)e^{2t}}.$$

The above inequality along with Claim 2 lead us to deduce that there exists $T(\delta) > 0$ (possibly larger than $T'(\delta)$) such that (38) holds whenever $t \geqslant T(\delta)$, with $\gamma = \frac{\gamma'}{2n}$, as desired.

**Step 3: $z_i(t)$ cannot circulate indefinitely between the connected components of $\mathcal{S}_\delta$.** Since $z_i \in L^\infty([0, +\infty))$ by Corollary C.3, from (32) we gather that $\dot{z}_i \in L^\infty([0, +\infty))$ as well. And since any two connected components of $\mathcal{S}_{\delta_0}$ are separated by a distance at least $\delta_0$, we deduce that it takes a time at least

$$T_0 := \frac{\delta_0}{\|\dot{z}_i\|_{L^\infty([0,+\infty))}}$$

for $z_i$ to go from one connected component of $\mathcal{S}_{\delta_0}$ to another one. Fix $\delta \in (0, \delta_0)$ such that

$$\delta < \frac{T_0\gamma\delta_0}{8R\|A\|_{\mathrm{op}}}, \tag{43}$$

where $R := \max_{j \in [n]} \|z_j\|_{L^\infty([0,+\infty))}$. Denote by

$$\mathscr{C}_1, \dots, \mathscr{C}_M$$

the connected components of $\mathcal{S}_\delta$, each of which being the intersection of $\mathcal{K}$ with a Euclidean ball of radius $\delta$ centered at some point of $\mathcal{S}$ (see Fig. 9). For any $k \in [M]$,

$$\sup_{x \in \mathscr{C}_k} \|Ax\|^2 - \inf_{x \in \mathscr{C}_k} \|Ax\|^2 \leqslant 4R\|A\|_{\mathrm{op}}\delta. \tag{44}$$

We introduce the following binary relation on $[M]$:

$$k > \ell \iff \inf_{x \in \mathscr{C}_k} \|Ax\|^2 > \sup_{x \in \mathscr{C}_\ell} \|Ax\|^2,$$

which is transitive. The underlying idea is the following: if $t$ is sufficiently large, and if $z_i$ starts from some connected component $\mathscr{C}_\ell$, then the only connected components $\mathscr{C}_k$ which $z_i$ is able to visit later on are those for which $k > \ell$. This travel of $z_i$ has to stop after some time since $[M]$ is finite, $>$ is transitive, and for any $\ell$, the relation $\ell > \ell$ does not hold.

Let $T = T(\delta)$ be as in Step 2. Suppose that $t_2 > t_1 \geqslant T$ and $k_1, k_2 \in [M]$ are distinct and such that $z_i(t_1) \in \mathscr{C}_{k_1}$, $z_i(t_2) \in \mathscr{C}_{k_2}$ and $z_i(t) \notin \mathcal{S}_\delta$ for any $t \in (t_1, t_2)$. Per Step 2 (more specifically, (38)),

$$\|Az_i(t_2)\|^2 \geqslant \|Az_i(t_1)\|^2 + T_0\gamma\delta_0.$$

Therefore using (44) twice and since $\delta$ is chosen as in (43), we gather that

$$\inf_{x\in\mathscr{C}_{k_2}} \|Ax\|^2 \geqslant \|Az_i(t_2)\|^2 - 4R\|A\|_{\mathrm{op}}\delta \geqslant \|Az_i(t_1)\|^2 + T_0\gamma\delta_0 - 4R\|A\|_{\mathrm{op}}\delta$$

$$\geqslant \inf_{x\in\mathscr{C}_{k_1}} \|Ax\|^2 + T_0\gamma\delta_0 - 4R\|A\|_{\mathrm{op}}\delta$$

$$\geqslant \sup_{x\in\mathscr{C}_{k_1}} \|Ax\|^2 + T_0\gamma\delta_0 - 8R\|A\|_{\mathrm{op}}\delta \qquad (45)$$

$$> \sup_{x\in\mathscr{C}_{k_1}} \|Ax\|^2.$$

Whence $k_2 > k_1$. We therefore deduce that there exist some $T' \geqslant T$ and $k \in [M]$ such that $z_i(t) \notin \mathcal{S}_\delta\backslash\mathscr{C}_k$ for any $t \geqslant T'$.

**Step 4. Conclusion.** To conclude, it remains to be shown that $z_i(t)$ stays in $\mathscr{C}_k$ for $t$ large enough. For this, in addition to (43), we impose

$$\delta^{\frac{1}{4}} < \frac{\gamma T_0}{8R\|A\|_{\mathrm{op}}\delta_0}. \qquad (46)$$

For $r > 0$, we denote by $\mathscr{C}_k^r$ the intersection of $\mathcal{K}$ with the closed Euclidean ball of radius $\delta^r$ having the same center as $\mathscr{C}_k$. In particular, $\mathscr{C}_k^1 = \mathscr{C}_k$. If, after time $T'$, $z_i$ travels from $\mathscr{C}_k$ to the complement of $\mathscr{C}_k^{\frac{1}{4}}$, it spends a time at least

$$\frac{(\delta^{\frac{1}{4}} - \delta^{\frac{1}{2}})}{\|\dot{z}_i\|_{L^\infty([0,+\infty))}}$$

in $\mathscr{C}_k^{\frac{1}{4}}\backslash\mathscr{C}_k^{\frac{1}{2}}$. Per Step 2 (used with $\delta^{\frac{1}{2}}$), $\|Az_i\|^2$ has to increase by at least

$$\frac{\gamma\delta^{\frac{1}{2}}\left(\delta^{\frac{1}{4}} - \delta\right)}{\|\dot{z}_i\|_{L^\infty([0,+\infty))}} \geqslant \frac{\gamma\delta^{\frac{3}{4}}}{2\|\dot{z}_i\|_{L^\infty([0,+\infty))}} > 4R\|A\|_{\mathrm{op}}\delta \qquad (47)$$

during this travel (the last inequality in (47) stems from (46)). This implies that $z_i$ cannot reenter $\mathscr{C}_k$ after having reached the boundary of $\mathscr{C}_k^{\frac{1}{4}}$, due to (44). Thus $z_i(t) \notin \mathcal{S}_\delta$ for any sufficiently large $t$, which is impossible due to Step 2 and the uniform boundedness of $t \mapsto \|Az_i(t)\|$. Hence, for sufficiently large $t$, $z_i(t) \in \mathscr{C}_k^{\frac{1}{4}}$. Since $\delta$ may be chosen arbitrarily small, this concludes the proof of Theorem C.1. $\qquad\square$

### C.1.3 Proving Claims 1 and 2

We now address the proofs of the two claims which were instrumental in what precedes (along with a sketch of the proof of $\mathcal{V} \subset \mathcal{S}$, as implied).

*Proof of Claim 1.* The fact that $0 \in \mathcal{S}$ if $0 \in \mathcal{K}$ is immediate. We now show that $\mathcal{S}$ is finite and $\mathcal{S} \subset \partial\mathcal{K} \cup \{0\}$. Let $w \in \mathcal{S}\backslash\{0\}$. As

$$w = \sum_{j=1}^{m} \alpha_j v_j$$

for some $\alpha_j \geqslant 0$ with $\sum_{j=1}^m \alpha_j = 1$, and since (37) holds by definition, it follows that $\alpha_j = 0$ for any $j$ not attaining the maximum in (37). Let $\mathsf{I} \subset [m]$ denote the set of all such indices. We have

$$w = \sum_{j\in\mathsf{I}} \alpha_j v_j$$

with $\|Aw\|^2 = \langle Aw, Av_j \rangle$ for any $j \in \mathsf{I}$. Whence $w$ is the orthogonal projection onto $\mathrm{span}\{v_j\}_{j\in\mathsf{I}}$ with respect to $\langle A\cdot, A\cdot \rangle$. This yields $\mathcal{S} \subset \partial\mathcal{K}$. Moreover, since for each subset $\mathsf{I} \subset [m]$ there exists a unique such projection $w$, $\mathcal{S}$ is finite. $\qquad\square$

*Sketch of proof of $\mathcal{V} \subset \mathcal{S}$.* We notice that for any $i \in [n]$ and for $t$ large enough, we have

$$\dot{z}_i(t) = \sum_{j=1}^{n} \left( \frac{e^{e^{2t}\langle Az_i(t), Az_j(t)\rangle}}{\sum_{k=1}^{n} e^{e^{2t}\langle Az_i(t), Az_k(t)\rangle}} \right) (z_j(t) - z_i(t)) \tag{48}$$

$$\approx \sum_{j \in M_i(t)} \left( \frac{e^{e^{2t}\langle Az_i(t), Az_j(t)\rangle}}{\sum_{k=1}^{n} e^{e^{2t}\langle Az_i(t), Az_k(t)\rangle}} \right) (z_j(t) - z_i(t)), \tag{49}$$

where $M_i(t)$ is the subset of $[n]$ containing all indices $j$ such that

$$\max_{k \in [n]} \langle Az_i(t), Az_k(t)\rangle - \langle Az_i(t), Az_j(t)\rangle \leqslant e^{-t}$$

(all other terms in the sum (48) are negligible). Due to the convergence of $\mathrm{conv}(\{z_i(t)\}_{i \in [n]})$ toward $\mathcal{K}$, we also know that for $t$ large enough,

- all the points $z_i(t)$ are contained in a small neighborhood of $\mathcal{K}$,

- near any element of $\mathcal{V}$, there exists some particle $z_i(t)$.

Assume, for the sake of contradiction, that there exists a vertex $v_j \in \mathcal{V}$ such that $v_j \notin \mathcal{S}$. Set $\mathcal{C} := \mathrm{conv}(\{v_i\}_{i \in [m] \setminus \{j\}})$. In particular, $\mathrm{dist}(v_j, \mathcal{C}) > 0$ since $v_j$ is a vertex of $\mathcal{K}$. If $\mathsf{I} \subset [n]$ denotes the set of indices $i$ such that $z_i(t)$ lies near $v_j$, then $M_i(t) \cap \mathsf{I} = \varnothing$ for any $i \in \mathsf{I}$, since $v_j \notin \mathcal{S}$. For $i \in \mathsf{I}$, using (49), we find that $\mathrm{dist}(z_i(t), \mathcal{C})$ decays as $t \to +\infty$ as long as $i \notin M_i(t)$—indeed, (49) implies that $z_i(t)$ is attracted by $\mathcal{C}$. This implies that $v_j \notin \mathrm{conv}(\{z_k(t')\}_{k \in [n]})$ for $t'$ large enough. This is a contradiction since $\mathcal{K} \subset \mathrm{conv}(\{z_k(t)\}_{k \in [n]})$ for any $t \geqslant 0$ according to Proposition C.2. $\square$

*Proof of Claim 2.* To simplify the notation, we only prove Claim 2 when $A = I_d$. Assume that $t \geqslant 0$ and that $z_i(t) \notin \mathcal{S}_\delta$.

**First case.** Firstly, we prove the claim in the case where $z_i(t) \notin \mathcal{S}_{\delta_0}$. For this, we notice that the function

$$f : x \mapsto \max_{j \in [n]} \langle v_j, x\rangle - \|x\|^2$$

is continuous, and by definition of $\mathcal{S}$, $f$ is strictly positive on the compact set $\mathcal{K} \backslash \mathrm{Int}(\mathcal{S}_{\delta_0})$ (the complement in $\mathcal{K}$ of the interior of $\mathcal{S}_{\delta_0}$). Hence $f(x) \geqslant c'$ in this set for some constant $c' > 0$. Setting

$$\mathcal{K}_\varepsilon := \{x \in \mathbb{R}^d : \mathrm{dist}(x, \mathcal{K}) \leqslant \varepsilon\},$$

by continuity we find that $f(x) \geqslant c'/2$ for $x \in \mathcal{K}_\varepsilon \backslash \mathrm{Int}(\mathcal{S}_{\delta_0})$ and for sufficiently small $\varepsilon > 0$ (fixed in the sequel). For sufficiently large $t$, we have $z_i(t) \in \mathcal{K}_\varepsilon$ for any $i \in [n]$, thus

$$\max_{j \in [n]} \langle z_i(t), z_j(t) - z_i(t)\rangle \geqslant \max_{j \in [m]} \langle z_i(t), v_j - z_i(t)\rangle \geqslant \frac{c'}{2}.$$

Since $c'$ is independent of $\delta$, we deduce the claim in this case (notice that it suffices to prove the claim for sufficiently small $\delta$).

**Second case.** Secondly, we prove the claim when $z_i(t) \in \mathcal{S}_{\delta_0} \backslash \mathcal{S}_\delta$. The proof mainly relies on the following result:

**Lemma C.4.** *For any $w \in \mathcal{S}$, there exists $\beta > 0$ such that if[6] $x \in \mathcal{K} \cap B(w, \delta_0)$, then*

$$\max_{j \in [m]} \langle x, v_j - x\rangle \geqslant \beta \|x - w\|. \tag{50}$$

We postpone the proof of Lemma C.4 and show how to conclude the proof of Claim 2. Fix $\delta > 0$. We set

$$\eta := \frac{\beta\delta}{6R}$$

---

[6]Here, $B(y, r)$ denotes the closed ball with center $y \in \mathbb{R}^d$ and radius $r > 0$.

where
$$R := \max_{j \in [n]} \|z_j\|_{L^\infty(\mathbb{R})}.$$

Since $\operatorname{conv}(\{z_j(t)\}_{j \in [n]})$ converges to $\mathcal{K}$ as $t \to +\infty$, there exists $T(\delta) > 0$ such that for any $t \geqslant T(\delta)$, if $z_i(t) \in B(w, \delta_0) \backslash B(w, \delta)$ for some $w \in \mathcal{S}$, then

$$\|z_i(t) - x\| \leqslant \eta$$

for some $x \in \mathcal{K} \cap (B(w, \delta_0) \backslash B(w, \delta))$. Therefore, using Lemma C.4,

$$\begin{aligned}
\max_{j \in [m]} \langle z_i(t), v_j - z_i(t) \rangle &\geqslant \max_{j \in [m]} \langle x, v_j - x \rangle - 3R\eta \\
&\geqslant \beta\delta - 3R\eta \\
&= \frac{\beta}{2}\delta.
\end{aligned}$$

To summarize, we have found that for any $\delta > 0$ there exists $T(\delta) > 0$ such that if $t \geqslant T(\delta)$ and $z_i(t) \in \mathcal{S}_{\delta_0} \backslash \mathcal{S}_\delta$, then

$$\max_{j \in [m]} \langle z_i(t), v_j - z_i(t) \rangle \geqslant \frac{\beta}{2}\delta. \tag{51}$$

Combining (51) with

$$\max_{j \in [n]} \langle z_i(t), z_j(t) - z_i(t) \rangle \geqslant \max_{j \in [m]} \langle z_i(t), v_j - z_i(t) \rangle$$

concludes the proof of Claim 2 in this second case. $\square$

*Proof of Lemma C.4.* Let us first address the case where $w = 0$. Writing any $x \in \mathcal{K} \backslash \{0\}$ as a convex combination of the vertices: $x = \sum_{j=1}^m \alpha_j v_j$, we find

$$0 = \left\langle x, \sum_{j=1}^m \alpha_j(v_j - x) \right\rangle = \sum_{j=1}^m \alpha_j \langle x, v_j - x \rangle. \tag{52}$$

We can exclude having $\langle x, v_j - x \rangle = 0$ for all $j \in [m]$, as this would necessarily imply that $\|x\|^2 = 2\sum_{j=1}^m \alpha_j \langle x, v_j - x \rangle = 0$. We deduce from (52) that

$$\max_{j \in [m]} \langle x, v_j - x \rangle > 0$$

for any $x \in \mathcal{K} \backslash \{0\}$. Hence, it is sufficient to prove (50) for $\|x\|$ small enough. We notice that for any $x \in \mathcal{K} \backslash \{0\}$ written as above,

$$\|x\|^2 = \sum_{j=1}^m \alpha_j \langle v_j, x \rangle.$$

Hence $x \mapsto \max_{j \in [m]} \langle v_j, x \rangle$ is positive for $x \in \mathcal{K} \backslash \{0\}$. Since this function is continuous and homogeneous in $x$, we deduce the existence of $\beta > 0$ such that

$$\max_{j \in [m]} \langle v_j, x \rangle \geqslant 2\beta \|x\|$$

for any $x \in \mathcal{K}$. For $x \in \mathcal{K}$ with $\|x\|$ sufficiently small, we obtain (50).

We now assume that $w \in \mathcal{S} \backslash \{0\}$. We set

$$\mathsf{I}_w := \left\{ j \in [n] \colon \|w\|^2 = \langle w, v_j \rangle \right\}$$

and

$$\mathcal{A} := \operatorname{span}\left(\{v_j - w \colon j \in \mathsf{I}_w\}\right),$$

which is orthogonal to $w$. We also introduce

$$\mathcal{R} := \left(\mathbb{R}w \oplus \mathcal{A}\right)^\perp,$$

and we denote by $\pi_{\mathcal{R}}$ the orthogonal projection on $\mathcal{R}$. We claim that there exists some $\rho > 0$ such that for any $j \in [m]$, we have

$$\langle w - v_j, w \rangle \geqslant \rho \|\pi_{\mathcal{R}} v_j\|.$$

This follows from the observation that $[m]$ is finite, and that $\|\pi_{\mathscr{R}} v_j\| > 0$ implies $\langle w - v_j, w \rangle > 0$. Therefore, for any $x \in \mathcal{K}$, writing $x$ as a convex combination of the vertices, namely $x = \sum_{j=1}^{m} \alpha_j v_j$, we find that

$$\rho \|\pi_{\mathscr{R}} x\| \leqslant \sum_{j=1}^{m} \alpha_j \|\pi_{\mathscr{R}} v_j\| \leqslant \sum_{j=1}^{m} \alpha_j \langle w - v_j, w \rangle = \langle w - x, w \rangle. \tag{53}$$

Fix $x \in \mathcal{K} \cap B(w, \delta_0)$. We write $x = w + \delta' u$ with $0 \leqslant \delta' \leqslant \delta_0$ and $\|u\| = 1$. Then we have the orthogonal decomposition

$$u = bw + a + r \tag{54}$$

where $a \in \mathcal{A}$, $r \in \mathscr{R}$ and $b \in \mathbb{R}$. Since $a$ is a convex combination of the form

$$a = \sum_{j \in I_w} \beta_j (v_j - w),$$

we have

$$\|a\|^2 = \sum_{j \in I_w} \beta_j \langle v_j - w, a \rangle,$$

whence

$$\max_{j \in I_w} \langle a, v_j - w \rangle \geqslant \|a\|^2.$$

We deduce that

$$\begin{aligned} \max_{j \in I_w} \langle x, v_j - x \rangle &= \max_{j \in I_w} \langle w + \delta' u, (v_j - w) - \delta' u \rangle \\ &= -\delta' b \|w\|^2 - \delta'^2 + \delta' \max_{j \in I_w} \langle a, v_j - w \rangle \\ &\geqslant -\delta' b \|w\|^2 - \delta'^2 + \delta' \|a\|^2. \end{aligned} \tag{55}$$

Notice that $b \leqslant 0$ by combining (53) and (54). Since $\|u\| = 1$ and using (53) we have

$$1 = b^2 + \|a\|^2 + \|r\|^2 \leqslant \|a\|^2 + \kappa b^2 \leqslant \kappa (\|a\|^2 + b^2)$$

where $\kappa := 1 + \rho^{-2} \|w\|^4$. We deduce that either $\|a\|^2 \geqslant (2\kappa)^{-1}$ or $-b = |b| \geqslant (2\kappa)^{-\frac{1}{2}}$. Plugging this knowledge in (55) and using the fact that $\|w\| > 0$, we finally deduce the existence of an $\alpha > 0$ (independent of $\delta > 0$ and $x \in \mathcal{K} \cap B(w, \delta_0)$) such that

$$\max_{j \in [m]} \langle x, v_j - x \rangle \geqslant \alpha \delta' - \delta'^2 = \alpha \|x - w\| - \|x - w\|^2.$$

This proves (50) when $\|x - w\| \leqslant \alpha/2$.

It thus remains to show that (50) holds for all $x \in \mathcal{K} \cap (B(w, \delta_0) \backslash B(w, \frac{\alpha}{2}))$. To this end, we notice that $x \mapsto \max_{j \in [m]} \langle x, v_j - x \rangle$ is continuous in the connected set $\mathcal{K} \cap (B(w, \delta_0) \backslash B(w, \frac{\alpha}{2}))$, non-negative according to (36), and it is nowhere 0 (by definition of $\mathcal{S}$). Therefore, it is strictly positive, and denote by $\alpha' > 0$ some lower bound. Then for $x \in \mathcal{K} \cap (B(w, \delta_0) \backslash B(w, \frac{\alpha}{2}))$, we have

$$\max_{j \in [m]} \langle x, v_j - x \rangle \geqslant \alpha' \geqslant \frac{\alpha'}{\delta_0} \|x - w\|.$$

This concludes the proof of Lemma C.4. $\qquad \square$

## C.2 A cluster at the origin

We complete this section by addressing the case $V = -I_d$, for which the convergence of the solutions of (1) is the simplest, since a unique cluster forms at the origin. We also suppose that $Q^\top K = I_d$: in other words, we consider the dynamics

$$\dot{x}_i(t) = -\sum_{j=1}^{n} \left( \frac{e^{\langle x_i(t), x_j(t) \rangle}}{\sum_{k=1}^{n} e^{\langle x_i(t), x_k(t) \rangle}} \right) x_j(t), \qquad t \in [0, +\infty), \tag{56}$$

with a prescribed initial condition $\{x_i(0)\}_{i \in [n]} \subset \mathbb{R}^d$.

**Theorem C.5** (Convergence toward the origin). *Suppose $V = -I_d$ and $Q^\top K = I_d$. Then, for any initial sequence of tokens $\{x_i(0)\}_{i \in [n]} \subset \mathbb{R}^d$, and for any $i \in [n]$, we have $\|x_i(t)\| \to 0$ as $t \to +\infty$.*

**Remark 6.** *In the setting of Theorem C.5, the self-attention matrix $P(t)$ defined in (2) converges, as $t \to +\infty$, to the $n \times n$ matrix with all entries equal to $1/n$.*

### C.2.1 Proof of Theorem C.5

We begin by showing that for any $i \in [n]$, the solution to (56) is uniformly bounded for all $t > 0$. In the sequel, we fix an initial configuration $\{x_i(0)\}_{i \in [n]} \subset \mathbb{R}^d$.

**Lemma C.6.** *The trajectories of* (56) *are uniformly bounded in time—namely, there exists $R > 0$ (depending solely on $n$ and the initial configuration) such that the solution $x_i(\cdot)$ to* (56) *satisfies $\|x_i(t)\| \leqslant R$ for any $i \in [n]$ and $t \geqslant 0$.*

*Proof of Lemma C.6.* We fix $i \in [n]$. For $t \geqslant 0$, we denote by $D_i(t)$ the set of points $x_k(t)$ such that $\langle x_i(t), x_k(t) \rangle \geqslant 0$. We also set

$$S_i(t) := \sum_{k \in D_i(t)} e^{\langle x_i(t), x_k(t) \rangle} \langle x_i(t), x_k(t) \rangle,$$

and

$$R_i(t) := \sum_{k=1}^{n} e^{\langle x_i(t), x_k(t) \rangle}.$$

Since $1 + x \leqslant e^x$ whence $e^{-x} x \leqslant 1$, we deduce that

$$\frac{1}{2} \frac{\mathrm{d}}{\mathrm{d}t} \|x_i(t)\|^2 = -\frac{\sum_{k=1}^{n} e^{\langle x_i(t), x_k(t) \rangle} \langle x_i(t), x_k(t) \rangle}{R_i(t)} \leqslant \frac{-S_i(t) + n}{R_i(t)}.$$

Now since $1 - x \leqslant e^{-x}$ whence $e^x \leqslant 1 + e^x x$, we find that $R_i(t) \leqslant n + S_i(t)$. Consequently, if we assume that $\|x_i(t)\|^2 \geqslant 2n$ then $S_i(t) \geqslant 2n$, and therefore

$$\frac{1}{2} \frac{\mathrm{d}}{\mathrm{d}t} \|x_i(t)\|^2 \leqslant \frac{-S_i(t) + n}{n + S_i(t)} \leqslant -1.$$

This shows that $\|x_i(t)\| \leqslant \max\{\|x_i(0)\|, \sqrt{2n}\}$ for any $t \geqslant 0$, which concludes the proof. $\qquad \square$

By virtue of Lemma B.1, we are able to characterize the stationary configurations for the dynamics (56)—namely, the set of points $(\bar{x}_1, \ldots, \bar{x}_n) \in (\mathbb{R}^d)^n$ satisfying

$$\sum_{j=1}^{n} \left( \frac{e^{\langle \bar{x}_i, \bar{x}_j \rangle}}{\sum_{k=1}^{n} e^{\langle \bar{x}_i, \bar{x}_k \rangle}} \right) \bar{x}_j = 0$$

for all $i \in [n]$.

**Lemma C.7.** *The only stationary configuration for the dynamics* (56) *is $\bar{x}_1 = \ldots = \bar{x}_n = 0$.*

*Proof.* Assume that $(\bar{x}_1, \ldots, \bar{x}_n) \in (\mathbb{R}^d)^n$ is a stationary configuration for the dynamics (56). We consider $f : \mathbb{R}^d \to \mathbb{R}$ defined as

$$f : x \mapsto \log \left( \sum_{j=1}^{n} e^{\langle x, \bar{x}_j \rangle} \right).$$

Per Lemma B.1, $f$ is convex, whence

$$f(x) \geqslant f(\bar{x}_i) + \langle \nabla f(\bar{x}_i), x - \bar{x}_i \rangle$$

for $x \in \mathbb{R}^d$ and $i \in [n]$. Since $\nabla f(\bar{x}_i) = 0$ for any $i \in [n]$, we gather that $f(x) \geqslant f(\bar{x}_i)$, whence $\bar{x}_i$ is a global minimizer of $f$ for any $i \in [n]$. By convexity, $f$ is constant on $\mathrm{conv}(\{\bar{x}_i\}_{i \in [n]})$. Since $f$ is analytic on the affine space $E$ spanned by the points $\bar{x}_i$, $i \in [n]$, it is then constant on $E$ as well. Now assume that not all of the points $\bar{x}_i$ are equal, and pick an index $i_0 \in [n]$ such that $\bar{x}_{i_0}$ is not equal to the projection of the origin onto $E$. Then there exists some $j_0 \in [n]$ such that $\langle \bar{x}_{i_0} - \bar{x}_{j_0}, \bar{x}_{i_0} \rangle \neq 0$. For any $s \in \mathbb{R}$, we set $P_s := \bar{x}_{j_0} + s(\bar{x}_{i_0} - \bar{x}_{j_0}) \in E$, and we notice that $f(P_s) \geqslant \langle P_s, \bar{x}_{i_0} \rangle$, where the lower bound tends to $+\infty$ either when $s \to +\infty$ or when $s \to -\infty$. This contradicts the fact that $f$ is constant on $E$. We conclude that the $\bar{x}_i$ are all equal for $i \in [n]$. The only value they can then take is necessarily 0. $\qquad \square$

**Lemma C.8.** *The trajectories of* (56) *satisfy* $\int_0^{+\infty} \|\dot{x}_i(t)\|^2 \, \mathrm{d}t < +\infty$ *for any* $i \in [n]$.

*Proof.* The function

$$\mathscr{L} : t \mapsto \sum_{i=1}^n \sum_{j=1}^n e^{\langle x_i(t), x_j(t) \rangle}$$

is non-increasing, as demonstrated by the following simple computation:

$$\frac{\mathrm{d}\mathscr{L}(t)}{\mathrm{d}t} = 2 \sum_{i=1}^n \sum_{j=1}^n e^{\langle x_i(t), x_j(t) \rangle} \langle \dot{x}_i(t), x_j(t) \rangle = 2 \sum_{i=1}^n \left\langle \dot{x}_i(t), \sum_{j=1}^n e^{\langle x_i(t), x_j(t) \rangle} x_j(t) \right\rangle$$

$$= -2 \sum_{i=1}^n \sum_{j=1}^n e^{\langle x_i(t), x_j(t) \rangle} \|\dot{x}_i(t)\|^2.$$

Being non-negative, $\mathscr{L}(t)$ thus converges as $t \to +\infty$. Since $\langle x_i(t), x_j(t) \rangle \geqslant R$ for some (possibly negative) $R \in \mathbb{R}$ by virtue of Lemma C.6, we deduce that

$$\int_0^{+\infty} \|\dot{x}_i(t)\|^2 \, \mathrm{d}t \leqslant e^{-R} \int_0^{+\infty} \sum_{i=1}^n \sum_{j=1}^n e^{\langle x_i(t), x_j(t) \rangle} \|\dot{x}_i(t)\|^2 \, \mathrm{d}t = e^{-R} (\mathscr{L}(0) - \lim_{t \to +\infty} \mathscr{L}(t)),$$

which concludes the proof. $\qquad\qquad\square$

We are now able to conclude the proof of Theorem C.5.

*Proof of Theorem C.5.* We set $\mathbf{X}(t) := (x_1(t), \ldots, x_n(t)) \in (\mathbb{R}^d)^n$. If $\mathbf{X}(t)$ does not converge to $0$, the compactness provided by Lemma C.6 implies that there is a sequence $\{t_k\}_{k=1}^{+\infty}$ with $t_k \to +\infty$, and $\mathbf{X}^* = (x_1^*, \ldots, x_n^*) \in (\mathbb{R}^d)^n \backslash \{0\}$, such that $\mathbf{X}(t_k) \to \mathbf{X}^*$ as $k \to +\infty$. To conclude the proof, it suffices to show that $\mathbf{X}^*$ is a stationary configuration of the dynamics: this directly leads to a contradiction per Lemma C.7. Therefore, assume that $\mathbf{X}^*$ is not a stationary configuration of the dynamics. We denote by $\mathbf{X}^*(t) = (x_1^*(t), \ldots, x_n^*(t))$ the solution of (56) with initial condition $\mathbf{X}^*$. Then, there exists $i \in [n]$ such that $\dot{x}_i^*(0) \neq 0$. We set $\varepsilon = \|\dot{x}_i^*(0)\|$. We select $T_0 > 0$ (possibly small) such that $\|\dot{x}_i^*(t)\| \geqslant \varepsilon/2$ for $t \in [0, T_0]$. It follows from (13) (which is verified according to Corollary A.4) that for any $\delta > 0$ there exists $k_0 \in \mathbb{N}$ such that $\|\mathbf{X}(t_k + t) - \mathbf{X}^*(t)\| \leqslant \delta$ for any $t \in [0, T_0]$ and any $k \geqslant k_0$. By (9) (which is verified according to Corollary A.4), we obtain that $\|\dot{x}_i(t_k + t) - \dot{x}_i^0(t)\| \leqslant C\delta$ for $t \in [0, T_0]$ and any $k \geqslant k_0$. Choosing $\delta > 0$ sufficiently small, we obtain that $\|\dot{x}_i(t_k + t)\| \geqslant \varepsilon/4$ for $t \in [0, T_0]$ and any $k \geqslant k_0$. This contradicts Lemma C.8. $\qquad\square$

# D   Clustering toward hyperplanes: proof of Theorem 4.1

To ensure clarity, we present the proof of Theorem 4.1 under the assumption that $V$ is diagonalizable. However, this assumption is not necessary. In Remark 8, we explain how the proof can be modified to accommodate for non-diagonalizable $V$.

Let us therefore assume that $V$ is diagonalizable. Let $(\varphi_1, \ldots, \varphi_d)$ be an orthonormal basis of eigenvectors associated to eigenvalues $(\lambda_1, \ldots, \lambda_d)$, ordered in a decreasing manner with respect to their modulus: $|\lambda_1| \geqslant \ldots \geqslant |\lambda_d|$. (Starting from this point and throughout, we use the symbol $\lambda$ exclusively to denote the eigenvalues of $V$.) Except for $\lambda_1 \in \mathbb{R}$, all the other eigenvalues (and eigenvectors) may be complex. We denote by $(\varphi_1^*, \ldots, \varphi_d^*)$ the dual basis of $(\varphi_1, \ldots, \varphi_d)$.

## D.1   Some monotonicity properties and bounds

To start, we present some general facts that are prove useful in all subsequent sub-cases.

**Lemma D.1.** *Suppose $k \in [d]$ is such that $\lambda_k \geqslant 0$. Then $t \mapsto \max_{j \in [n]} \varphi_k^*(z_j(t))$ is a non-increasing and bounded function, and $t \mapsto \min_{j \in [n]} \varphi_k^*(z_j(t))$ is a non-decreasing and bounded function. In particular, $t \mapsto \varphi_k^*(z_i(t))$ is uniformly bounded as a function on $[0, +\infty)$ for any $i \in [n]$.*

*Proof.* For any $k \in [d]$ and any $t \geqslant 0$, set

$$\alpha_k(t) = \min_{j \in [n]} \varphi_k^*(z_j(t)), \qquad \beta_k(t) = \max_{j \in [n]} \varphi_k^*(z_j(t)).$$

Let $i \in [n]$ be an index such that $\alpha_k(t) = \varphi_k^*(z_i(t))$. Then we have

$$\frac{\mathrm{d}}{\mathrm{d}t}\varphi_k^*(z_i(t)) = \sum_{j=1}^n P_{ij}(t)\varphi_k^*\left(V(z_j(t) - z_i(t))\right) = \lambda_k \sum_{j=1}^n P_{ij}(t)(\varphi_k^*(z_j(t)) - \varphi_k^*(z_i(t))) \geqslant 0$$

where the last inequality stems from the fact that $\lambda_k \geqslant 0$ and the choice of index $i$. This proves that $\alpha_k(\cdot)$ is non-decreasing, as desired. Arguing similarly, one finds that $\beta_k(\cdot)$ is non-increasing. As a consequence, $\alpha_k(0) \leqslant \alpha_k(t) \leqslant \beta_k(t) \leqslant \beta_k(0)$ for any $t \geqslant 0$, which shows that $\alpha_k(\cdot)$ and $\beta_k(\cdot)$ are bounded. $\qquad \square$

**Corollary D.2.** *If $V$ only has real non-negative eigenvalues (namely $\mathrm{spec}(V) \subset [0, +\infty)$), then $z_i(\cdot) \in L^\infty([0, +\infty))$.*

**Lemma D.3.** *Fix $k \in [d]$ and $i \in [n]$. Then there exists a constant $C > 0$ such that*

$$\left|\varphi_k^*\left(e^{tV}z_i(t)\right)\right| \leqslant Ce^{|\lambda_k|t}$$

*holds for all $t \geqslant 0$.*

*Proof.* We naturally make use of the equation for $x_i(t) := e^{tV}z_i(t)$. Fix $t \geqslant 0$. We have

$$\begin{aligned}
\frac{\mathrm{d}}{\mathrm{d}t}\left|\varphi_k^*(x_i(t))\right|^2 = 2\mathbf{Re}\left(\overline{\varphi_k^*(x_i(t))}\frac{\mathrm{d}}{\mathrm{d}t}\varphi_k^*(x_i(t))\right) &= 2\mathbf{Re}\left(\sum_{j=1}^n P_{ij}(t)\varphi_k^*(Vx_j(t))\overline{\varphi_k^*(x_i(t))}\right)\\
&= 2\mathbf{Re}\left(\sum_{j=1}^n P_{ij}(t)\lambda_k\varphi_k^*(x_j(t))\overline{\varphi_k^*(x_i(t))}\right)\\
&\leqslant 2|\lambda_k|\max_{j \in [n]}|\varphi_k^*(x_j(t))|^2.
\end{aligned}$$

Choosing $i \in [n]$ running over the set of indices such that $|\varphi_k^*(x_i(t))|$ is maximal, we obtain

$$\frac{\mathrm{d}}{\mathrm{d}t}\max_{j \in [n]}|\varphi_k^*(x_j(t))|^2 \leqslant 2|\lambda_k|\max_{j \in [n]}|\varphi_k^*(x_j(t))|^2.$$

We conclude the proof by applying Grönwall's lemma. $\qquad \square$

### D.2   Proof of Theorem 4.1

We now prove Theorem 4.1. We again recall that $\lambda_1$ is simple and positive, and the eigenvalues of $V$ are ordered in decreasing order of modulus: $\lambda_1 > |\lambda_2| \geqslant \ldots \geqslant |\lambda_d|$.

*Proof of Theorem 4.1.* We look to prove that for any $i \in [n]$, the component of $z_i(t)$ along the principal eigenvector $\varphi_1$, i.e. $\varphi_1^*(z_i(t))$, converges as $t \to +\infty$. We also show that there exists a set of at most 3 real numbers (depending on the initial datum $(z_1(0), \ldots, z_n(0))$) such that for any $i \in [n]$ the limit of $\varphi_1^*(z_i(t))$ belongs to this set. Theorem 4.1 directly follows from these facts.

Let $i \in [n]$ be fixed. Recall from Lemma D.1 that $\varphi_1^*(z_i(t))$ is uniformly bounded for any $t \in [0, +\infty)$. We set

$$a := \lim_{t \to +\infty} \min_{j \in [n]} \varphi_1^*(z_j(t)), \qquad b := \lim_{t \to +\infty} \max_{j \in [n]} \varphi_1^*(z_j(t)). \tag{57}$$

(Note that by Lemma D.1, $a \geqslant \min_{j \in [n]} \varphi_1^*(z_j(0))$ and $b \leqslant \max_{j \in [n]} \varphi_1^*(z_j(0))$.) For $c \in \{0, a, b\}$, we define the candidate limiting hyperplanes for $z_i(t)$:

$$H_c := \{x \in \mathbb{R}^d \colon \varphi_1^*(x) = c\}.$$

We show that $z_i(t)$ converges either to $H_0$, to $H_a$ or to $H_b$. If $a = b = 0$, then according to (57) all particles converge to $H_0$ and there is nothing left to prove. We now distinguish two scenarios:

(i) either for any $\varepsilon > 0$, $|\varphi_1^*(z_i(t))| \leqslant \varepsilon$ for $t$ large enough—in which case, we deduce that $z_i(t)$ converges toward $H_0$ as $t \to +\infty$—,

(ii) or $|\varphi_1^*(z_i(t_k))| > \varepsilon_0$ for some $\varepsilon_0 > 0$ and for some sequence of positive times $\{t_k\}_{k=1}^{+\infty}$ with $t_k \to +\infty$.

Since case (i) is straightforward, let us handle case (ii). Without loss of generality, we can extract a subsequence of times (which we do not relabel, for simplicity of notation) along which

$$\varphi_1^*(z_i(t_k)) > \varepsilon_0. \tag{58}$$

Let $\varepsilon \in (0, \varepsilon_0]$ be fixed and to be chosen later. We set

$$w_j(t) := \left\langle Qe^{tV} z_i(t), Ke^{tV} z_j(t) \right\rangle,$$

so that

$$\frac{1}{\lambda_1} \frac{\mathrm{d}}{\mathrm{d}t} \varphi_1^*(z_i(t)) = \sum_{j=1}^{n} \frac{e^{w_j(t)}}{\sum_{k=1}^{n} e^{w_k(t)}} \left( \varphi_1^*(z_j(t)) - \varphi_1^*(z_i(t)) \right). \tag{59}$$

We look to obtain a lower bound for the right-hand side in the above identity. Let us use the shorthand

$$c_{k\ell} := \langle Q\varphi_k, K\varphi_\ell \rangle$$

for $k, \ell \in [d]$. By assumption, $c_{11} > 0$. We have $\varphi_k^*(e^{tV} z_i(t)) = e^{t\lambda_k} \varphi_k^*(z_i(t))$ and the following spectral expansion holds:

$$e^{tV} z_i(t) = \sum_{k=1}^{d} e^{t\lambda_k} \varphi_k^*(z_i(t))\varphi_k.$$

Using this fact, as well as Lemma D.3, we gather that

$$
\begin{aligned}
\left| w_j(t) - c_{11} e^{2\lambda_1 t} \varphi_1^*(z_i(t))\varphi_1^*(z_j(t)) \right| &= \left| \sum_{(k,\ell)\neq(1,1)} c_{k\ell}\varphi_k^*(e^{tV} z_i(t))\varphi_\ell^* \left( e^{tV} z_j(t) \right) \right| \\
&\leqslant \sum_{(k,\ell)\neq(1,1)} |c_{k\ell}| \left| \varphi_k^* \left( e^{tV} z_i(t) \right) \right| \left| \varphi_\ell^* \left( e^{tV} z_j(t) \right) \right| \\
&\leqslant C^2 \|Q^\top K\|_{\mathrm{op}} \sum_{(k,\ell)\neq(1,1)} e^{(|\lambda_k|+|\lambda_\ell|)t} \\
&\leqslant \underbrace{C^2 \|Q^\top K\|_{\mathrm{op}}(d-1)^2}_{=:C'} e^{(\lambda_1+|\lambda_2|)t} \tag{60}
\end{aligned}
$$

holds for all $t \geqslant 0$ and $j \in [n]$. Now since $\lambda_1 > 0$, Lemma D.1 implies that for any $t \geqslant 0$ there exists an index $i_0(t) \in [n]$ such that

$$\varphi_1^*(z_{i_0(t)}(t)) \geqslant b. \tag{61}$$

With $j_0(t) \in \arg\max_{j\in[n]} w_j(t)$, using (60) and (61) we see that

$$w_{j_0(t)}(t) \geqslant w_{i_0(t)}(t) \geqslant c_{11}\varphi_1^*(z_i(t))be^{2\lambda_1 t} - C'e^{(\lambda_1+|\lambda_2|)t}. \tag{62}$$

Now for any $t$ within the sequence $\{t_k\}_{k=1}^{+\infty}$, combining the first inequality in (62) with the fact that $c_{11} > 0$, (58) and (60), we deduce that

$$\varphi_1^*(z_{j_0(t)}(t)) - \varphi_1^*(z_{i_0(t)}(t)) \geqslant -\frac{2C'}{c_{11}\varepsilon}e^{-(\lambda_1-|\lambda_2|)t}. \tag{63}$$

As $\lambda_1 > |\lambda_2|$, for $t$ large enough, we find that we can lower bound the above expression by $-\frac{\varepsilon}{4}$. We now define the set of indices

$$N(t) := \{j \in [n] \colon \varphi_1^*(z_i(t)) - \varphi_1^*(z_j(t)) \geqslant 0\}.$$

Take $t$ within the sequence $\{t_k\}_{k=1}^{+\infty}$ such that $\varphi_1^*(z_i(t)) \leqslant b - \varepsilon$ and large enough so that (63) is lower bounded by $-\frac{\varepsilon}{4}$ (if such a $t$ does not exist, we immediately conclude that $\varphi_1^*(z_i(t)) \to b$ as $t \to +\infty$). Using (61) and the subsequent derivations, we deduce that

$$\varphi_1^*(z_{j_0(t)}(t)) - \varphi_1^*(z_i(t)) \geqslant \frac{3\varepsilon}{4},$$

and since $\varphi_1^*(z_j(t)) - \varphi_1^*(z_i(t)) \geqslant 0$ for $j \notin N(t)$, we expand in (59) to get

$$\frac{1}{\lambda_1} \frac{\mathrm{d}}{\mathrm{d}t} \varphi_1^*(z_i(t)) \geqslant \frac{e^{w_{j_0(t)}(t)}}{\sum_{k=1}^n e^{w_k(t)}} \frac{3\varepsilon}{4} + \sum_{j \in N(t)} \frac{e^{w_j(t)}}{\sum_{k=1}^n e^{w_k(t)}} \left(\varphi_1^*(z_j(t)) - \varphi_1^*(z_i(t))\right). \tag{64}$$

On another hand, for $j \in N(t)$, we may use (60) to find

$$w_j(t) \leqslant c_{11} \varphi_1^*(z_i(t))^2 e^{2\lambda_1 t} + C' e^{(\lambda_1 + |\lambda_2|)t}. \tag{65}$$

We set

$$C_0 := \max_{j \in [n]} \varphi_1^*(z_j(0)) - \min_{j \in [n]} \varphi_1^*(z_j(0)).$$

Using the monotonicity properties from Lemma D.1, as well as (65) in (64), we obtain

$$\frac{1}{\lambda_1} \frac{\mathrm{d}}{\mathrm{d}t} \varphi_1^*(z_i(t)) \geqslant \frac{3\varepsilon}{4n} - C_0 n \frac{\exp\left(c_{11}\varphi_1^*(z_i(t))^2 e^{2\lambda_1 t} + C' e^{(\lambda_1 + |\lambda_2|)t}\right)}{\exp\left(c_{11}\varphi_1^*(z_i(t)) b e^{2\lambda_1 t} - C' e^{(\lambda_1 + |\lambda_2|)t}\right)}.$$

Given our choice of $t$, we have $\varphi_1^*(z_i(t))^2 - b\varphi_1^*(z_i(t)) \leqslant -\varepsilon(b - \varepsilon)$, so, we conclude from the inequality just above that

$$\frac{1}{\lambda_1} \frac{\mathrm{d}}{\mathrm{d}t} \varphi_1^*(z_i(t)) \geqslant \frac{3\varepsilon}{4n} - C_0 n \exp\left(-c_{11}\varepsilon(b - \varepsilon)e^{2\lambda_1 t} + 2C' e^{(\lambda_1 + |\lambda_2|)t}\right). \tag{66}$$

Since $\lambda_1 > |\lambda_2|$, it follows from (66) that there exists $T > 0$ such that for any $t$ within the sequence $\{t_k\}_{k=1}^{+\infty}$ for which $t \geqslant T$ and $\varphi_1^*(z_i(t)) \in [\varepsilon, b - \varepsilon]$, there holds

$$\frac{\mathrm{d}}{\mathrm{d}t} \varphi_1^*(z_i(t)) \geqslant \frac{\lambda_1 \varepsilon}{2n}.$$

This shows the existence of a larger time horizon $T' > T$ such that $\varphi_1^*(z_i(t)) \geqslant b - \varepsilon$ whenever $t \geqslant T'$. And since $\varepsilon$ can be taken arbitrarily small, we deduce that $\varphi_1^*(z_i(t))$ converges toward $b$, namely that $z_i(t)$ converges toward $H_b$, as $t \to +\infty$.

Arguing in the same way as above, and assuming without loss of generality that $a < 0$, we may find that all indices $i \in [n]$ for which $\varphi_1^*(z_i(t_k)) \leqslant -\varepsilon_0$ for some $\varepsilon_0 > 0$ and some sequence $t_k \to +\infty$, the particle $z_i(t)$ converges toward $H_a$ as $t \to +\infty$. This concludes the proof. $\square$

## D.3 Remarks

**Remark 7.** *Theorem 4.1 establishes the convergence of $\varphi_1^*(z_i(t))$ for any $i \in [n]$ as $t \to +\infty$, but does not preclude the fact that $\|z_i(t)\|$ may diverge toward $+\infty$ (along the hyperplane) as $t \to +\infty$. This is indeed expected (and observed numerically—see Fig. 6) when $V$ has some negative eigenvalues. We also note that when all the eigenvalues of $V$ are non-negative, Corollary D.2 shows that all the $z_i(t)$ remain bounded.*

**Remark 8** (The case where $V$ is not diagonalizable). *If $V$ is not assumed to be diagonalizable, Lemma D.3 (or, at least the proof thereof) requires some modifications. Let $\delta := \lambda_1 - |\lambda_2| > 0$. Let $\varepsilon > 0$ be fixed and to be chosen later. We decompose $V$ in Jordan blocks, and we consider*

$$\mathbb{C}^d = \bigoplus_{k=1}^m \mathscr{F}_k, \tag{67}$$

*where $\mathscr{F}_k$ is the span of the Jordan chain corresponding to the $k$-th Jordan block. By a slight abuse of notation (solely for the purpose of this remark), we denote by $\lambda_k$ the eigenvalue associated to the $k$-th Jordan block. We recall that we can choose a basis $(\varphi_{k,1}, \ldots, \varphi_{k,j_k})$ of each $\mathscr{F}_k$ in a way that $V_{|\mathscr{F}_k}$ reads in this basis as*[7]

$$\begin{bmatrix} \lambda_k & \varepsilon & & \\ & \ddots & \ddots & \\ & & \ddots & \varepsilon \\ & & & \lambda_k \end{bmatrix}. \tag{68}$$

---

[7]Recall that Jordan blocks are commonly written with a $+1$ in the superdiagonal. This can be replaced by any non-zero complex scalar as done here—see [HJ12, Chapter 3, Corollary 3.1.21].

*We observe that if $\varepsilon$ is chosen sufficiently small (depending only on $\delta$), Lemma D.3 may be replaced by the following estimate in each $\mathcal{F}_k$:*

$$\exists C > 0, \ \forall t \geqslant 0, \ \forall i \in [n], \qquad \left\| \pi_{\mathcal{F}_k} \left( e^{tV} z_i(t) \right) \right\| \leqslant C e^{(|\lambda_k| + \delta)t}. \tag{69}$$

*Here, $\pi_{\mathcal{F}_k}$ denotes the orthogonal projection onto $\mathcal{F}_k$. To prove estimate (69), we follow the proof of Lemma D.3, with $\frac{\mathrm{d}}{\mathrm{d}t} \|\pi_{\mathcal{F}_k}(x_i(t))\|^2$ playing the role of $\frac{\mathrm{d}}{\mathrm{d}t} |\varphi_k^*(x_i(t))|^2$. The key observation is that combining (67) and (68) we obtain*

$$\|\pi_{\mathcal{F}_k}(V x_i(t))\| \leqslant (|\lambda_k| + \delta) \|\pi_{\mathcal{F}_k}(x_i(t))\|,$$

*provided $\varepsilon$ is chosen sufficiently small. Then (69) follows as in Lemma D.3.*

*With (67) at hand, the proof of Theorem 4.1 carries through, under the impactless modification that $Ce^{(\lambda_1 + |\lambda_2| + \delta)t}$ replaces (60) (and subsequent estimates are modified in the same way).*

# E   A mix of hyperplanes and convex polytopes: proof of Theorem 5.1

In this section, we establish the proof for Theorem 5.1. Since the proof is essentially a combination of the proofs of Theorems 4.1 and C.1, we may occasionally skip certain details and refer to the proofs of these two results. As done throughout this work, we set

$$A := (Q^\top K)^{\frac{1}{2}}.$$

We denote by $\pi_{\mathcal{F}} : \mathbb{R}^d \to \mathcal{F}$ the projection onto $\mathcal{F}$ parallel to $\mathcal{G}$, and by $\pi_{\mathcal{G}} : \mathbb{R}^d \to \mathcal{G}$ the projection onto $\mathcal{G}$ parallel to $\mathcal{F}$. The set $\pi_{\mathcal{F}}(\mathrm{conv}(\{z_i(t)\}_{i \in [n]}))$ is a convex subset of $\mathcal{F}$ which is non-increasing with respect to $t$ (the proof of this fact is identical to that of Proposition C.2). It therefore converges toward some convex polytope $\mathcal{K}$ as $t \to +\infty$.

Fix $i \in [n]$. We have

$$
\begin{aligned}
\pi_{\mathcal{F}}(\dot{z}_i(t)) &= \sum_{j=1}^n \left( \frac{e^{\left\langle A e^{tV} z_i(t), A e^{tV} z_j(t) \right\rangle}}{\sum_{k=1}^n e^{\left\langle A e^{tV} z_i(t), A e^{tV} z_k(t) \right\rangle}} \right) \pi_{\mathcal{F}}(V(z_j(t) - z_i(t))) \\
&= \sum_{j=1}^n \left( \frac{e^{\left\langle A e^{tV} z_i(t), A e^{tV} (z_j(t) - z_i(t)) \right\rangle}}{\sum_{k=1}^n e^{\left\langle A e^{tV} z_i(t), A e^{tV} (z_k(t) - z_i(t)) \right\rangle}} \right) \pi_{\mathcal{F}}(V(z_j(t) - z_i(t))).
\end{aligned}
$$

From this point on, we follow the proof of Theorem C.1, and we solely highlight the changes compared to the original proof. Roughly speaking, this new proof amounts to adding projections $\pi_{\mathcal{F}}$ at several places. We denote by $\mathcal{S} \subset \mathcal{F}$ the set of points $w \in \mathcal{K}$ such that

$$\|\pi_{\mathcal{F}}(Aw)\|^2 = \max_{j \in [m]} \left\langle \pi_{\mathcal{F}}(Aw), \pi_{\mathcal{F}}(Av_j) \right\rangle.$$

The fact that $\mathcal{S} \subset \partial \mathcal{K}$ and that $\mathcal{S}$ has finite cardinality is proved precisely as Claim 1 (in the proof of Theorem C.1), simply by replacing all occurrences of $A\cdot$ by $\pi_{\mathcal{F}}(A\cdot)$. Once again, $\mathcal{S}_\delta$ denotes the set of all points in $\mathcal{K}$ at distance $\leqslant \delta$ to some point of $\mathcal{S}$.

Step 2 in the proof of Theorem C.1 (i.e., (38)) is replaced by the following statement:

**Step 2':**   There exists a constant $\gamma = \gamma(\mathcal{K}) > 0$ (depending only on the geometry of $\mathcal{K}$) such that for any $\delta \in (0, \delta_0]$, there exists $T = T(\delta) > 0$ such that if $t \geqslant T$ and $\pi_{\mathcal{F}}(z_i(t)) \notin \mathcal{S}_\delta$, then

$$\frac{\mathrm{d}}{\mathrm{d}t} \|\pi_{\mathcal{F}}(A z_i(t))\|^2 \geqslant \gamma \delta.$$

We now proceed in proving this statement.

*Proof of Step 2'.*   We set

$$a_j(t) := \left\langle \pi_{\mathcal{F}}(A z_i(t)), \pi_{\mathcal{F}}(A(z_j(t) - z_i(t))) \right\rangle$$

and

$$r_j(t) := \left\langle A e^{tV} z_i(t), A e^{tV} (z_j(t) - z_i(t)) \right\rangle - a_j(t) e^{2\lambda_1 t}.$$

We find

$$\frac{1}{2}\frac{\mathrm{d}}{\mathrm{d}t}\|\pi_{\mathscr{F}}(Az_i(t))\|^2 = \langle \pi_{\mathscr{F}}(A\dot{z}_i(t)), \pi_{\mathscr{F}}(Az_i(t))\rangle$$

$$= \sum_{j=1}^{n}\left(\frac{e^{\langle Ae^{tV}z_i(t),\, Ae^{tV}z_j(t)\rangle}}{\sum_{k=1}^{n}e^{\langle Ae^{tV}z_i(t),\, Ae^{tV}z_k(t)\rangle}}\right)\langle \pi_{\mathscr{F}}(A(z_j(t)-z_i(t))), \pi_{\mathscr{F}}(Az_i(t))\rangle$$

$$= \sum_{j=1}^{n}\left(\frac{e^{\langle Ae^{tV}z_i(t),\, Ae^{tV}(z_j(t)-z_i(t))\rangle}}{\sum_{k=1}^{n}e^{\langle Ae^{tV}z_i(t),\, Ae^{tV}(z_k(t)-z_i(t))\rangle}}\right)\langle \pi_{\mathscr{F}}(A(z_j(t)-z_i(t))), \pi_{\mathscr{F}}(Az_i(t))\rangle$$

$$= \sum_{j=1}^{n}\underbrace{\left(\frac{e^{a_j(t)e^{2\lambda_1 t}+r_j(t)}}{\sum_{k=1}^{n}e^{a_k(t)e^{2\lambda_1 t}+r_k(t)}}\right)a_j(t)}_{=:b_j(t)}. \tag{70}$$

We now make use of the following adaptation of Claim 2.

**Claim 3.** *There exists some constant $\gamma' = \gamma'(\mathcal{K}) > 0$ depending only on the geometry of $\mathcal{K}$ such that the following holds. Fix $\delta \in (0, \delta_0]$. There exists $T = T(\delta) > 0$ such that if $t \geqslant T$ and $z_i(t) \notin \mathcal{S}_\delta \times \mathcal{G}$, then there exists $j \in [n]$ such that $a_j(t) \geqslant \gamma'\delta$.*

Compared to Step 2 in the proof of Theorem C.1, we now have to estimate the coefficients $r_j(t)$. To this end, setting $y_j(t) := Ae^{tV}z_j(t)$ for $j \in [n]$, we notice that $r_j(t) = P_1(t) + P_2(t) + P_3(t)$ where

$$P_1(t) = \langle \pi_{\mathscr{F}}(y_i(t)), \pi_{\mathscr{G}}(y_j(t)-y_i(t))\rangle,$$
$$P_2(t) = \langle \pi_{\mathscr{G}}(y_i(t)), \pi_{\mathscr{F}}(y_j(t)-y_i(t))\rangle,$$
$$P_3(t) = \langle \pi_{\mathscr{G}}(y_i(t)), \pi_{\mathscr{G}}(y_j(t)-y_i(t))\rangle.$$

By virtue of Lemma D.3 we have $|\pi_{\mathscr{F}}(y_j(t))| \leqslant Ce^{\lambda_1 t}$ and $|\pi_{\mathscr{G}}(y_j(t))| \leqslant Ce^{t|\lambda_2|}$ for any $t \geqslant 0$ (or $Ce^{t|\lambda_2|+\varepsilon}$ if $V_{|\mathscr{G}}$ is not diagonalizable—see Remark 8), hence

$$|r_j(t)| \leqslant Ce^{t(\lambda_1+|\lambda_2|)}. \tag{71}$$

Since $\pi_{\mathscr{F}}(z_j(t))$ is uniformly bounded in $t \in [0, +\infty)$ for any $j \in [n]$ due to Corollary C.3, we get $a_j(\cdot) \in L^\infty(0, +\infty)$. So, we may set

$$\kappa := \max_{j\in[n]}\sup_{t\geqslant 0}|a_j(t)|.$$

Let $t \geqslant 0$. We define

$$B(t) := \left\{j \in [n]: a_j(t)e^{2\lambda_1 t} + r_j(t) \geqslant 0\right\}.$$

Let $j_0(t) \in \arg\max_{j\in[n]}(a_j(t)e^{2\lambda_1 t} + r_j(t))$. Note that $j_0(t) \in B(t)$ since

$$a_{j_0}(t)e^{2\lambda_1 t} + r_{j_0}(t) \geqslant a_i(t)e^{2\lambda_1 t} + r_i(t) = 0.$$

We notice the following three properties:

- For $j = j_0(t)$, we have $b_{j_0(t)}(t) \geqslant \frac{a_{j_0(t)}(t)}{n}$ (recall the definition of $b_j$ in (70));

- for any $j \in B(t)\backslash\{j_0\}$, we have $b_j(t) \geqslant 0$;

- for any $j \notin B(t)$, we have

$$b_j(t) \geqslant -\kappa \exp\left(-a_{j_0}(t)e^{2\lambda_1 t} + Ce^{(\lambda_1+|\lambda_2|)t}\right).$$

Indeed, using the fact that $j \in B(t)$ and (71), we find

$$\frac{\exp\left(a_j(t)e^{2\lambda_1 t} + r_j(t)\right)}{\displaystyle\sum_{k=1}^{n}\exp\left(a_k(t)e^{2\lambda_1 t} + r_k(t)\right)} \leqslant \frac{1}{\displaystyle\sum_{k=1}^{n}\exp\left(a_k(t)e^{2\lambda_1 t} + r_k(t)\right)}$$

$$\leqslant \frac{1}{\exp\left(a_{j_0}(t)e^{2\lambda_1 t} + r_{j_0}(t)\right)}$$

$$\leqslant \exp\left(-a_{j_0}(t)e^{2\lambda_1 t} + Ce^{(\lambda_1+|\lambda_2|)t}\right).$$

Making use of these properties in (70) yields the desired lower bound—indeed, if $t$ is sufficiently large and $z_i(t) \notin \mathcal{S}_\delta \times \mathcal{G}$, we have $\{j \in [n] \colon a_j(t) \geqslant \gamma'\delta\} \neq \varnothing$ according to Claim 3, and so we deduce that

$$\frac{1}{2}\frac{\mathrm{d}}{\mathrm{d}t}\|Az_i(t)\|^2 \geqslant \frac{\gamma'\delta}{n} - \kappa n e^{-\gamma'\delta e^{2\lambda_1 t} + Ce^{(\lambda_1 + |\lambda_2|)t}}.$$

Taking $t$ possibly larger (and depending on $\delta$), we obtain the result of Step 2'. $\qquad\square$

Steps 3 and 4 in the proof of Theorem C.1 are essentially unchanged—we replace all the occurrences of $\|A \cdot \|$ by $\|\pi_{\mathscr{F}}(A\cdot)\|$ (for instance in (44) and (45)). Although $\|Az_i(t)\|$ may not be uniformly bounded in $t$, it is important to note that $\|\pi_{\mathscr{F}}(Az_i(t))\|$ is uniformly bounded. Similarly, while $\dot{z}_i(t) \notin L^\infty([0, +\infty))$, we do have $\|\frac{\mathrm{d}}{\mathrm{d}t}\pi_{\mathscr{F}}(z_i(\cdot))\|_{L^\infty([0,+\infty))} < +\infty$. The sets $\mathcal{S}_\delta$, $\mathscr{C}_k$ and $\mathscr{C}_k^r$ are replaced by $\mathcal{S}_\delta \times \mathcal{G}$, $\mathscr{C}_k \times \mathcal{G}$ and $\mathscr{C}_k^r \times \mathcal{G}$ respectively. The conclusion is that $\|\pi_{\mathscr{F}}(Az_i(t))\|^2$ has to increase by at least

$$\frac{\gamma\delta^{\frac{1}{2}}(\delta^{\frac{1}{4}} - \delta)}{\|\dot{z}_i\|_{L^\infty([0,+\infty))}} \geqslant \frac{\delta^{\frac{3}{4}}}{2\|\dot{z}_i\|_{L^\infty([0,+\infty))}} > 4R\|A\|_{\mathrm{op}}\delta$$

during a travel from $\mathscr{C}_k \times \mathcal{G}$ to the complement of $\mathscr{C}_k^{\frac{1}{4}} \times \mathcal{G}$. As in the proof of Theorem C.1 this implies that for any $i \in [n]$ there exists $s \in \mathcal{S}$ such that $z_i(t)$ remains at distance at most $\delta$ away from $\{s\} \times \mathcal{G}$. This being true for any $\delta > 0$, we obtain the desired result.

# F   Numerical experiments

## F.1   Setup

Unless indicated otherwise, all figures presented in this paper were generated by discretizing the underlying dynamics (either (1) or (4)) using a fourth order Runge-Kutta scheme with a step size of 0.1. All points in the initial sequence were drawn independently from the uniform distribution over the hypercube $[-5, 5]^d$. Random matrices (e.g., $Q, K, V$) have entries drawn independently from the uniform distribution on $[-1, 1]$. Codes and animated plots of all examples may be found online at

https://github.com/borjanG/2023-transformers.

We now present some experiments which motivate some conjectures and claims made in what precedes.

## F.2   Eigenvalues of ALBERT's value matrices

In Figure 10 we illustrate the eigenvalues of the value matrices $V_h$ for a couple of heads $h$ in a pre-trained ALBERT model. We focus on ALBERT-xlarge-v2 available online at https://huggingface.co/albert-xlarge-v2. This version uses 16 heads, with sequences of length $n = 256$ and tokens of dimension $d = 128$. While not all value matrices $V_h$ per head $h \in [16]$ satisfy the assumptions made in Section 4, we illustrate the eigenvalues of a couple of them which do.

## F.3   Experiments related to Theorem 2.1

We begin with the setup of Theorem 2.1, which we recall was proven to hold in the case $d = 1$. Herein we present a couple of examples (Figures 11 and 12) which elucidate the role that $d$ and $n$ appear to play in this fact.

Notably, as seen in Fig. 4, we believe that the conclusion of Theorem 2.1 could plausibly be extended to any $d > 1$, assuming $V > 0$.

## F.4   Illustrating Theorem 4.1 in $\mathbb{R}^3$

To precisely illustrate the appearance of at most three hyperplanes in the setting of Theorem 4.1, we gave an example in $\mathbb{R}^2$. We expand on this and provide a couple of toy examples in $\mathbb{R}^3$ for the purpose of visualization (we recall that these are toy models, as Transformers in practice are

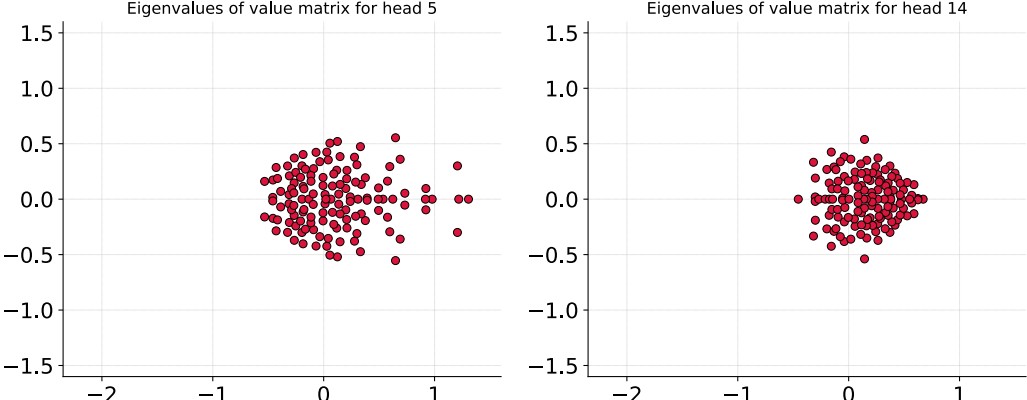

Figure 10: The eigenvalues of $V_5$ and $V_{14}$ in the pre-trained ALBERT satisfy the eigenvalue assumption made in Definition 1. Furthermore, the second assumption made in Definition 1 is satisfied by $(Q_5, K_5)$ and $(Q_{14}, K_{14})$ (the inner products evaluated along the eigenvector of norm 1 equal 1.3060 and 0.6719 respectively). In other words, the triples $(Q_h, K_h, V_h)$ corresponding to heads $h = 5$ and $h = 14$ in ALBERT satisfy all the assumptions made in the statement of Theorem 4.1.

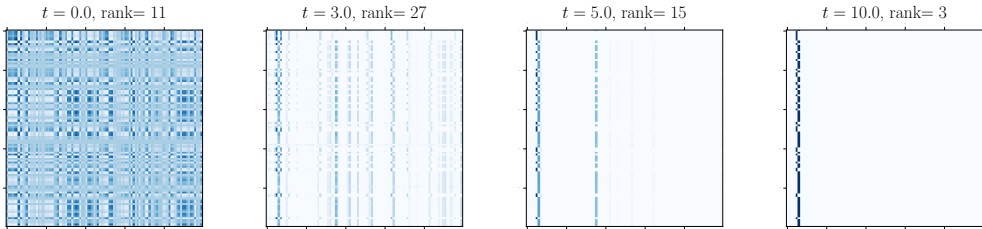

Figure 11: We expand on Fig. 3—for the same setup, consider $n = 100$. The sequence length $n$ does not appear to influence the rank of $P(t)$, which is expected since the rank of $P$ corresponds to the number of leaders.

high-dimensional), and namely focus in both examples on the case where the two latter eigenvalues are complex. In Fig. 14, we see the effect of having eigenvalues with a negative real part, and the complementary case is illustrated in Fig. 13.

## F.5 Complementing Figure 7

In Figure 7, we illustrate the appearance of clustering in high-dimension (the ALBERT setup: $n = 256$ and $d = 128$) for generic random matrices $(Q, K, V)$. The value matrix $V$ in question has 65 positive eigenvalues, and we show the conjectured convergence of the 65 coordinates along the corresponding eigenvectors to one of possibly 3 (generically 2) real scalars. In Figure 15, we complement this illustration by showing the possible oscillatory and divergent behavior of the remaining coordinates.

## F.6 Beyond $Q^\top K > 0$ in Theorems 3.1 and 5.1

As seen throughout all the presented proofs, assumptions on the value matrix $V$ are significantly more rigid than assumptions on the matrices $Q$ and $K$. For instance, should the eigenvalue $\lambda$ with the largest real part of $V$ be negative, all rescaled tokens will diverge to infinity. Should $\lambda$ be complex, we do not expect any clustering to occur (for the rescaled tokens). Yet, none of the conclusions of Theorems 3.1 or 5.1 seem to change for generic choices of $Q^\top K$. This is illustrated in Figures 16 and 17 respectively.

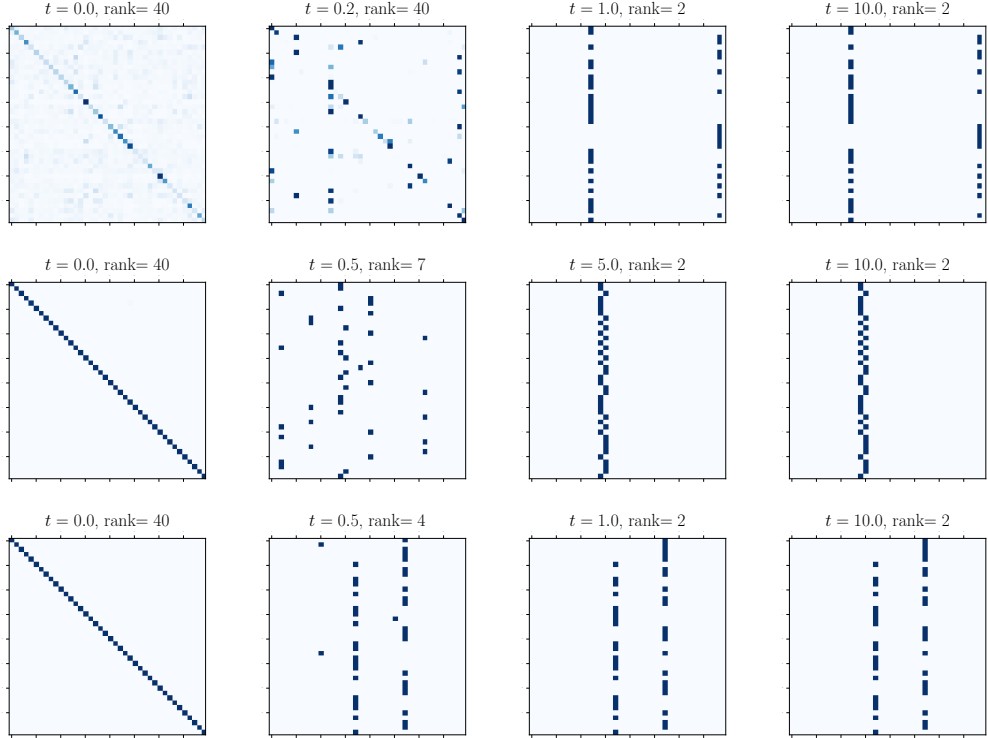

Figure 12: We consider $n = 40$, $Q = K = I_d$ and a random matrix $V > 0$ in dimensions $d = 10$ (first row), $d = 40$ (second row), and $d = 80$ (third row). The conclusion of Theorem 2.1 appears to transfer to the higher dimensional case, and this would actually follow from Conjecture 4.2 (should it hold).

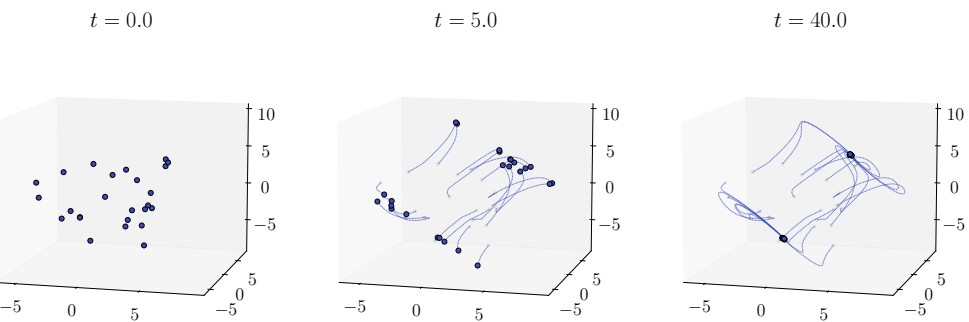

Figure 13: We consider $n = 25$, $Q = K = I_d$, and $V$ a random matrix with positive entries and eigenvalues $\{1, 0.1 + 0.08i, 1 - 0.08i\}$. The pair of complex eigenvalues have a positive real part. We not only see convergence to one of two hyperplanes determined by the direction $\varphi_1 = (0.38, 0.8, 0.47)$, but in fact, the particles appear to collapse to two points. In other words, the "hyperplanes" are of codimension 3, which is in line with Conjecture 4.2.

## F.7 Beyond pure self-attention: adding a feed-forward layer

Practical implementations of the Transformer architecture combine the self-attention mechanism with a feed-forward neural network. While extending the mathematical analysis from this paper to such a broader setting would be challenging, we can offer some numerical insights into the expected outcomes.

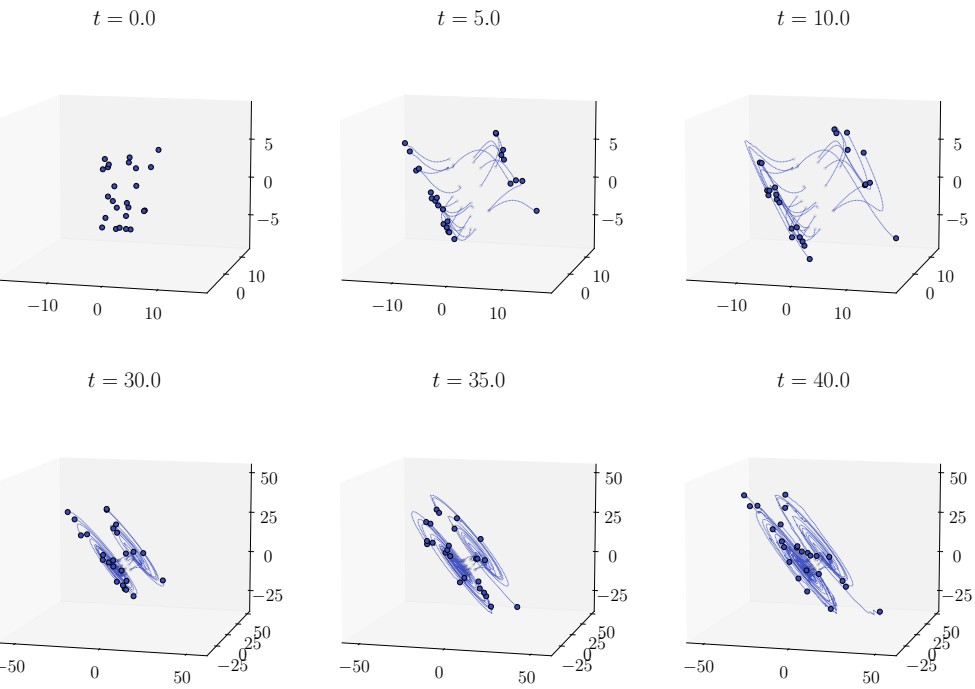

Figure 14: We consider $n = 25$, $Q = K = I_d$, and $V$ a random matrix with positive entries and eigenvalues $\{1, -0.05 + 0.25i, -0.05 - 0.25i\}$. The pair of complex eigenvalues have a negative real part, which entails the rotation of the particles. We see that the particles rotate within a couple of 2-dimensional hyperplanes determined by $\varphi_1 = (-0.3, -0.8, -0.45)$, as implied by Theorem 4.1.

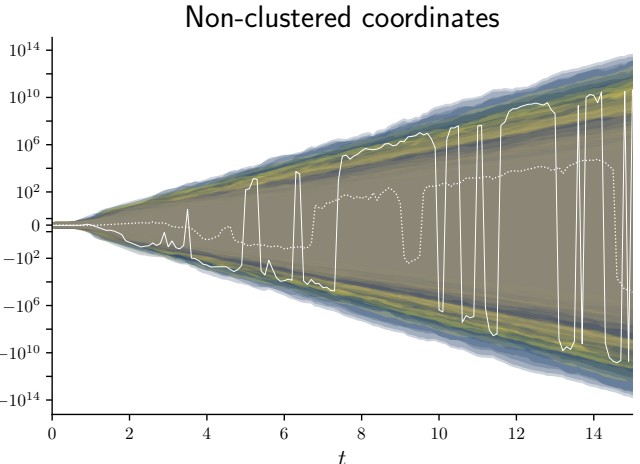

Figure 15: We complement Figure 7 and plot the variance of the set $\{\varphi_j^*(z_i(t)) : i \in [n]\}$ of all coordinates $j$ corresponding to negative eigenvalues of $V$. We also show the mean along tokens of a couple of coordinates (white lines). Coordinates diverge rapidly to $\pm\infty$ over time $t$; $y$-axis is in log scale.

The feed-forward neural network which can be adjoined to the Transformer dynamics in one of two ways. The first way consists in running the pure self-attention dynamics up to time $t \leqslant T$ (or equivalently, for $O(T)$ layers), and then applying a pure feed-forward neural network to the concatenated vector of clustered features at time $T$. This amounts to seeing the feed-forward network as a map from $\mathbb{R}^{nd}$ to $\mathbb{R}^m$ (for some $m \geqslant 1$), which can be studied independently with existing

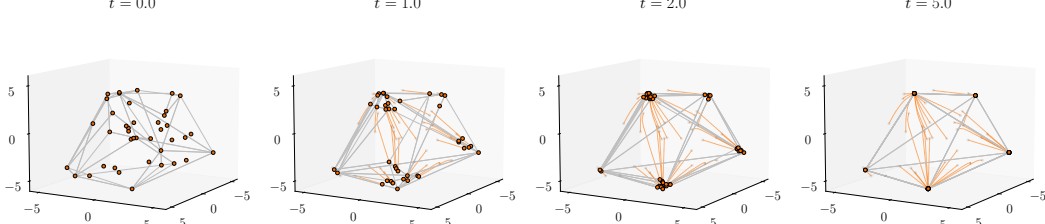

Figure 16: Here, $V = I_d$, while $Q^\top K$ violates the PSD assumption–it is a random matrix (with entries drawn from the uniform distribution on $[-1, 1]$). Nonetheless, the clustering pattern entailed by Theorem 3.1 persists.

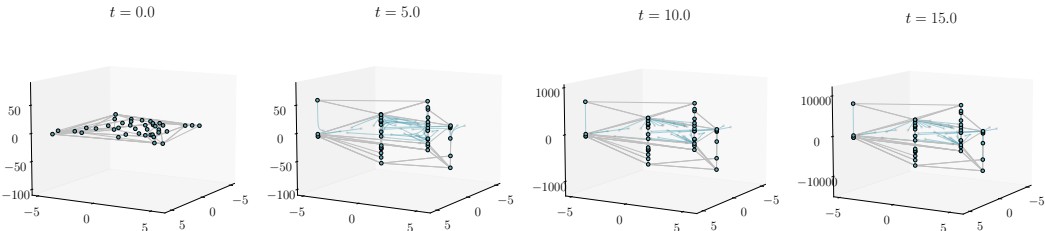

Figure 17: Here, $V$ is paranormal, while $Q^\top K$ violates the PSD assumption–it is a random matrix (with entries drawn from the uniform distribution on $[-1, 1]$). Nonetheless, the clustering pattern entailed by Theorem 5.1 persists.

theory. The second way consists in using both the self-attention and feed-forward mechanisms in parallel at every layer $t$. In this case, clustering in the exact sense of Theorems 3.1 and Theorems 5.1 would be difficult to anticipate since the weights of the feed-forward network play the role of a value matrix $V$ (as they can be absorbed within $V$), and the conclusions of these theorems strongly depend on the identity-like structure.

In Figure 18, we focus on the second of the above-discussed examples, and illustrate a possible generalization of Theorem 4.1 to this setup. For simplicity, we focus on a 2-layer neural network: we apply a component-wise nonlinear activation function $\sigma$ (either the ReLU or $\tanh$) to the self-attention dynamics, and then multiply by a weight matrix $W \in \mathbb{R}^{d \times d}$. Namely, we consider

$$\dot{z}_i(t) = W\sigma \left( V \sum_{j=1}^{n} \left( \frac{e^{\langle Qe^{tV}z_i(t), Ke^{tV}z_j(t)\rangle}}{\sum_{k=1}^{n} e^{\langle Qe^{tV}z_i(t), Ke^{tV}z_k(t)\rangle}} \right) (z_j(t) - z_i(t)) \right) \tag{72}$$

for $i \in [n]$ and $t \geqslant 0$. A bias vector $b \in \mathbb{R}^d$ (whether inside or outside the activation function) can also be included to allow for translations. The clustering property appears to persist, the pattern depending on the weight matrix $W$ and on the activation function $\sigma$. We leave this problem open to further investigation.

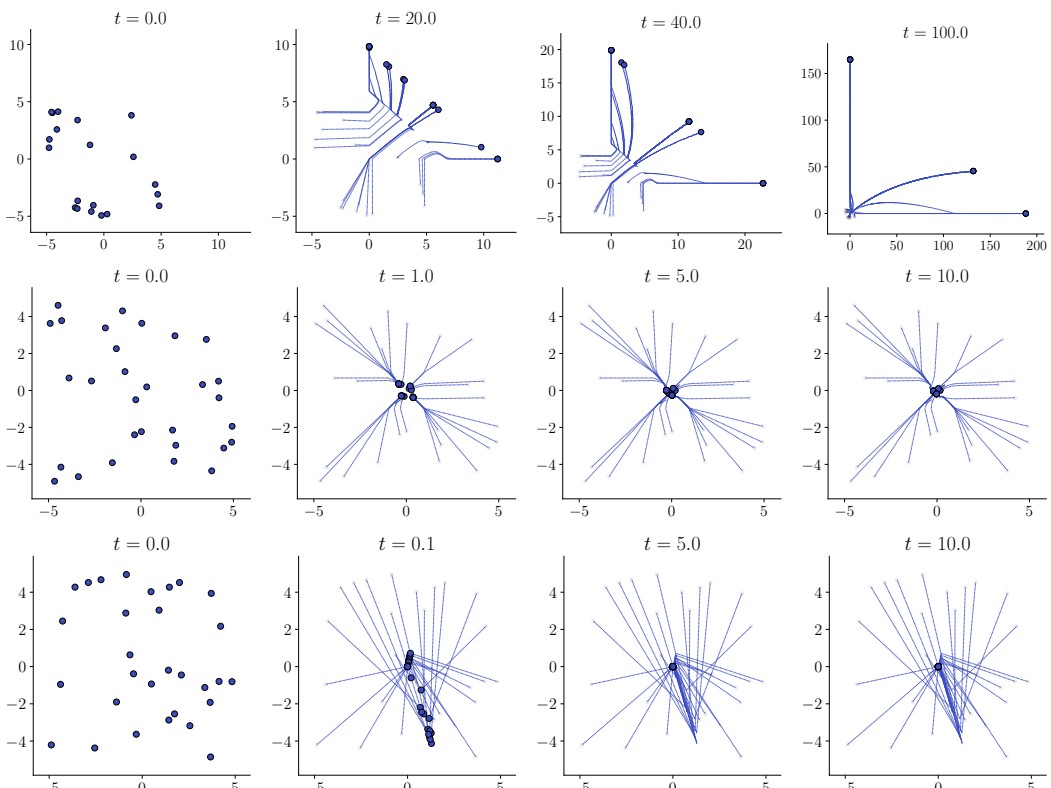

Figure 18: The setup of Theorem 4.1 with a 2-layer neural network appended to the dynamics (i.e., (72)). Top: $\sigma = \text{ReLU}$ with $W = I_d$. Middle: $\sigma = \tanh$ with $W = I_d$. Bottom: $\sigma = \text{ReLU}$ with $W$ being a random matrix. In the first row, we see that the particles first evolve as to reach the upper right quadrant $(\mathbb{R}_{>0})^d$ (due to the ReLU). Once they reach it, every particle eventually follows one of three hyperplanes determined by the spectrum of $V$ and the projection onto $(\mathbb{R}_{>0})^d$. In the other two cases, all particles appear to collapse to $0$.