# OpenReview forum: "The emergence of clusters in self-attention dynamics"
_NeurIPS.cc/2023/Conference — NeurIPS 2023 poster_

### Official Review · Reviewer_P5zn · 2023-06-28

**Soundness:** 3 good
**Presentation:** 4 excellent
**Contribution:** 3 good
**Rating:** 7
**Confidence:** 4

**Summary:**

For $Q,K,V$ fixed and different structures of $V$, analyze the distribution of $x(t)$ as $t\to\infty$, where tokens are seen as particles and the self-attention mechanism is seen as particle interaction, i.e. as a McKean-Vlasov SDE. The conclusion is as $t\to\infty$, i.e. going through the layers, $x(t)$ converges to a clustered configuration, and the attention matrix $P$ becomes low-rank, i.e. tokens depend on few tokens.

**Strengths:**

Following [LLH+20] and [SABP22], the authors interpret self-attention and transformers through the lens of interacting particle systems/ McKean-Vlasov SDEs. This is perhaps the first paper to show via the above interpretation the emergence of clusters/ leaders in transformers.

**Weaknesses:**

1. Please state within the main paper the relationship between the clustered configuration and the initial configuration.

2. The result in [SABP22] seems to be the opposite of clustering. Please let me know if I misunderstood. Otherwise please comment on the clustering/ non-clustering effects and use-cases for downstream applications.

3. Please remark on why "transformer dynamics present unique mathematical challenges that cannot be addressed using the tools developed for these more primitive models. (lines 147-148)."

4. Please strengthen the contribution with layer normalization.

**Questions:**

1. Please comment on the effect of different embeddings. (since tokens converge to the boundary and vertices of a convex polytope, and the polytope depends on the initial distribution of tokens and the embedding).

2. Please give some intuition on what the vertices mean. (are vertices more descriptive since everything inside is a convex combination of the vertices? how is the final distribution on the vertices related to the initial distribution? e.g. is the mean unchanged? will it speed up training/ execution if a neural network was designed to give the convex polytope and the weights on the vertices instead of repeated iteration of self-attention?)

3. Please comment on the downstream effects of "linearly separable representation of tokens (line 266-267)". Perhaps $x(t)$ as $t\to\infty$ can be used as new embeddings.

4. Read the proof for -V in the appendix. Would it also help to think of clustering dynamics in terms of potential and thus flipping the sign of $V$ flips the potential landscape?

5. Please comment on the shrinking convex hull/ polytope (since it is finite) (Appendix C.1.1). What about infinite number of tokens?

6. If the tokens cluster, why would depending on one token (low rank attention matrix $P$) within a cluster be more likely than depending on another token within the cluster? Or is it because $P$ is low rank and thus tokens cluster? Or a combination of both effects?

7. Please comment on the effects of skip and non-skip connections [DCL21].

**Limitations:**

Authors addressed the limitation of fixed $Q,K,V$, simple structures for $V$. Authors mentioned multi-head attention as future work.

---

> ### Author Rebuttal · Authors · 2023-08-09
>
> We thank the reviewer for their feedback, and respond to each point individually below.
>
> **Weaknesses.**
>
> 1. We do not believe there necessarily is a relation, as there is no easy way to predict the clustered configuration from the initial one beyond simply running the dynamics. Moreover, the number of clusters is often not equal to the number of vertices of the polytope given by the convex hull of the initial sequence.
>
> 2.  In our opinion the results in [SABP22] are of a slightly different nature. The authors show, by virtue of an additional bandwidth hyperparameter $\varepsilon$ in the exponential, that the continuity equation for the transformer dynamics (replacing self-attention by the Sinkform kernel) can be seen as an approximation for the heat equation ($V=I_d$) when the bandwidth goes to $0$. In our case, the bandwith is fixed and equal to $1$. We believe that in practical applications, the self-attention mechanism serves a dimension-reduction purpose. Our clustering theorems can be seen as an indicator of this thesis.
>
> 3. The related models which we mention (Vicsek, Cucker-Smale, and so on) have significantly more basic interactions--often involving symmetric interactions and a radial dependence--, thus rendering the pre-developed mathematical tools inapplicable to our setting.
>
> 4. Layer normalization amounts to considering the dynamics on the unit sphere on $\mathbb{R}^d$. In ongoing work, we have observed, both theoretically and numerically, that clustering persists in a sense much alike the results presented in the present paper. The proof techniques are however different due to the intrinsic symmetries entailed by the sphere, and thus involve significant additional developments.
>
> ---
>
> **Questions.**
>
> 1.  It is not clear how the limiting convex polytope depends on the initial distribution of tokens even without the consideration of different embeddings.
> We believe that this could be an exciting avenue for future research.
>
> 2. We do not believe that the vertices of the limiting polytope have a clear meaning as they are not straightforwardly related to the initial distribution, as alluded to in previous answers. Slight changes in the initial distribution can change the limiting polytope. The mean is not preserved.  We thank the Reviewer for the last observation which is very compelling--indeed, should one manage to gather a clearer understanding of how the limiting polytope depends on the initial distribution and on $Q^\top K$, the repeated iteration of self-attention could perhaps be replaced in training and this could be seen as a possible convex relaxation.
>
> 3. This is a possibility, we thank the Reviewer for this observation. What we had in mind was to potentially use transformers as a plain clustering algorithm in the spirit of the mean-shift algorithm, or perhaps for binary classification tasks, by considering the inputs of the entire dataset as an input sequence.
>
> 4. Yes, there is a natural potential, or more precisely a natural Lyapunov function $\mathscr{L}$, which appears in the proof of Lemma C.8. The function $\mathscr{L}$ is non-increasing when $V=-I_d$, which corroborates the global decay of the norms of the points toward $0$ (Theorem C.5). When $V=I_d$, the function $\mathscr{L}$ is non-decreasing along the flow of the original dynamics (15), and indeed all points tend to diverge toward infinity when evolved through this dynamics, as a corollary of Theorem 3.1 after removing the rescaling.
>
> 5. With an infinite number of tokens, there is also shrinking of the convex hull as in Proposition C.2. The limiting shape is convex, but it is not necessarily a convex polytope, it may be for instance a sphere.
>
> 6. There is empirically one leader in each cluster, but it is difficult to know which one it is without running the full dynamics. A limiting low-rank self-attention matrix allows to identify not only clusters, but also leaders within the clusters by reading the rows and columns. When we establish clustering, except for the case $d=1$ we are not able to prove convergence toward a low-rank self-attention matrix, although this is what we observe in practice.
>
> 7. The outcome of [DCL21] is that without skip connections, the output of the transformers dynamics converges doubly exponentially to a rank-1 matrix. Therefore, [DCL21] highlights the necessity of skip-connections for transformers to be efficient.
> [DCL21] proves clustering toward a unique cluster for transformers without skip-connections; our work proves that clustering occurs for transformers with skip-connection, which are the ones used in practice.

---

> > ### Comment · Reviewer_P5zn · 2023-08-14
> >
> > I would like to thank the authors for answering my many questions. I am positive towards this paper and am raising my score to 7. I have read all the other reviews and rebuttals. May I ask what is the ``probabilistic'' structure of the self-attention mechanism in the rebuttal to reviewer Qc4C? Thanks!

---

> > > ### Author Response · Authors · 2023-08-15
> > >
> > > We thank the Reviewer again for their feedback. With regard to the question - what was meant by "probabilistic" is the fact that the dynamics are given by a convex combination of the tokens, namely, the self-attention coefficients are in $[0,1]$ and add up to $1$.
> > > We hope this clarifies our comment.

---

> > > > ### Comment · Reviewer_P5zn · 2023-08-15
> > > >
> > > > I see, thanks!

---

### Official Review · Reviewer_8epq · 2023-07-05

**Soundness:** 2 fair
**Presentation:** 2 fair
**Contribution:** 2 fair
**Rating:** 6
**Confidence:** 2

**Summary:**

This paper analyses the self-attention mechanism in (trained) Transformers under the lens of dynamical systems. The authors focus on a bare-bone self-attention architecture without the bells and whistles of standard Transformers (e.g. multi-head attention, layer norm) and assume time-independent weights, i.e. shared weights across "layers".

They start their analysis by studying simple self-attention architectures, including 1d settings, and progressively move to more complex and realistic settings, including value matrices with simple and positive leading-eigenvalue.
Across all the scenarios studied, the authors observe that in the limit, tokens cluster towards a few objects, such as vertices of a polytope or hyperplanes. Thus, the authors mathematically confirm that a small number of leaders drive the transformer dynamics.

**Strengths:**

This work focuses on the mathematical under-pinnings of self-attention and transformer architectures, a research direction that remains unexplored, yet is much needed, given the importance of transformer architectures in modern deep learning. Such a theoretical foundation is original and significant for the community.

The authors showcase the clustering behaviour of self-attention in progressively realistic scenarios. The main points and conclusions of this work are clear.

**Weaknesses:**

Overall, it is unclear whether the assumptions used in to prove theorems (e.g. good triples, $Q^T K \succ 0$) are realistic for standard Transformers.
Furthermore, the analysis is performed assuming only time-independent weights. While that design choice has been used in practice (e.g. ALBERTA), it is not the standard design choice for transformers. It is unclear whether the theorems introduced in this work hold for time-dependent weights.

The paper is very math-heavy for a general machine learning audience, and as such it is often hard to follow.

**Questions:**

1. In eq. 4, the authors introduce an exponential factor, which is "instrumental in the proofs of all results that follow". Would the theorems hold without the use of this factor? What are the practical implications of this additional factor?

2. Furthermore, according to the authors, this operation is a "mathematically justified surrogate for the layer normalization". Do the authors expect this rescaling operation to be useful in an actual self-attention implementation, either replacing layer norm or used along with it?

3. In Definition 2, the authors define what constitutes a good triple $(Q, K, V)$. How realistic is this assumption, and how robust is the theorem to the assumption holding approximately?

**Limitations:**

The authors should focus more on the limitations of the assumptions used for the theoretical results introduced in this work. It is unclear whether the theorems hold in practice, or whether they are based on very strong and unrealistic assumptions. See also weaknesses and limitations.

---

> ### Author Rebuttal · Authors · 2023-08-09
>
> We thank the reviewer for their feedback, and respond to each point individually below.
>
> **Weaknesses.**
>
> We have attempted to paint a more complete picture on the necessity of some assumptions made to facilitate the development of this theory through numerical experiments (Figures 1, 2, 3 of the PDF) and additional comments (e.g. the answer to Question 1 by Reviewer Qc4c).
>
> ---
>
> **Questions.**
>
> 1. The main effect, namely the emergence of leaders (visible in the self-attention matrix) remains true without the rescaling factor, see Theorem 2.1. Without the rescaling factor however, there is no clustering: this rescaling factor, in practice, has an effect of normalization on tokens, without which tokens typically diverge to $\pm\infty$.
>
> 2.  We indeed believe that the exponential rescaling can be used in actual implementations, as a surrogate to layer norm. (There is a slight difference however in the sense that layer norm amounts to having particles evolving on the unit sphere of $\mathbb{R}^d$, hence bounded in all directions.) It could potentially be used in conjunction with layer norm, but it is not totally clear to us what this would entail.
>
> 3. We recognize that point (i), namely $Q^\top K\succ 0$, appears stringent. This assumption is made since the proof follows the lines of that of Theorem 3.1 (which is conceived on elements using this condition).
> However, experiments indicate that (i) is not necessary for the conclusion of the theorem to hold.
> Point (ii) is a generalization of the first bullet point in Definition 1 (the latter is satisfied by some pre-trained value matrices of individual heads in ALBERT, see Figure 10 in Supplementary Material).
> Assumptions on $V$ are generally more sensitive to perturbations than assumptions on $(Q, K)$. Should the eigenvalue $\lambda$ with largest real part be negative, all rescaled particles will diverge to infinity. Should $\lambda$ be complex, we do not expect any clustering phenomenon. Yet none of the conclusions seem to change if $Q^\top K$ is taken arbitrary.

---

> > ### Comment · Reviewer_8epq · 2023-08-14
> > **Official Comment by Reviewer 8epq**
> >
> > I would like to thank the authors for their rebuttal; they have addressed my questions. I am still positive towards this paper and I think its contribution is significant. I am keeping my score to 6. I hope the authors incorporate some of the discussion in the camera-ready version of the paper.

---

### Official Review · Reviewer_U2EX · 2023-07-05

**Soundness:** 4 excellent
**Presentation:** 4 excellent
**Contribution:** 4 excellent
**Rating:** 7
**Confidence:** 4

**Summary:**

In this work, the authors develop a theoretical analysis of self-attention mechanism. In particular, the authors study the setting of a trained transformer  and their goal is to characterize the output of a deep transformer with multiple layers of self-attention. For simplicity authors focus on weight sharing and also do not use MLPs and multiple heads. The key results include:
i) the convergence guarantees of self-attention matrix to a low rank matrix.
ii) the characterization of convergence of initial inputs to final outputs as a function of query, key and value matrix,

In their characterization, the authors prove a clustering structure. For instance when value matrix is identity, then the authors show that the depending on the initial condition the output of dynamics of self attention converges to one of the vertices of a polytope. In another setting, the authors show that the output converges to a point that lies on one of three hyperplanes. All of these results point to a clustering phenomenon at the output of the transformer. This phenomenon has strong parallels to neural collapse studied in standard supervised learning setting.

**Strengths:**

This is a very well written paper. It studies a very important problem of understanding the core of transformers, i.e., self attention.

Originality: The paper has quite an original idea and approach to solve the problem.

Quality: This is a high quality paper that can inspire important developments on the path of demistifying transformers.

Clarity: The paper is a joy to read.

Significance: I believe that the paper takes an important step in the right direction and can inspire the theory community in machine learning to furthering our understanding of transformers.

**Weaknesses:**

The paper does not have significant weaknesses. I do have some questions and concerns that I would want to ask.

1. I see that the convergence analysis and result that self attention converges to a low rank matrix is given for the case of d=1. What about the more general setting? This seems to be true empirically but what can we say in theory.

2. I see that the authors state that discrete time system's analysis is straightforward. Intuitively, it feels not so straightforward and should rely on assumptions on length of step size. Can the authors elaborate?

3. The authors shed some light on the ALBERT's pretrained matrices satisfying the condition in the theorem in some cases. A more extensive analysis of this can be quite useful to understand if these assumptions are indeed necessary to achieve such a clustering structure in practice or they are a matter of theoretical convenience.

4. I would appreciate if authors could add some more intuition on what is geometric meaning of Definition 1.

5. While I understand that authors have focused only on self attention purely. This makes a lot of sense from theory point of view. From an empricial point of view, I believe there is value in doing numerical experiments to see how sensitive are the clustering illustrations to addition of an MLP layer on top.

**Questions:**

Please see the weakness section.

**Limitations:**

The authors point to several open problems and directions. They can be interpreted as gaps that can still be filled. A more extensive discussion on limtiations can be useful.

---

> ### Author Rebuttal · Authors · 2023-08-09
>
> We thank the reviewer for their feedback, and respond to each point individually below.
>
> **Weaknesses.**
>
> 1. A result similar to Theorem 2.1 certainly holds for $d\geq 2$ and $V=I_d$, but its statement and proof would need the exclusion of numerous pathological and non-generic initial configurations of tokens. The proof is anticipated to be highly technically challenging (and admittedly tedious), and we have chosen to postpone it to future work.
>
> 2. We agree with the referee, we will give more details on the discrete-time setting in the camera-ready version. Note that we impose that $I_d+V\Delta t$ is invertible, and this is implicitly is an assumption on the step size, which holds for instance for sufficiently small $\Delta t$. Let us give here some details on the proof of Theorem 3.1 in the discrete-time setting, following the intermediate results of Appendix C.1. First, Proposition C.2 (convex hull shrinkage) holds intuitively because for any $i$ and $k$, $z_i^{[k+1]}=\frac{1}{1+\Delta t}(z_i^{[k]}+\Delta t\sum_{j} P_{ij}^{[k]}z_j^{[k]})$ belongs to the convex hull of the $z_j^{[k]}$. Then we define the candidate set of limit points as in (35), and Claim 1 holds without any change in the statement and in the proof. Then, as in Steps 2 and 3 in Section C.1.2, we first prove that if $z_i^{[k]}$ is not already close to one point of the candidate set, it will keep moving toward the boundary, and finally we prove that tokens cannot circulate indefinitely between different points on the boundary. This proves convergence of each token toward a point of the set given by (35).
>
> 3. We believe that some assumptions are purely made out of theoretical convenience.
> As indicated to Referee Qc4c in Weakness 1, we have carried out additional experiments that supplement our claims beyond assumptions made for the theory.
> For instance, the assumption $Q^\top K\succ 0$ in Theorems 3.1 and 5.1 does not appear necessary in experiments.
>
> 4. The first condition in Definition 1 means that $V$ has one dominant eigenvalue: the evolution, which shares some similarities with the ODE $\dot{x}=Vx$, is quicker in the eigenspace $\mathrm{span}(\varphi_1)$ (suppose $V$ diagonalizable for simplicity) of the dominant eigenvalue. Therefore, after some time (or equivalently, after a few layers), the tokens have the corresponding coordinate much larger than the other coordinates, and this is one of the key ingredients in our analysis.
> The second condition means that $Q^\top K$ is positive definite but solely on the subspace $\mathrm{span}(\varphi_1)$. This condition is useful to guarantee that the components of $x_i$, $x_j$ along $\mathrm{span}(\varphi_1)$ represent the principal contribution to the weight $\exp(\langle Qx_i,Kx_j\rangle)$.
>
> 5. This can be done in two ways. The first way consists in first running the pure self-attention dynamics up to time $T$ (or equivalently, for $O(T)$ layers), with then applying a pure MLP to the concatenated vector of clustered features at time $T$. This amounts to seeing the MLP as a map from $\mathbb{R}^{nd}$ to $\mathbb{R}^{nd}$, which can be studied independently by existing theory.  The second way consists in alternating the self-attention and the MLP layers. In this case, clustering in the sense of Theorems 3.1 and Theorems 5.1 should not be expected since the weights of the MLP play the role of a value matrix $V$, and the conclusions of these theorems strongly depend on the identity-like structure. In Figure 3 of the PDF we illustrate an extension of Theorem 4.1 in the presence of such an MLP layer.

---

> > ### Comment · Reviewer_U2EX · 2023-08-16
> > **Thanks**
> >
> > I thank the authors for these clarifications. I would appreciate if the authors can revise the paper in the light of the above discussion. I am happy to maintain my score for the paper.

---

### Official Review · Reviewer_Qc4C · 2023-07-10

**Soundness:** 4 excellent
**Presentation:** 3 good
**Contribution:** 3 good
**Rating:** 7
**Confidence:** 4

**Summary:**

This paper studies the asymptotic behavior of a sequence of tokens processed by infinitely deep self-attention only Transformers, viewed as interacting particle systems (1).

The authors first study the one dimension case and show that the self-attention matrix converges to a low-rank boolean matrix.

They then focus on particular choices of the value matrix $V$ to obtain clustering in higher dimension. Specifically, when $V = I_d$, the paper shows that a time-rescaled version of the tokens converge to the boundary of a convex polytope.

The paper then focuses on more realistic choices of $V$. When it has a single leading eigenvalue, the clustering happens toward one of at most $3$ hyperplanes. When the leading eigenvalue has multiplicity, the limit geometry is a convex polytope in some directions and a linear subspace in the others.

All the theoretical results are numerically illustrated.


**Strengths:**

1. The paper is very clear and well written.
2. The overall contribution is strong, as it is the first paper provably showing the emergence of clusters in self-attention only Transformers.
3. More specifically, each theorem, (well summarized in Table. 1) is an interesting result of clustering in interacting particles systems.
4. Most assumptions on (Q, K, V) are clearly discussed.
5. The figures illustrating the theorems are well presented and insightful.

**Weaknesses:**

1. Regarding the assumption in Th. 3.1 and 5.1 that $Q^TK > 0$ is positive definite: I think that the strength of this assumption should be emphasized. See Questions.

2. An intuitive explanation (as for instance the one made after th 4.1 (l. 257 to 262)) would be welcome after th 3.1. Especially, how does the convex polytope $K$ depend on $Q^TK$ ?

Typo:
l. 242, I believe a $Q$ is missing in the quadratic form.



**Questions:**

1. What is the main obstacle for considering the time-dependent dynamics where $Q$, $K$ and $V$ depend on time, when $d = 1$ for instance ? Are there some assumptions that can be made on the time dependency to obtain similar results ?

2. Regarding Weakness 1. For instance, in practical implementation, I believe $Q^TK$ is low-rank. Maybe it should also be clarified that by $Q^TK > 0$  you also mean symmetric ? In practical implementation, is $Q^TK$ close to a symmetric matrix ?

3. In Th 4.1, you do not assume anymore that $Q^TK$ is symmetric ? Have you verified whether the second condition in Definition 1. holds on some pre-trained triple (Q, K, V) on ALBERT ?



**Limitations:**

Yes.

---

> ### Author Rebuttal · Authors · 2023-08-09
>
> We thank the reviewer for their feedback, and respond to each point individually below.
>
> **Weaknesses.**
>
> 1. We recognize that this assumption is substantial. However, it does not appear to be necessary for our conclusions; rather, it serves to direct the proof. To reinforce the broader validity of our conclusion beyond just this specific assumption, we have carried out additional experiments (Figures 1, 2 in the PDF), suggesting that our clustering results are more universal.
>
> 2. Theorem 3.1 amounts to a couple of effects entailed by the dynamics. First of all, the convex hull of the particles is shrinking over time (Proposition C.2). This is due to the fact that the distance of the particle nearest to any half-space (not containing the particles) is decreasing. On the other hand, the convex hull ought not collapse since particles which have not concentrated near the boundary of the limiting polytope will continue to increase in magnitude until they themselves reach this boundary (Step 2 in the proof). The latter is due to the time-rescaling and the "probabilistic" structure of the self-attention mechanism. (A version of this comment will be added to the camera-ready version.) As for the convex polytope $\mathcal{K}$--it depends both on the initial sequence of particles and on $Q^\top K$. Unfortunately, we do not believe there is a way to predict $\mathcal{K}$ explicitly besides running the full dynamics.
>
> We also thank the Reviewer for spotting this rather important typo.
>
> ---
> **Questions.**
>
> 1. We believe that if we assume that $Q(t)^\top K(t)$ is bounded from below and above by positive multiples of the identity uniformly in $t$, the same conclusions as in Theorems 2.1, 3.1, 4.1 and 5.1 hold. This requires adaptations in the proofs. However, the conclusions of the theorems do not hold in general if $V$ depends on time.
>
> 2. Indeed $Q^\top K\succ 0$ means in particular that $Q^\top K$ is symmetric (the partial order within the set of symmetric matrices). We agree with the Reviewer that this assumption is at odds with the low-rankness in practical implementations, but, as alluded to in the answer to Weakness 1, we do not believe that it is essential to the result. However, at this moment, we can't identify a clear method to eliminate it from our proof.
>  It does not appear evident that the pre-trained ALBERT matrices are approximately symmetric.
>
> 3.  Indeed, no symmetry or positive-definiteness of $Q^\top K$ is required. Since we only look at the behavior of the particles along the direction given by $\varphi_1$, it suffices to assume that the induced quadratic form is positive only along said direction. With regard to the pre-trained triple: the first point is satisfied by $V_5$ and $V_{14}$ (indicated in the Supplementary material), and the second point is too satisfied by $(Q_5, K_5)$ and $(Q_{14}, K_{14})$ (the inner products evaluated at the eigenvector of norm 1 equal 1.3060 and 0.6719 respectively). In other words, the triples $(Q_h, K_h, V_h)$ corresponding to heads $h=5$ and $h=14$ in ALBERT satisfy the assumptions of Definition 1. This comment will be added to the Supplementary Material of the camera-ready version.

---

### Author Rebuttal · Authors · 2023-08-09

We thank the reviewers for their thoughtful feedback. We echo their concerns with regard to several assumptions we had made on the weight matrices for our analysis.

All in all, our goal was to consider the simplest setting of transformers amenable to rigorous mathematical analysis. To enhance the validity of the clustering theorems, we have conducted a few numerical experiments which violate the assumptions made for the development of the theory and observed that the clustering pattern persists. Namely,
-  In Figure 1, we replicate the experiment illustrating Theorem 3.1 in the setting where $Q^\top K$ is a random matrix with entries sampled from $\mathrm{Unif}([-1,1])$. The clustering pattern persists even for this choice of weights.
- In Figure 2, we replicate the experiment illustrating Theorem 5.1 in the setting where $Q^\top K$ is a random matrix with entries sampled from $\mathrm{Unif}([-1,1])$. The same clustering pattern appears.
- In Figure 3, we add an additional MLP layer in the setup of Theorem 4.1. More precisely, we use a 2-layer neural network: we apply a component-wise activation function (either ReLU or Tanh), and then multiply by a time-independent weight matrix $W$. We again see clustering, the pattern depending on the weight matrix $W$ and on the activation function.
    In particular, in the first row (corresponding to $W=I_d$ and ReLU), we see that the particles first evolve as to reach the upper right quadrant $\mathbb{R}^d_{>0}$ (due to the ReLU). Once they reach it,
    every particle eventually follows one of three hyperplanes determined by the spectrum of $V$ and the projection onto $\mathbb{R}^d_{>0}$.
    In the other two cases, all the particles appear to collapse to $0$.

All experiments were conducted following the setup presented in Appendix F of the Supplementary Material. They will be added and documented in the Supplementary Material of the camera-ready version.
We comment on the difficulty of generalizing the theory to more general scenarios in the point by point replies to individual reviewers.

---

### Decision · Program_Chairs · 2023-09-21

**Decision:**

Accept (poster)

**Comment:**

This paper studies the asymptotic behavior of a sequence of tokens processed by a self-attention network and viewed as interacting particle systems. More precisely, Eq. (1) gives a differential equation for the evolution of the tokens $x(t)$ as a function of a self-attention matrix $P(t)$ (this result was derived in prior work). The main contribution of the paper is to show that the matrix $P(t)$ converges to a low-rank matrix with $0,1$ entries (this is for instance illustrated in Figure 2). Effectively, $P(\infty)$ shows the existence of clusters between tokens. This is an interesting perspective to study self-attention, which is still not well understood from an analytical perspective.

The reviewers are in general positive about the paper, they for instance find it clearly written, with an interesting contribution and the potential to lead to novel research directions. I therefore recommend acceptance. Please include the changes discussed in the rebuttal to the camera-ready version.